# Annexin- and calcium-regulated priming of legume root cells for endosymbiotic infection

Ambre Guillory[1,5], Joëlle Fournier[1,5], Audrey Kelner[1,5], Karen Hobecker[2], Marie-Christine Auriac[1], Lisa Frances[1], Anaïs Delers[1], Léa Pedinotti[3], Aurélie Le Ru[4], Jean Keller[3], Pierre-Marc Delaux[3], Caroline Gutjahr[2], Nicolas Frei Dit Frey[3] & Fernanda de Carvalho-Niebel[1] ✉

Legumes establish endosymbioses with arbuscular mycorrhizal (AM) fungi or rhizobia bacteria to improve mineral nutrition. Symbionts are hosted in privileged habitats, root cortex (for AM fungi) or nodules (for rhizobia) for efficient nutrient exchange. To reach these habitats, plants form cytoplasmic cell bridges, key to predicting and guiding fungal hyphae or rhizobia-filled infection thread (IT) root entry. However, the underlying mechanisms are poorly studied. Here we show that unique ultrastructural changes and calcium ($Ca^{2+}$) spiking signatures, closely associated with *Medicago truncatula* Annexin 1 (MtAnn1) accumulation, accompany rhizobia-related bridge formation. Loss of *MtAnn1* function in *M. truncatula* affects $Ca^{2+}$ spike amplitude, cytoplasmic configuration and rhizobia infection efficiency, consistent with a role of MtAnn1 in regulating infection priming. *MtAnn1*, which evolved in species establishing intracellular symbioses, is also AM-symbiosis-induced and required for proper arbuscule formation. Together, we propose that MtAnn1 is part of an ancient $Ca^{2+}$-regulatory module for transcellular endosymbiotic infection.

Plants benefit from associations with microbes in the soil which improve the acquisition of essential nutrients for growth. These associations include the ancestral symbiosis with arbuscular mycorrhizal (AM) fungi providing phosphorous, or the more recently evolved interactions with endosymbiotic bacteria for nitrogen acquisition[1–3]. Microbial endosymbionts are hosted in privileged intracellular niches, where optimal conditions are met for efficient nutrient exchange (e.g. the root cortex hosting arbuscules formed by AM fungi, or root nodules, hosting nitrogen-fixing bacteria). Extensive studies on model legumes such as *Medicago truncatula*, combined with evolution-based studies across a broad species range revealed that nodule-forming nitrogen-fixing endosymbiosis (RNS) evolved ~100 million years ago in species from related angiosperm plant

lineages, by co-opting signalling components from the ancient AM symbiosis and recruiting new key genes through neo-functionalization and/or rewiring of expression[1,3,4]. These endosymbioses share a common symbiotic pathway that uses calcium ($Ca^{2+}$) as a key secondary messenger to trigger downstream signalling[5,6]. Upon perception of mycorrhizal Myc and rhizobial Nod factor signals, $Ca^{2+}$ oscillations (spiking) are triggered in and around the root hair nucleus through the concerted action of nuclear envelope channel/pump complexes[5–7]. While $Ca^{2+}$ spiking is critical in the nucleus (after its decoding by $Ca^{2+}$/calmodulin-dependent protein kinases CCaMK or Does not Make Infection, DMI3)[8,9] to regulate host transcriptional reprogramming[10–12], the functional relevance of sustained $Ca^{2+}$ spiking in the cytoplasmic compartment has not yet been established. The rhizobia- and AM-

[1]LIPME, INRAE, CNRS, Université de Toulouse, Castanet-Tolosan, France. [2]Max-Planck-Institute of Molecular Plant Physiology, Potsdam-Golm, Germany. [3]LRSV, Université de Toulouse, CNRS, UPS, Toulouse INP, Castanet-Tolosan, France. [4]FRAIB-TRI imaging platform, Université de Toulouse, CNRS, UPS, Castanet-Tolosan, France. [5]These authors contributed equally: Ambre Guillory, Joëlle Fournier, Audrey Kelner. ✉e-mail: fernanda.de-carvalho-niebel@inrae.fr

induced *MtAnn1*[13] and *MtAnn2*[14] genes, which encode cytosolic $Ca^{2+}$-binding annexins that can bind to membranes to modulate a variety of $Ca^{2+}$-regulated processes (e.g. ion conductance, exocytosis/endocytosis)[15,16], are potential players in this process.

Following signal exchange to reach compatibility, infection of root tissues by AM fungi or rhizobia bacteria occurs in most legumes transcellularly, through the de novo construction of apoplastic compartments, delimited by host cell wall/ membrane interfaces, which physically separate microsymbionts from the host cytoplasm[17,18]. The hyphae of AM fungi progress through epidermal and cortical root tissues until the fungus reaches the inner cortex to form highly branched arbuscules inside plant cells. Arbuscules comprise a massive network of membrane tubules at the symbiotic host-fungus interface[19,20], which likely ensures the release of nutrients, particularly phosphate, to the plant. In RNS, rhizobia enter the roots of most legume species via apoplastic tubular structures called infection threads (ITs)[18,21,22] that grow from root hairs to the developing nodule underneath[23] under the influence of key plant hormones, intercellular signalling and long-distance regulatory pathways[24–26].

ITs are formed in root hairs after rhizobia are entrapped and enclosed in a globular infection chamber[27]. This plant-driven process requires targeted cell wall remodelling at the IT interface[27,28] for polar growth, under the control of the infectosome protein complex[29–31]. Transcellular IT infection also imposes tight regulation of the plant's cell cycle status through the recruitment of mitotic regulators in root hairs and acquisition of specific pre-mitotic states before and during IT passage in the cortex[32–34]. Finally, the route of IT entry is also fine-tuned by the host, which creates a broad cytoplasmic column filled with endoplasmic reticulum (ER) and surrounded by microtubule arrays to guide IT progression[35,36]. In root hairs, this bridge connects the tip of the IT to the nucleus to direct IT elongation. In the adjacent cortex, differentiated root cells with a large central vacuole enter an activated state (hereafter termed pre-infection priming), with characteristic transvacuolar cytoplasmic strands emanating from the nucleus before a transvacuolar cytoplasmic bridge, the pre-infection thread (PIT[37]), forms in anticipation of future IT passage. PIT bridges have been reported to occur in nodules of several legume and actinorhizal species[37–39]. Moreover, PIT bridges closely resemble Pre-Penetration Apparatus (PPA)[17,40] cytoplasmic columns that precede AM fungal hyphae transcellular passage. The fact that PIT/PPA-like remodelling is not seen in pathogen interactions[41], suggests their specific and probably ancient recruitment for endosymbiosis. Parallels have been drawn between PIT/PPA and premitotic phragmosome transvacuolar bridges formed in cells preparing for mitosis, as these processes share similarities in tightly controlling plant cell division status, membrane trafficking and microtubule dynamics[32,34,39,42].

The development of fluorescent protein fusions labelling the ER network has made it possible to visualise the dynamics of ER-enriched bridge remodelling in vivo during PPA/PIT formation[35,40,43]. Complementary in vivo studies using Cameleon $Ca^{2+}$ sensors revealed that primed cortical cells exhibit a low-frequency $Ca^{2+}$ spiking signature that switches to high frequency at the onset of infection by AM fungi or rhizobia[43]. CCaMK/DMI3 is critical for PPA formation in *Medicago*[41]. Moreover, a gain-of-function variant in *Lotus* can trigger cytoplasmic remodelling in the cortex in the absence of symbionts[44]. Thus, CCaMK/DMI3-mediated signalling can bypass early symbiotic activation to trigger the priming response in the cortex. These studies suggest that cytoplasmic bridge reprogramming involves fine-tuning of $Ca^{2+}$ spiking signatures, but the underlying genetic pathways or molecular players have not been identified.

In this study, using resolutive microscopy methods in different *M. truncatula* genetic backgrounds, we provide in-depth insights on ultrastructural rearrangements and in vivo $Ca^{2+}$ signalling dynamics in plant cells preparing for rhizobia infection. We show that an intimate link between specific $Ca^{2+}$ spiking signatures and the accumulation of annexin MtAnn1 characterises this priming response. MtAnn1 is a clear indicator of cytoplasmic reprograming for infection, and its expression in this context is induced by the master symbiotic transcription factor Nodule Inception (NIN)[1]. Through phylogenetics and mutant phenotyping, we provide evidence that *MtAnn1*, recruited during evolution in plants establishing root endosymbioses, likely plays an ancestral role for successful rhizobia and AM fungi infection. We propose that *MtAnn1* was recruited to shape $Ca^{2+}$-regulated transvacuolar cytoplasmic rearrangements for efficient endosymbiotic infection.

## Results

### Cell-specific ultrastructural changes in plant cells prior to IT entry

PIT bridge formation is a widespread process in RNS nodules[37,38], but detailed studies of this process are lacking. Previous work in *M. sativa* and *M. truncatula skl* mutant roots showed that microtubule rearrangements accompany PIT formation[39], however overall cellular features underlying this process remain unknown. Here, we set up an adapted methodology to gain access to ultrastructural changes underlying rhizobia pre-infection priming in *M. truncatula* roots. Segments of early infected roots, harvested 5–6 dpi with rhizobia, were prepared for light (Fig. 1a, b and Supplementary Fig. 1a) and transmission electron microscopy (TEM) (Fig. 1c–g) using an Epon-embedding procedure. These analyses revealed specific cytoplasmic rearrangements in primed outer cortical cells (C1) next to an infected root hair site (Fig. 1a–c) compared to mock control (Fig. 1h–j) root samples. TEM analyses showed that the thick cytoplasmic bridge was highly enriched in ER (white arrows, Fig. 1e) and large quantities of small vacuole-like structures with sizes ranging from 0.5 to 5 μm (v in Fig. 1e). These activated cells also featured significant and unusually high mitochondrial clustering (asterisks, Fig. 1e) around the nucleus. This mitochondria-nuclei association was consistently observed in independent pre-infection (PI) primed cells compared to actively dividing cells in the nodule primordia (NP) (Fig. 1d, e and Supplementary Fig. 1b–k). At a later stage, when the IT had progressed across this small vacuole- and ER-rich cytoplasmic bridge (Fig. 1f), the nucleus was often found in close proximity to the IT (Fig. 1g). Thus, the plant nucleus not only guides the progression of ITs but also establishes close physical interactions with them. Overall, drastic ultrastructural changes take place specifically in outer cortical cells (C1 or C2) primed for rhizobia infection, namely a unique transcellular cytoplasmic organisation, enriched with ER, small vacuole-like structures and a significant increase in mitochondria-nucleus association.

### MtAnn1 and $Ca^{2+}$ spiking dynamics mark symbiotically engaged and infected plant cells

$Ca^{2+}$ spiking responses are transiently regulated in cortical cells preparing for infection[43], but how these responses relate to cytoplasmic remodelling is unknown. To study this, we used an in vivo approach based on co-imaging a fluorescent $Ca^{2+}$ sensor and a cytoplasmic marker to simultaneously monitor $Ca^{2+}$ spiking and cytoplasmic remodelling in individual cells. We used *M. truncatula* A17 (wild-type) and the supernodulating mutant *sunn*, which shows wild-type infection but at higher levels, facilitating live imaging studies of early rhizobia infection events[27,29,45]. These in vivo analyses cannot provide as much data from the wild-type background in which infection thread events are more rare than in *sunn*. However, results were confirmed in the two genetic backgrounds (A17, *sunn*) by independent experiments. Furthermore, as sites of interest were individually imaged multiple times in living roots, responses were dynamically monitored and confirmed overtime by follow-up observations.

*S. meliloti*-inoculated roots

Primed cell

IT passage

IT close to the nucleus

Framed area in b

Framed area in c

Control roots

Framed area in i

We used the red fluorescent nuclear NR-GECO1 sensor for $Ca^{2+}$ imaging because of its better compatibility with the green fluorescence cytoplasmic marker and higher sensitivity to detect symbiotic $Ca^{2+}$ spiking, as compared to a Förster resonance energy transfer (FRET) cameleon sensor[46]. To validate this sensor in our system, we first monitored $Ca^{2+}$ spiking in *S. meliloti*-responsive root hairs of the

nodulation susceptible zone, just below the infection zone, in both A17 and *sunn*, 1–4 dpi with rhizobia. Using NR-GECO1, we confirmed that *S. meliloti* triggers periodic, high-frequency $Ca^{2+}$ spiking in these root hairs with profiles indistinguishable from those induced by purified Nod factors, as previously reported[46,47] (Supplementary Fig. 2 and Movies 1, 2). Subsequent analysis of rhizobia-infected sites in *sunn* at

**Fig. 1 | Cell-specific structural changes in *M. truncatula* cells primed for infection. a, b** Representative images of longitudinal sections of *M. truncatula* A17 roots 5 days after inoculation with *S. meliloti*. These are consecutive 1 μm sections stained with Basic Fuchsin to reveal cell outlines and contents. Arrows indicate ITs in root hair epidermal cells and arrowheads indicate a C1 cortical cell exhibiting pre-infection priming. After analysis of 200–250 sections (1 μm) per *S. meliloti*-inoculated root segment, 7 individual pre-infection priming events were captured, 5 of which were further analysed by TEM (data are from 3 independent experiments). The primed cell (arrowhead) and framed area in (**b**) are shown in 80 nm TEM sections in (**c–e**). Note that the primed C1 outer cortical cell (arrowhead in **a–c** and enlarged image in **e**) exhibits a unique organisation compared with neighbouring cells. Primed cells have a cytoplasmic bridge enriched in endoplasmic reticulum (white arrows in **e**), and numerous vesicles or small vacuole-like structures (v). They also have a large nucleus (n) with mitochondria (asterisks) clustering around (quantified in Supplementary Fig. 1). **f, g** At a later stage, when the IT (arrows) has progressed across the cytoplasmic bridge towards inner tissues, the nucleus of the crossed cell appears closely associated to the IT ($n = 4$, 2 independent experiments) (**g**). **h–j** Representative images of *M. truncatula* A17 root sections after mock (water) treatment for 5 days (2 independent experiments). **h** A 1 μm longitudinal section of a mock control root stained with Basic Fuchsin. **i, j** TEM of 80 nm sections of a mock control root. The framed area in (**i**) is shown in (**j**). Note that non-symbiotically activated cells have a large vacuole and nuclei near the cell periphery. Abbreviations: cortical cells (C1, C2, C3), nucleolus (nuc). Scale bars: **a, b, h** = 50 μm, **c, i** = 20 μm, **d–g, j** = 5 μm. See also Supplementary Fig. 1. Source data are provided as a Source Data file.

2–7 dpi, revealed specific low frequency $Ca^{2+}$ spiking profiles in the outer cortex underlying rhizobia infection sites (Supplementary Fig. 3). Cortical cells that spiked included those that later became infected (arrowhead, Supplementary Fig. 3). Furthermore, only cortical cells in close contact with the infected epidermal cell spiked, while other distant cortical cells did not. This suggests that potential short distance signalling from the infected cell is required for pre-infection priming of cortical cells. Our results are consistent with previous data using a FRET-cameleon $Ca^{2+}$ probe[43], and confirm that low-frequency $Ca^{2+}$ spiking is associated with rhizobial pre-infection priming of cortical cells.

*MtAnn1*, encoding a cytosolic $Ca^{2+}$ and membrane-binding annexin protein, is induced by rhizobia both in root hairs and in outer cortical cells preparing for infection[13,48], where $Ca^{2+}$ spiking responses are coincidently transiently regulated (Supplementary Fig. 3)[43]. Thus, the use of a cytosolic MtAnn1-GFP fusion expressed under its native rhizobia-induced promoter seemed a suitable tool to monitor cytoplasmic remodelling during rhizobia infection. Comparative localisation analyses were performed in *M. truncatula* A17 and/or *sunn* genotypes using a GFP-ER marker, which labels the cytoplasmic ER endomembrane network and the MtAnn1-GFP fusion. This revealed similar labelling of root hair cytoplasmic bridges by both markers (Supplementary Fig. 4). Co-localisation studies confirmed that MtAnn1-GFP and a red mcherry-ER fusion co-labelled cytoplasmic bridges in both root hair (Supplementary Fig. 5) and cortical cells (Supplementary Fig. 6). MtAnn1-GFP fusion expressed under native *MtAnn1* promoter specifically labelled cytoplasmic bridges of pre-infection-primed (C1a in Supplementary Fig. 6) and infected OC cells (C1b in Supplementary Fig. 6), whereas no labelling was observed in surrounding OC cells expressing cytoplasmic red mCherry-ER marker (driven by p35S). Combined, these data support the conclusion that the MtAnn1-GFP fusion specifically marks cytoplasmic bridges, where it partially co-localises with the ER network.

Co-expression of red NR-GECO1 $Ca^{2+}$ sensor and green MtAnn1-GFP fusion was then used to live track the spatio-temporal dynamics of $Ca^{2+}$ spiking and MtAnn1 accumulation simultaneously in *M. truncatula* A17 (Fig. 2). The MtAnn1/NR-GECO1 double-labelling greatly facilitated detection of root hairs in early stages of bacterial entrapment (RHE) and IT polar growth (IT) (Fig. 2a, b). RHE- or IT- root hairs consistently showed both MtAnn1-GFP fusion signal and high frequency $Ca^{2+}$ spiking profiles (Fig. 2a–e). However, the amplitude of $Ca^{2+}$ spikes dropped significantly from RHE to IT stage root hairs (Fig. 2c, d, f). Conversely, relative levels of MtAnn1-GFP fluorescence tended to increase in IT versus RHE root hairs (Fig. 2a, b and Supplementary Fig. 7), consistent with root hair transcriptome data[48]. Overall, our data suggest that decreased $Ca^{2+}$ spiking amplitude and increased MtAnn1 levels are associated with polar IT growth.

MtAnn1-GFP also labelled cytoplasmic bridges of cortical cells primed for infection in both wild-type A17 and *sunn* (Fig. 3a–c, Supplementary Figs. 6 and 8). MtAnn1-GFP-labelled cells were those in close contact with the infected root hair, which consistently showed

low frequency $Ca^{2+}$ spiking (Fig. 3d, e). Furthermore, this low-frequency $Ca^{2+}$ spiking profile switches to a high-frequency pattern when the cortical cell is infected (Supplementary Fig. 8), consistent with previously published data[43]. Strikingly, nearby cortical cells not labelled with MtAnn1 did not show any spike in both A17 and *sunn* (Fig. 3b–e and Supplementary Fig. 8). Collectively, our data establish that IT development and pre-infection priming involve stage- and cell-type-specific $Ca^{2+}$ spiking signatures closely linked to MtAnn1-GFP bridge accumulation.

### Rhizobia root hair infection is required for pre-infection priming of adjacent cortical cells

MtAnn1-GFP labelling and $Ca^{2+}$ spiking mark primed outer cortical cells (Fig. 3 and Supplementary Fig. 8), but it is unclear if IT formation is required for this process. Co-expression of NR-GECO1 and MtAnn1-GFP in infection-defective *ern1* (*ERF Required for Nodulation 1*)[49–51], revealed that $Ca^{2+}$ spiking and cytoplasmic-bridge formation are still seen in root hairs with arrested chambers (Supplementary Fig. 9a–c). MtAnn1-GFP labelling and $Ca^{2+}$ spiking were also seen in the outer cortex adjacent to root hairs with arrested-infections in *ern1* (Supplementary Fig. 9d–f), but the pattern appeared to be deregulated (more cells labelled with MtAnn1-GFP or $Ca^{2+}$ spiking sometimes in *MtAnn1* unlabelled cells, never seen in the wild-type control A17 or *sunn*, Fig. 3 and Supplementary Fig. 8). This suggests that ERN1 is not required to promote pre-infection priming, but in its absence the priming response is deregulated.

CCaMK/DMI3 is critical for PPA formation in *Medicago*[41], however, it is unknown if it is also required for PIT formation. Since *dmi3*[8] is impaired in root hair infection, it is not possible to access PIT formation in this genetic background. We thus took advantage of stable transgenic lines complemented with DMI3 only in epidermis[52], to investigate if $Ca^{2+}$ spiking and MtAnn1-GFP could still label cortical cells in this line compared to wild-type A17 control (Fig. 3). Complemented *dmi3 pEXT:DMI3* lines showed $Ca^{2+}$ spiking, rescued IT development and MtAnn1 labelling in root hairs (Supplementary Fig. 9g and Movie 3). However, no $Ca^{2+}$ spiking or MtAnn1-GFP labelling was seen in the cortex (Supplementary Fig. 9h), compared to the control A17 or *sunn* (Fig. 3 and Supplementary Fig. 8). No cytoplasmic remodelling in the cortex of *dmi3 pEXT:DMI3* could be detected by further analysis of a constitutively expressed GFP (Supplementary Fig. 9i, j). Thus, MtAnn1-labelled cytoplasmic bridges were clearly visible only in infected root hairs of *dmi3 pEXT:DMI3* and not in adjacent cortical cells (no MtAnn1 labelling or GFP-labelled cytoplasm remodelling, Supplementary Fig. 9). This suggests that PIT infection priming uses a DMI3-dependent pathway that must be active in the cortex.

Taken together, these analyses indicate that root hair infection chamber formation is sufficient in *ern1* to trigger MtAnn1-GFP-labelled bridge in the root hair itself and adjacent cortex. The absence of cortical priming in *dmi3* is consistent with the conclusion that, like in PPA, DMI3 is required for PIT reprogramming.

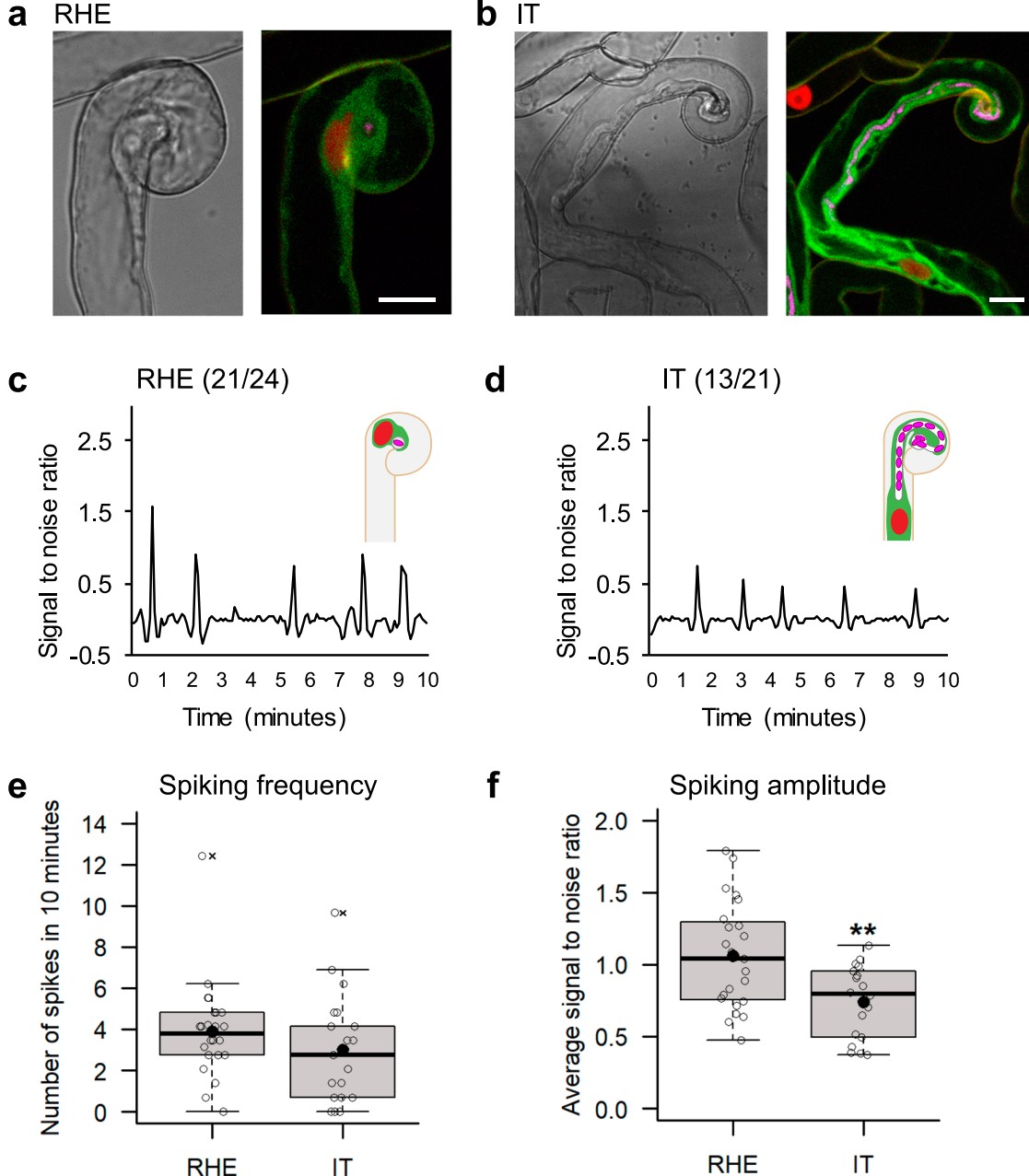

**Fig. 2 | Ca²⁺ spiking amplitude drops along root hair IT development. a, b** Representative bright-field and corresponding confocal fluorescence images of **a** a root hair with entrapped CFP expressing (magenta)-*S. meliloti* (RHE) and **b** a root hair with a growing infection thread (IT) in *sunn*. Nuclei expressing the NR-GECO1 Ca²⁺ sensor appear in red, and the MtAnn1-GFP fluorescence (green) labels the cytoplasmic zone around the IT and the cytoplasmic bridge connecting and surrounding the nucleus. Representative Ca²⁺ spiking traces of RHE (**c**) and growing IT stages (**d**) in *sunn*. The relative concentration of Ca²⁺ ions in the nucleus is reflected by the intensity of NR-GECO1 fluorescence, expressed as signal-to-noise ratio (SNR, cf. 'Methods' section). The number of root hairs with nuclear spiking/total number of root hairs are indicated between parentheses. Nuclei are counted as spiking when showing more than 2 peaks in 10 min. Quantification of nuclear Ca²⁺ spiking: spiking frequency (**e**), expressed as number of spikes in 10 minutes per nucleus, and spiking amplitude (**f**), expressed as average SNR of spikes per nucleus. Box plots represent the distribution of individual values (indicated by open circles) from root hairs with entrapped rhizobia RHE, (*n* = 24 in **e** and *n* = 23 in **f**) or with an IT (*n* = 21 in **e** and *n* = 18 in **f**) in *sunn*, 2–4 dpi with *S. meliloti* from 3 independent experiments. First and third quartile (horizontal box edges), minimum and maximum (outer whiskers), median (centreline), mean (solid black circle) and outliers (crosses) are indicated. Differences were not significant in spiking frequency in IT vs. RHE in (**e**) (*p* = 0.1531, two-tailed Mann-Whitney test). Asterisks indicate statistically significant differences in spiking amplitude in RHE vs. IT in (**f**) (*p* = 0.0037, two-tailed t-test). Scale bars: **a**, **b** = 10 μm. See also Supplementary Fig. 2 and Movies 1, 2 for Ca²⁺ spiking responses in A17 and/or *sunn* root hairs, and Supplementary Figs. 4–7 for MtAnn1-GFP localisation and fluorescence quantification in root hair cytoplasmic bridges. Source data, including split channels for merged fluorescence, are provided as a Source Data file.

## NIN controls pre-infection priming and the associated expression of *MtAnn1*

The master symbiotic transcription factor Nodule INception (NIN) is genetically downstream of DMI3 and its mutation abolishes rhizobia-induced *MtAnn1* expression[48,53]. To test if NIN is required for *MtAnn1* expression, we compared the spatio-temporal expression profile of a *pMtAnn1:GUS* fusion in *M. truncatula* A17 wild-type and *nin* roots. In control conditions, the *pMtAnn1:GUS* fusion was active in root tips,

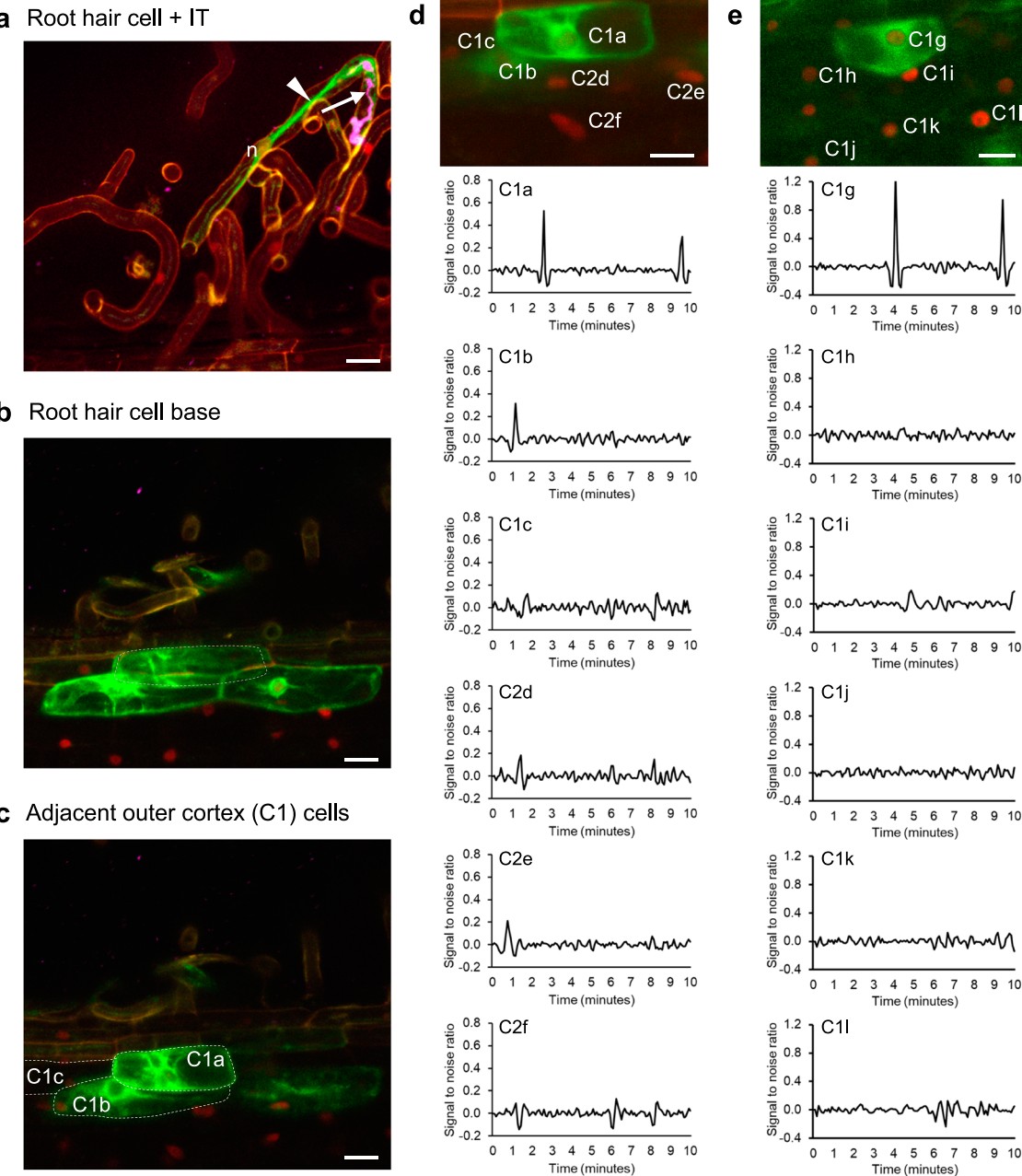

**Fig. 3 | Strong MtAnn1-GFP fusion fluorescence and low frequency Ca²⁺ spikes are hallmarks of pre-infection priming in the cortex. a–e** Representative images of rhizobia infection sites in roots co-expressing MtAnn1-GFP (green) and NR-GECO1 (red) in *M. truncatula* A17. **a–c** Confocal images illustrate an infected root hair (CFP-labelled rhizobia in the IT, arrow, in magenta) and the nucleus (n) in front, guiding the growth of the IT in a cytoplasmic bridge (arrowhead) (**a**), the base of the infected root hair cell (**b**, grey dotted line) and the neighbour C1a-c outer cortical cells (**c**, dotted lines). C1a-C1c are also shown in (**d**). **d, e** Representative Ca²⁺ spiking traces, expressed as signal-to-noise ratio (SNR), from outer cortical cells (C1a-c, C2d-f, C1g-l, top panels) adjacent to two independent infected root hair sites. Cortical cells C1a-c, C2d-f are adjacent to the infection site shown in (**a**, **b**) (C1a-c are also shown in **c**), while C1g-l cells are adjacent to another root hair infection site (not shown). Traces of cortical cells that are in direct contact (C1a, C1b, C1g) or not (C1b-c, C2d-f in **d** and C1h-l in **e**) with the infected root hair site are shown. Images were obtained from A17 roots 4 dpi (**a–d**) or 3 dpi (**e**) with CFP-labelled *S. meliloti*. Data were obtained from 2 independent experiments after the analysis of Ca²⁺ traces in *n* = 12 individual cells. Note that only cortical cells co-expressing MtAnn1-GFP (C1a-b in **d**, C1g in **e**) show detectable low frequency Ca²⁺ spiking. Images in **a-c** are maximal z-projections of sub-stacks and images in **d-e** are maximal projections of whole time series. Abbreviations: first and second outer cortex layers (C1, C2), infection thread (IT). Scale bars: **a–e** = 20 μm. See also Supplementary Figs. 3, 6 and 8, 9 for Ca²⁺ spiking responses and MtAnn1-GFP dynamics in primed cells in the cortex in *sunn* and in infection-defective *ern1* and *dmi3* mutants. Source data, including split channels for merged fluorescence, are provided as a Source Data file.

lateral roots and some endodermal cells (Fig. 4a). Upon *S. meliloti* inoculation, strong promoter activity was detected at rhizobial infection sites, both in root hairs and outer cortex (Fig. 4b), and later in other nodule primordium cortical cells[13] but not in *nin* (Fig. 4c). This confirms the NIN-dependent control of *MtAnn1* expression. Conversely, epidermal expression of NIN (*pEXPA:NIN*[54]) resulted in

*pMtAnn1:GUS* fusion activity in the absence of rhizobia in *nin* (Fig. 4d). This indicates that NIN can bypass early symbiotic signalling to promote *pMtAnn1:GUS* activity. Transactivation assays in *Nicotiana benthamiana* (Fig. 4e and Supplementary Fig. 10) showed that NIN can transcriptionally activate *pMtAnn1:GUS*. This activation was abolished when a NIN mutated version was used (Fig. 4e and Supplementary

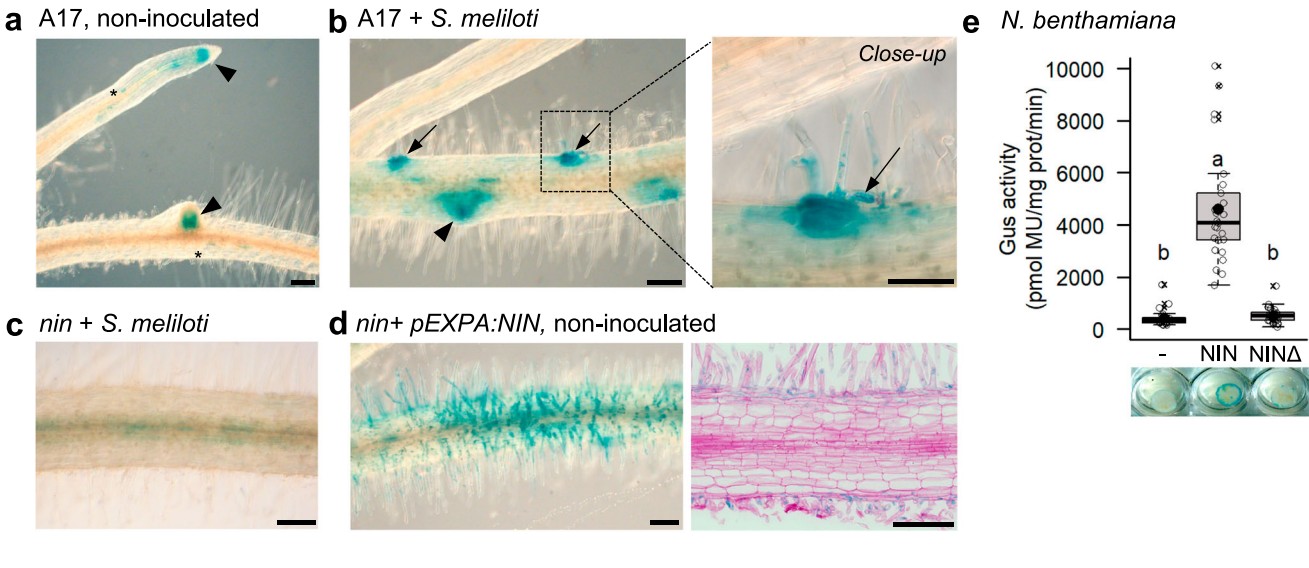

*pMtAnn1:GUS in A17 and nin*

**a** A17, non-inoculated

**b** A17 + *S. meliloti*

*Close-up*

**c** *nin + S. meliloti*

**d** *nin+ pEXPA:NIN*, non-inoculated

**e** *N. benthamiana*

*pENOD11:GFP-ER + pUBQ10:DsRed in nin*

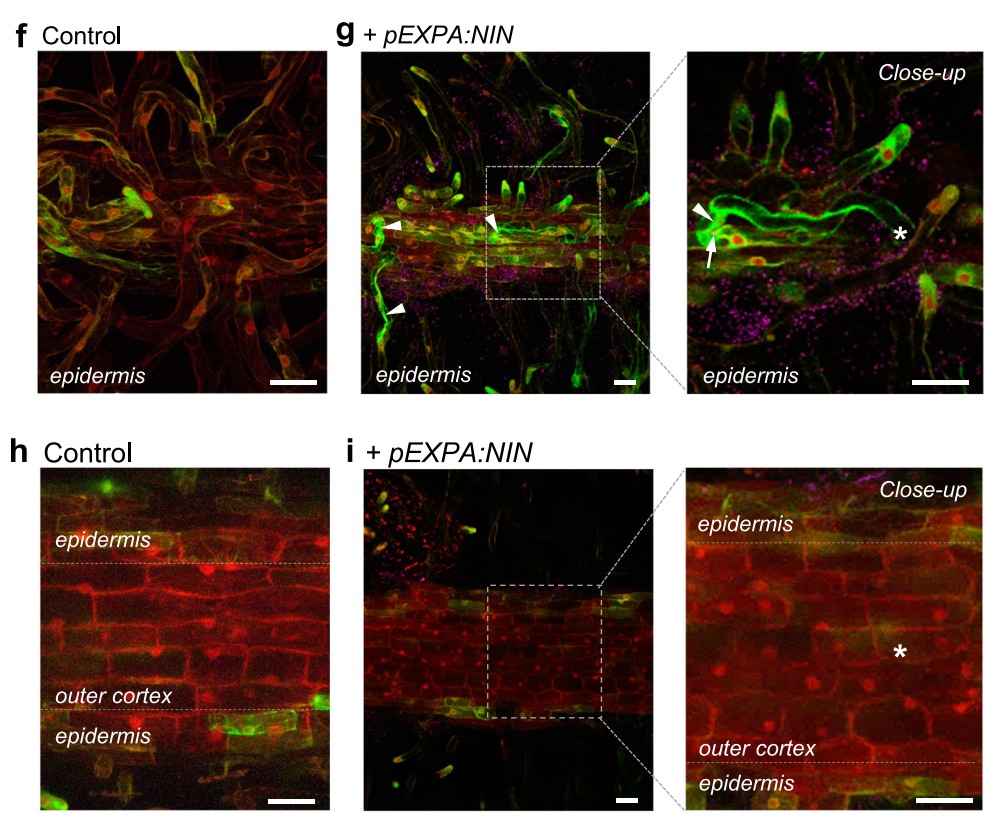

**f** Control

**g** + *pEXPA:NIN*

*Close-up*

epidermis

epidermis

epidermis

**h** Control

**i** + *pEXPA:NIN*

*Close-up*

epidermis

epidermis

outer cortex

outer cortex

epidermis

epidermis

Fig. 10). These data show that the expression of *MtAnn1* in rhizobia infection sites is under NIN dependence and suggests that NIN is a possible direct *MtAnn1* regulator.

*nin* fails to form infection chambers and ITs but can still trap rhizobia in a curled root hair[27]. The question remains if a cytoplasmic bridge can still form in *nin*. To assess this, we examined cytoplasmic bridge formation in *nin* transformed with *pEXPA:NIN* or a *pEXPA:GUS* control (Fig. 4f–i). Since *MtAnn1* is transcriptionally dependent on NIN (Fig. 4a–e), we could not use the *pMtAnn1:MtAnn1-GFP* fusion for these studies, but instead used a *pENOD11:GFP-ER* fusion co-expressed with *pEXPA:NIN* or *pEXPA:GUS* (control) in *nin*. Green GFP-ER fluorescence marked the perinuclear ER network and the tip of growing root hairs in

control and NIN-complemented samples (Fig. 4f, g). However, thick cytoplasmic bridges marked by GFP-ER were only detected in NIN-complemented root hairs (Fig. 4g). No GFP-ER signal was observed in the root cortex of either control or NIN-complemented roots (Fig. 4h, i). Moreover, the constitutive DsRed marker was also unable to detect cytoplasmic remodelling in *nin* (Fig. 4h, i), whereas this marker allowed visualisation of remodelling in control *sunn* roots (as illustrated in Supplementary Fig. 11). This suggests that a functionally active NIN in the cortex is required to trigger cytoplasmic remodelling.

Together, these data infer that NIN acts downstream of DMI3 to control *MtAnn1* gene transcription (possibly directly) and pre-infection priming.

**Fig. 4 | NIN is required for pre-infection priming and *MtAnn1* expression.** *pMtAnn1:GUS* activity in roots of *M. truncatula* A17 (**a**, **b**) or *nin* (**c**, **d**) inoculated or not with *S. meliloti* at 4 dpi. GUS activity (blue) is visualised in endodermis (asterisks), root tips or lateral roots (arrowheads) and rhizobial infection sites (arrows). *LacZ*-expressing rhizobia are in magenta (Close-up in **b**). **c** Infection-induced *pMtAnn1:GUS* activity is abolished in *nin*. **d** NIN under *pEXPA* promoter induces *pMtAnn1:GUS* activity in root epidermis without rhizobia, also shown in Basic Fuchsin counterstained 10 μm sections. Data are from two independent experiments (A17, *n* = 41; *nin*, *n* = 31; *nin* + *pEXPA:NIN* = 37). **e** Transactivation assay of *pMtAnn1:GUS* with NIN or NINΔ (DNA-binding domain deletion version) in *N. benthamiana* leaves. Box plots show distribution of values (open circles) of individual plants (*n* = 25 per sample) from 4 independent experiments. First and third quartile (horizontal box edges), minimum and maximum (outer whiskers), median (centerline), mean (solid black circle) and outliers (crosses) are indicated. Different letters above boxes indicate statistically significant differences (*p* < 2e-16, by One-way ANOVA *α* = 5% followed by Tukey honest significant difference tests). Representative leaf disc images are shown. Expression of *pENOD11:GFP-ER* and *pUBQ10:DsRed* fusions in *nin* control (+*pEXPA:GUS*) (**f**, **h**) and complemented (+*pEXPA:NIN*) roots (**g**, **i**). Red DsRed fluorescence labels cytosol and nucleoplasm. Green GFP-ER fluorescence labels ER network and growing root hair tips (**f**, **g**), and rhizobia-induced cytoplasmic columns in complemented roots (arrowhead in **g**). Arrow indicates enclosed rhizobia (RHE). Cortical cells of control or epidermally-complemented *nin* do not show GFP-ER or DsRed-labelled cytoplasmic rearrangements (**h**, **i**), whereas these are clearly visible with DsRed in *sunn* (see Supplementary Fig. 11). Data are from 2 independent experiments (*n* = 7 for *nin* + *pEXPA:GUS* and *n* = 19 for *nin* + *pEXPA:NIN*). Asterisks mark xy positions of infected root hair base. Scale bars: **a**–**d** = 100 μm, **f**–**i** = 40 μm. See also Supplementary Figs. 10, 11. Source data, including split channels for merged fluorescence, are provided as a Source Data file.

## *MtAnn1* mutation affects efficiency of rhizobial infection and nodule differentiation

Since *MtAnn1* expression is closely linked to early stages of rhizobial entry, we investigated the impact of its mutation on rhizobial infection by analysing *M. truncatula Tnt1* R108 insertion mutant lines (Supplementary Fig. 12). Mutant lines showed reduced (*ann1-1*) or nearly abolished (*ann1-2* and *ann1-3*) *MtAnn1* but not *MtAnn2* expression compared to R108 and R108S, a wild-type sibling from the same mutant population (Supplementary Fig. 12a–c). Rhizobia infection was assessed in *ann1-2* and *ann1-3*, where *MtAnn1* expression was most reduced, by monitoring bacterial β-galactosidase reporter activity in root and nodule tissues. ITs formed in root hairs and progressed into the cortex in mutant roots, but with lower efficiency (Fig. 5a, b). This did not affect nodule numbers, but their infection levels and relative *VPY* expression (Fig. 5c–n and Supplementary Fig. 12d, e). At 4 dpi, there was a consistent decrease in nodule primordia (NP) colonisation in mutants (50–55% of infected NP (+) in mutants, compared to 70–75% in wild-type lines) (Fig. 5c). These NPs were often infected by multiple ITs compared with controls (Fig. 5d–h), suggesting an inefficient process. At 10 dpi, mutant roots showed a mixture of well and poorly colonised nodules, not seen in wild-type R108/R108S (Fig. 5i–l). At this stage, nodule numbers were unaffected in mutant roots, but infection levels were significantly reduced (Fig. 5m, n). RNAi knockdown of *MtAnn1* in *M. truncatula* A17 resulted in a significant reduction in both NP/nodule number and infection at 5 dpi (Fig. 5o, p), consistent with the mutant infection phenotype. The further reduction in nodule number, not clearly seen in *ann1-3* mutant, may be due to the double *MtAnn1* and *MtAnn2* knockdown in RNAi roots (Supplementary Fig. 12f, g). Infected nodules/NP in RNAi roots show the same multiple ITs above as in *ann1* mutants (Fig. 5q, r), as well as abnormal NPs with poor or no colonisation, with several associated root hair ITs (arrow, in Fig. 5r), not seen in control NPs (Fig. 5q, arrowhead). Overall, impaired *MtAnn1* function in mutant and RNAi contexts affected rhizobial root and nodule infection, in R108 and A17 *M. truncatula* backgrounds.

While *ann1* mutants showed no major changes in root architecture or nodule number, striking variations in shape (less elongated overall) and colour (paler pink) were seen at 14–16 and 21 dpi (Fig. 6a, b and Supplementary Fig. 13), reminiscent of potential differentiation and/or nitrogen fixation defects[55]. To better understand why *ann1* nodules were smaller and less elongated, we analysed thin and ultrathin sections of wild-type and mutant nodules. As the two mutant alleles led to similar nodulation defects and the wild-type sibling (R108S) behaved reliably as R108 (see Fig. 5 and Supplementary Figs. 12, 13), we chose to perform sensitive thin/ultrathin sections on one representative of each genotype (R108S and *ann1-3*) (Fig. 6c–f). Meristematic (ZI), infection (ZII), amyloplast-rich interzone (IZ) and nitrogen fixation (ZIII) zones were clearly distinguished in R108S nodules (Fig. 6c, e and Supplementary Fig. 14). However, *ann1-3* nodules showed variable zonation patterns, either wild-type-like or with significantly abnormally large IZ regions, sometimes combined with senescent cells (arrows) (Fig. 6d, f and Supplementary Fig. 14). This was accompanied by a significant difference in the IZ area and nitrogen-fixing capacity of mutant nodules compared to wild-type R108S nodules (Fig. 6g, h), which ultimately led to reduced plant growth (Supplementary Fig. 13). Although *ann1-3* nodules formed bacteroids, they did not appear to complete their differentiation in Zone III-like bacteroids and instead remained interzone-type bacteroids[56], imbedded in a white vesicle-enriched region (black arrows) (Fig. 6j, l).

## MtAnn1-dependent modulation of Ca²⁺ spiking and ultrastructural organisation of the cytoplasmic bridge

As $Ca^{2+}$ spiking and MtAnn1 dynamics were closely linked in the perinuclear regions of primed cells (Figs. 2 and 3), we assessed if *MtAnn1* mutation impacts symbiotic $Ca^{2+}$ spiking. Although more challenging than previous live analysis in the hyper infectious *sunn* context (Fig. 2), we succeeded in monitoring $Ca^{2+}$ spiking responses at infection sites in both wild-type R108S and *ann1-3* transgenic roots expressing the NR-GECO1 $Ca^{2+}$ sensor. Similarly, as seen in *sunn* (Fig. 2), a drop in $Ca^{2+}$ spiking amplitude was observed in R108S in IT-containing RHs compared with RHE root hairs (Fig. 7a). Conversely, this drop was not clearly observed in *ann1-3* (Fig. 7b). Likewise, a global deregulated $Ca^{2+}$ spiking amplitude pattern was observed in many S. *meliloti*-responsive root hairs of *ann1-3* compared with wild-type R108S, while $Ca^{2+}$ spiking frequency was not significantly changed in any root hair categories (Fig. 7c and Supplementary Fig. 15). Together, these data indicate a global deregulation of $Ca^{2+}$ spiking amplitude in *ann1-3*, suggestive of the need of *MtAnn1* for modulating symbiotic $Ca^{2+}$ spiking responses.

$Ca^{2+}$ spiking and MtAnn1 dynamics accompany cytoplasmic bridge formation (Figs. 2 and 3), so we wondered how the deregulated $Ca^{2+}$ spiking response in *ann1-3* (Fig. 7b) could impact cytoplasmic bridge formation. *MtAnn1* is expressed during root infection (Fig. 4), but also in the nodule pre-infection and infection zones[57]. Thus, the defective nodule differentiation phenotype of *ann-1-3* could be related to a possible deregulated infection in the nodule. We thus analysed the cytoplasmic composition of infected cells in R108S and *ann1-3* apical nodule zones by TEM (Fig. 7d–g), which can reveal ultrastructural organelle features and distribution that confocal imaging cannot. In wild-type R108S, ITs progressed through a small vacuole-like enriched cytoplasmic bridge containing a dense ER network (Fig. 7d, e, e'), as shown similarly in A17 (Fig. 1). In *ann-1-3*, the small vacuole-like and ER-enriched cytoplasmic environment around the IT was strongly attenuated (Fig. 7f, g, g'). Thus, mutation in *MtAnn1* resulted in a deregulated $Ca^{2+}$ spiking response and a change in the ER and vesicle composition of the cytoplasmic bridge.

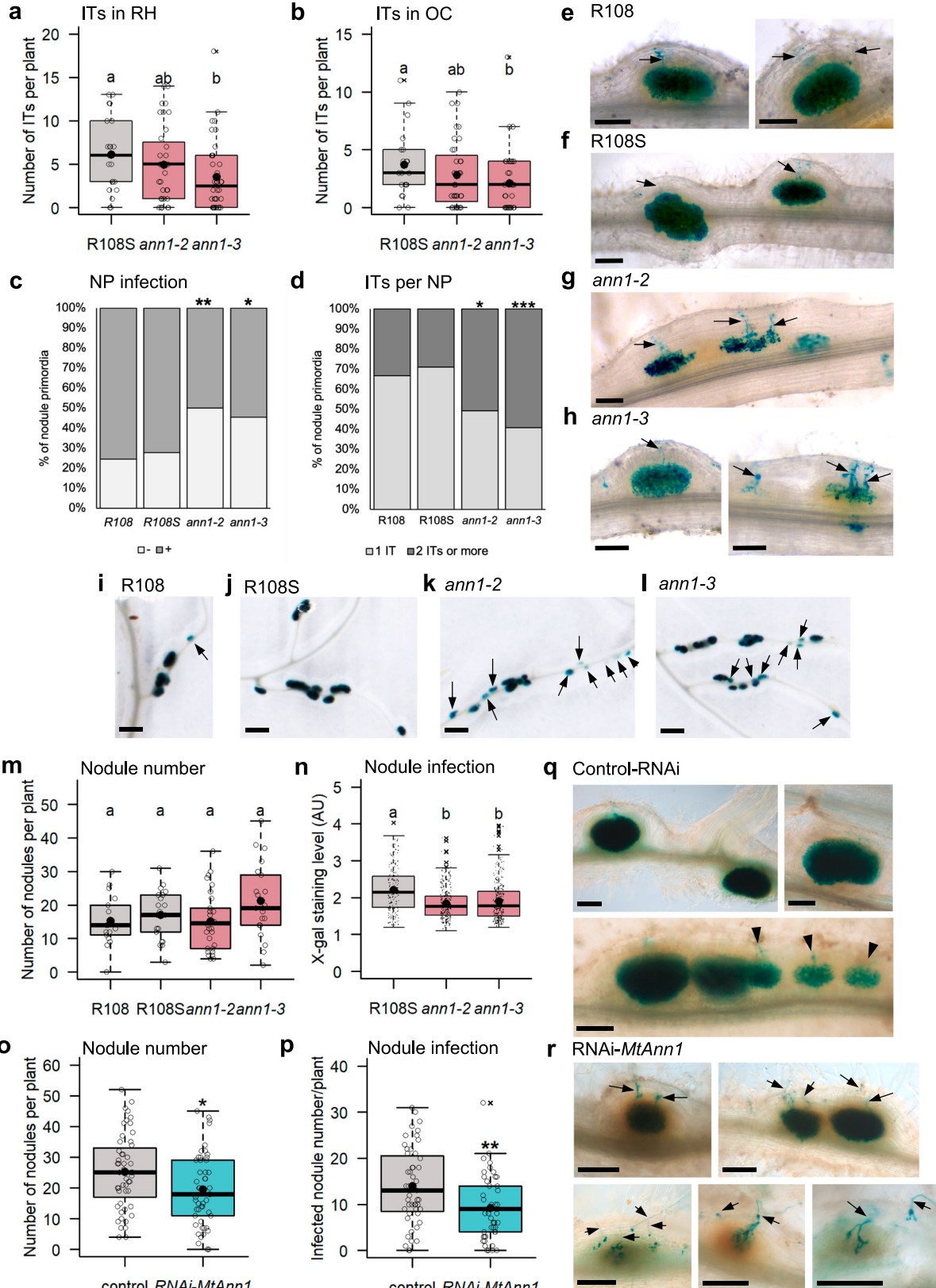

## MtAnn1 is required for efficient mycorrhizal root colonisation

Recently developed genomic and transcriptomic databases from around 300 plant species establishing different types of endosymbioses enabled to address key endosymbiosis-related ancestral gene functions[58]. We have thus exploited these extensive resources to trace the evolutionary history of *MtAnn1*. Previous studies of the

monophyletic plant annexin group, mainly composed of angiosperm sequences, suggested their emergence through duplications from a common ancestor prior to the monocot/eudicot split[59]. Consistent with these data, we found that the *MtAnn1* annexin cluster likely arose from sequential pre-Angiosperms, Eudicots and Papilionoideae-specific duplications (Supplementary Fig. 16 and Supplementary

**Fig. 5 | Root infection is negatively impacted in *ann1* mutant and RNAi transgenic roots.** Infection of *M. truncatula* R108 mutant (**a**–**n**) and A17 RNAi roots (**o**–**r**) for *MtAnn1* with *lacZ*-expressing (blue) *S. meliloti*. At 4 dpi, quantification of the number of ITs in root hairs (**a**) and reaching outer cortex (**b**), proportion of non-infected (-) vs. infected (+) nodule primordia (NP) (**c**) and NPs with 1 or more ITs (**d**), as depicted in (**e**–**h**) (R108 *n* = 17, R108S *n* = 21, *ann1-2 n* = 36 and *ann1-3 n* = 38). At 10 dpi (**i**–**l**), quantification of nodule number/plant (R108 *n* = 18, R108S *n* = 21, *ann1-2 n* = 30, *ann1-3 n* = 21) (**m**) and infection level (X-gal staining intensity) (**n**) (R108S *n* = 196, *ann1-2 n* = 295, *ann1-3 n* = 310). Arrows indicate ITs (**e**–**h**) and poorly-infected nodules (**i**–**l**). Quantification of number of nodules/NP (**o**) and infection level (**p**) in *pMtAnn1:GUS* control (*n* = 51) and *pMtAnn1:RNAi-MtAnn1* (*n* = 50) roots at 5 dpi (**q**, **r**). Control (arrowheads) and RNAi ITs (arrows) are indicated. Box plots

(**a**, **b**, **m**–**p**) show the distribution of values (dots or circles) from 2 (**a**–**d**, **m**, **n**) or 3 (**o**, **p**) independent experiments. First and third quartiles (horizontal box edges), minimum and maximum (outer whiskers), median (centerline), mean (solid black circle) and outliers (crosses) are shown. Classes with the same letter (**a**, **b**, **m**, **n**) are not significantly different (*p* = 0.0388 in **a**, *p* = 0.0449 in **b**, *p* = 9,17e-13 in **n**, Kruskal-Wallis *α* = 5%; *p* = 0,076 in **m**, one-way ANOVA). Asterisks (**c**, **d**, **o**, **p**) indicate statistical difference relative to R108S (in **c**, *p* = 0.8408 for R108, *p* = 0,0027 for *ann1-2*, *p* = 0,0198 for *ann1-3*; in **d**, *p* = 0.7947 for R108, *p* = 0.0226 for *ann1-2*, *p* = 0.001 for *ann1-3*, two-tailed Fisher's exact tests) or control (*p* = 0.0176, two-tailed Student t-test in **o**; *p* = 0.0023, two-tailed Mann-Whitney test in **p**). Scale bars: **e**–**h**, **i**–**l** = 100 μm, **q**, **r** = 1 mm. See Supplementary Figs. 12, 13. Source data are provided as a Source Data file.

Data 1). *MtAnn1* belongs to one of the Papilionoideae duplicated clades, which mainly includes species that establish intracellular endosymbioses, except for *Caryophyllales* and *Cephalotus* (status uncertain). *Ann1* orthologs were also detected in Papilionoideae species that lost RNS but maintained AMS. This indicates that the function of *Ann1* is not restricted to RNS but could expand to AMS. Strikingly, *Ann1* from different species show consistent transcriptional activation during both root nodule and AM symbioses[4], suggesting an ancestral role in root endosymbiotic infection.

To determine if *MtAnn1* indeed plays a role during AMS, we first investigated if *MtAnn1* promoter activity was spatio-temporally regulated in *M. truncatula* roots colonised by the AM fungus *Rhizophagus irregularis*. Strong activation of the *pMtAnn1:GUS* fusion was only observed in root segments colonised by *R. irregularis* at 4 wpi, especially in cortical cells containing arbuscules (Fig. 8a). *R. irregularis* colonisation rates were then monitored in *ann1-3* compared to R108S wild-type roots at 6 wpi. While R108S roots had dense colonisation by *R. irregularis*, with cortex cells fully filled with arbuscules, mutant roots had a lower proportion of colonised root sectors and arbuscule density (Fig. 8b–f). In addition, arbuscules formed in the mutant showed a slight but significant reduction in length and this was associated with a trend increase in the expression of the arbuscule senescence-related marker *MYB1*[60] (Fig. 8c, f–g). These data highlight the importance of *MtAnn1* for the efficient establishment of the ancient AM symbiosis.

## Discussion

Rhizobia root infection occurs transcellularly in most legumes via apoplastic IT structures that grow polarly to reach the developing nodule. Plant cells engaged in this process form cytoplasmic bridges that precede and accompany the route for ITs to progress. This process, analogous to PPA bridges formed during AM symbiosis, is likely governed by common ancestral cellular mechanisms, though knowledge remains limited. Here we report cell-specific Ca²⁺ responses and cytoplasmic ultrastructural rearrangements that prime *M. truncatula* plant cells for rhizobia infection. We show that this priming state, under DMI3-NIN genetic control, is marked by an intimate cellular link between Ca²⁺ spiking and annexin MtAnn1 accumulation. Moreover, MtAnn1 is needed for modulation of Ca²⁺ spiking responses and creation of an infection-permissive, dense, ER- and small vacuole-rich cytoplasmic environment. Ultimately, *MtAnn1* is required for successful root nodule differentiation and efficient AM root colonisation, implying an ancient recruitment of *MtAnn1* in endosymbiosis. Our work supports the need to fine-tune cell-specific Ca²⁺-spiking responses by annexin Ca²⁺ binding proteins for the transcellular passage of endosymbionts.

### Organelle dynamics shaping rhizobia pre-infection priming

Ultrastructure dissection of plant cells preparing for transcellular rhizobia infection revealed marked enrichment and distribution of organelles during this process (Fig. 1). This priming state is seen only in a few outer cortical cells in close contact with the infected root hair cell (Figs. 1, 3, Supplementary Figs. 6 and 8), highlighting the likely need for

signals from the infected cell to promote it. A marked clustering of mitochondria around the nucleus is observed in primed cells (Fig. 1 and Supplementary Fig. 1). This may reflect the energy-demanding nature of this process. However, dividing nodule primordia cells with presumably high metabolic activity show no such enrichment. Alternatively, perinuclear mitochondria may mark a specific regulated cell cycle status, likely to occur in these cells[32,34,61] or contribute to modulating cytosolic Ca²⁺ oscillations, as shown in other systems[62].

Consistent with previous findings[35], TEM analysis revealed that a dense ER-rich network occupy the transcellular cytoplasmic bridge in primed or rhizobia infected *M. truncatula* cells (Figs. 1 and 7). The bridge is also rich in small vacuole-like structures[63], but we cannot exclude that multivesicular bodies (MVBs), seen in the PPA bridge, are part of this organelle population. Future use of molecular markers[63] may help discriminate between them. Dynamic changes in plant vacuole size have been reported to occur in different cell types, and in early meristematic cells this is thought to increase cytosolic area for subsequent daughter cell formation events[63,64]. An analogous need for increased cytosolic area is possibly required for IT passage. The concomitant reduction of small vacuoles and ER (involved in vacuole biogenesis[65]) in *MtAnn1* mutants (Fig. 7) raises the question of their functional interconnection in this process.

### Role of MtAnn1 in modulating cell- and stage-specific symbiotic Ca²⁺ responses

High frequency Ca²⁺ spiking, triggered by Nod factors in root hairs is key for early rhizobia recognition[5,6,46]. Here, we showed that a second Ca²⁺ spiking phase occurs in root hairs undergoing rhizobia infection and involves a decrease in Ca²⁺ peak amplitude as ITs extend into root hairs (Fig. 2). This drop was also reported in root outer cortical cells crossed by an IT[43], consistent with this switch being key for IT progression. Ca²⁺ amplitude regulation fine-tunes different biological processes ranging from gene expression, stomatal closure, gravitropic responses to synaptic plasticity in animals[66–68]. The deregulated Ca²⁺ amplitude profile and defective infection caused by *MtAnn1* mutation (Figs. 5 and 7) points to a functional link between MtAnn1 and regulation of Ca²⁺ spiking amplitude during rhizobia infection.

We determined that Ca²⁺ spiking and strong MtAnn1-GFP labelling are hallmarks of rhizobia infection and pre-infection priming (Fig. 3 and Supplementary Fig. 8) and rely on the cell-autonomous activity of DMI3 and NIN (Fig. 4 and Supplementary Fig. 9). Thus, spatially distinct Ca²⁺ spiking responses are first triggered in non-infected root hairs independently of DMI3-NIN[69,70], before DMI3-NIN-dependent Ca²⁺ spiking is induced in cells engaged for infection. Using the NR-GECO1 sensor, we observe a unique low frequency Ca²⁺ spiking only in primed cells of the cortex in both A17 and *sunn* backgrounds (Fig. 3 and Supplementary Fig. 8), in agreement with previous findings using a FRET-cameleon sensor[43]. Although root hairs show similar remodelling and MtAnn1 labelling, they never show such low frequency Ca²⁺ spiking, a signature likely reflecting a commitment state to infection.

In *Medicago*, symbiotic Ca²⁺ spiking is generated in the nucleoplasm and perinuclear cytoplasm by the concerted action of nuclear

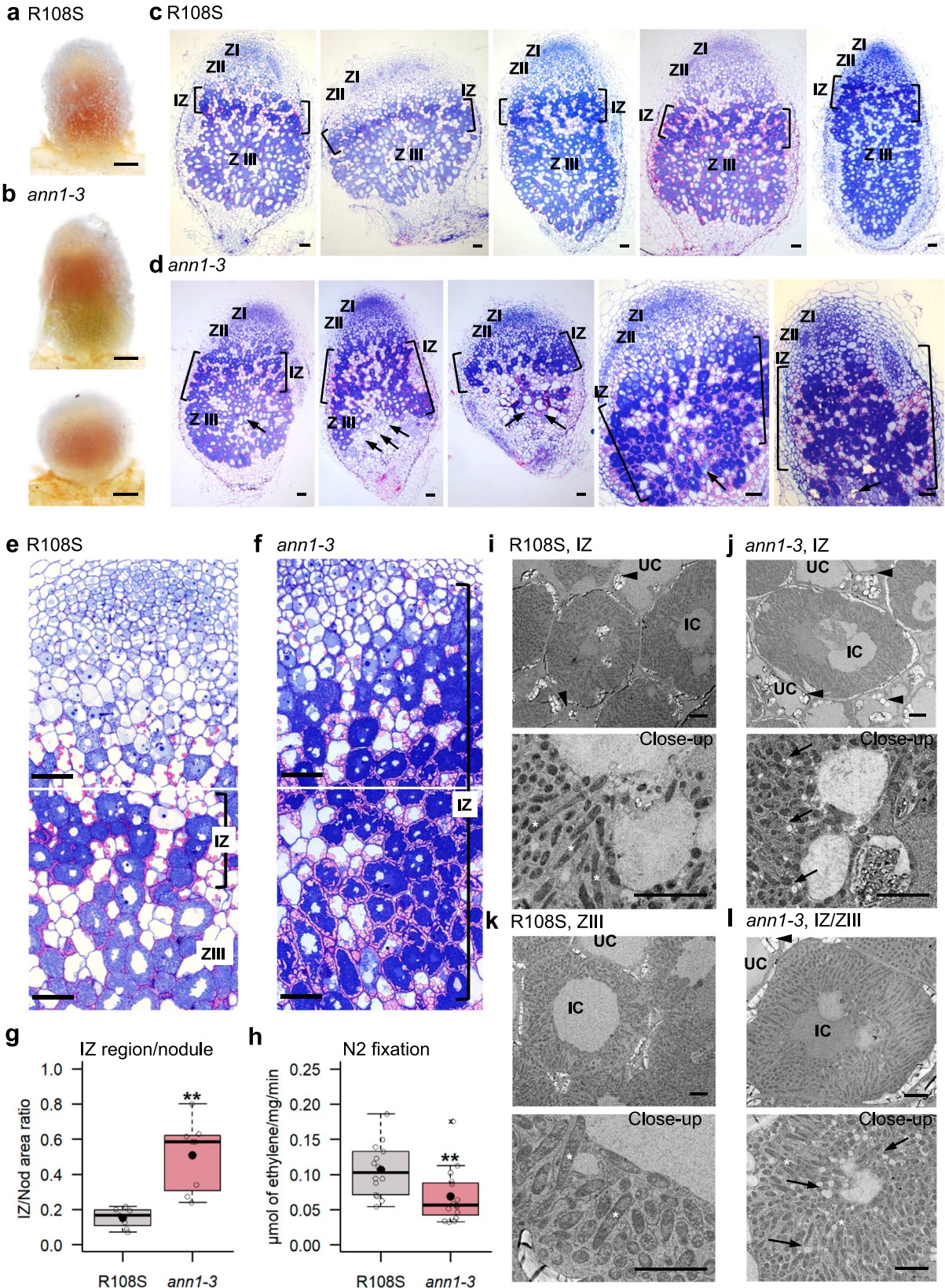

envelope ion channels (DMI1, CNGC15), a Ca²⁺ pump (MCA8) and under negative feedback regulation by a nuclear calmodulin (CaM2)[6,71]. MtAnn1 is a cytosolic phospholipid- and Ca²⁺-interacting protein of the annexin family[13], viewed as dynamic regulators that associate with membranes when intracellular Ca²⁺ levels change[15,16]. Annexin family members are involved in the regulation of ion transport, modulation of

cytoplasmic Ca²⁺ signals and ROS signalling in diverse abiotic or biotic responses[72–76]. Although predominantly cytoplasmic, annexins can also topologically associate with the ER[77,78] and modulate the activity of Ca²⁺-release channels[79]. As MtAnn1 partially localises to the ER network in cytoplasmic bridges (Supplementary Figs. 5, 6), it may exert such a regulatory function. The radical change in the cytosolic distribution of

**Fig. 6 | *MtAnn1* mutants show impaired nodule differentiation and function.**
Representative images (**a**, **b**) and 1 μm longitudinal sections stained with Toluidine Blue and Basic Fuchsin (**c**–**f**) of wild-type R108S and *ann1-3* mutant nodules 14–16 dpi. Meristematic (ZI), infection (ZII), interzone (IZ) and nitrogen fixing (ZIII) zones are labelled. The amyloplast-rich IZ is indicated between brackets (**c**–**f**). Arrows (**d**) indicate senescing cells. **g** Nodule IZ area was measured in 1 μm nodule section images (see Supplementary Fig. 14) of 14–16 dpi R108S (*n* = 7) and *ann1-3* (*n* = 8) nodules from 2 independent experiments. **h** Nitrogen fixation capacity was measured in R108S (*n* = 14 roots with 365 nodules) and *ann1-3* (*n* = 15 roots with 417 nodules) 21 dpi nodulated root segments from 3 independent experiments. Box plots in **g**, **h** represent the distribution of individual values (open circles). First and third quartiles (horizontal box edges), minimum and maximum (outer whiskers), median (centerline), mean (solid black circle) and outliers (crosses) are shown.

Asterisks in **g**, **h** indicate significant differences in *ann1-3* mutant vs R108S control ($p = 0.0012$, two-tailed Welsh t-test in **g**, $p = 0.0049$, two-tailed Student t-test in **h**). **i**–**l** TEM ultrastructural analyses of IZ or ZIII zones in R108S and *ann1-3* 14–16 dpi nodules (*n* = 5 for R108S and *n* = 4 for *ann1-3* from 2 independent experiments) and close-up views of IZ and ZIII or IZ/ZIII of R108S or *ann1-3* nodules with uninvaded cells (UC) and bacteroid-filled invaded cells (IC). Black arrowheads indicate amyloplasts located near intercellular spaces of IC and within UC cells in the IZ of R108S and *ann1-3*, and in the IZ/ZIII of *ann1-3*. Differentiated bacteroids (white asterisks) are seen in both R108S and *ann1-3*, but in *ann1-3* nodules they resemble interzone-type bacteroids[55] imbedded in white vesicle-enriched regions (black arrows). Scale bars: **a**, **b** = 1 mm, **c**–**f** = 50 μm, **i**–**l** = 5 μm. See also Supplementary Figs. 12-14 for complementary molecular and phenotypic description of *MtAnn1* mutants. Source data are provided as a Source Data file.

the ER and small vacuoles by the *MtAnn1* mutation (Fig. 7) also establishes a functional link between MtAnn1 and the creation of this specific cytoplasmic environment. $Ca^{2+}$ orchestrates membrane trafficking and fusion, while vacuoles, emitters and receivers of $Ca^{2+}$ signals, can be directly affected by altered cytosolic $Ca^{2+}$ levels[80,81]. Annexins are $Ca^{2+}$ sensing proteins involved in endomembrane trafficking in plants[82], either in conjunction with the membrane fusion machinery or by regulating exocytosis. Thus, MtAnn1 may directly or indirectly impact $Ca^{2+}$-regulated vesicle trafficking and/or vacuole morphology. However, the biological relevance of such a dense cytoplasmic strand for IT passage remains to be determined.

### Evolutionary recruitment of *MtAnn1* for root endosymbioses
In *Mtann1* mutants, root infection by rhizobia is reduced (Fig. 5) and this culminates in mature nodules with reduced size, altered shape, abnormal zonation and reduced nitrogen fixation ability. These phenotypes are reminiscent of the reduced infection levels and nodule size observed after *PvAnn1* RNAi knock-down in common bean[83]. Such phenotypes probably stem from the early deregulated $Ca^{2+}$ spiking, impaired cytoplasmic configuration and consequently defective infection caused by the *MtAnn1* mutation. Furthermore, as *MtAnn1* is expressed in nodule pre-infection and infection zones[57], the defective nodule differentiation phenotype is likely a consequence of maintained deregulated infection in the nodule. Finally, the close paralogue *MtAnn2*[14] may act redundantly with *MtAnn1* in early root infection (Fig. 5) where they are both co-expressed[48], but it is unlikely to compensate for *MtAnn1* loss of function in nodules as it has a distinct spatial expression profile.

We showed that the subclade containing *MtAnn1* emerged mainly in plant species establishing intracellular symbioses. Induced expression of *MtAnn1* in rhizobia and AM symbioses and its requirement for proper formation of nitrogen-fixing nodules and cortical arbuscules (Figs. 5 and 8) suggest ancestral and shared roles for *MtAnn1* in endosymbiotic infection. We propose that *MtAnn1* fine-tunes oscillatory $Ca^{2+}$ responses to create an optimal dense cytoplasmic environment for rhizobia IT and AM fungi hyphae to progress. As *MtAnn1* is transcriptionally regulated by NIN, even in the absence of symbiotic signalling (Fig. 4), we hypothesise that *MtAnn1* is a direct target of NIN and that its evolutionary recruitment for RNS involved the acquisition of putative NIN *cis*-regulatory motifs, which are indeed present in the *MtAnn1* promoter. Since NIN is a major regulator in RNS[84], the AM-associated expression of *MtAnn1* is possibly under the control of another, so far unidentified regulator.

In conclusion, we showed that cell specific $Ca^{2+}$ spiking signals shape rhizobia pre-infection priming to prepare the optimal dense cytoplasmic environment for IT passage in *Medicago*, and MtAnn1 emerges as a modulator of this process. MtAnn1, necessary for efficient rhizobia and AM colonisation, may have been recruited during evolution to integrate an ancestral $Ca^{2+}$-regulatory module for guiding endosymbiotic infection. MtAnn1, with a conserved K/H/RGD protein-

interacting motif[15], may work in partnership with other players and discovering them could help identify other critical components in this process.

## Methods
### Plant materials and microbial strains
*M. truncatula* Jemalong A17 and R108 wild-type ecotypes, a regenerated transgenic line (referred to as A2) expressing the *p35S:GFP-ER* construct[35] as well as *sunn* (*sunn-2* allele)[85], *dmi3* (*dmi3-1* allele)[8], *nin* (*nin-1* allele)[70] and *ern1* (*bit1* allele)[49,51] mutants were used in this work. The *dmi3* line carrying the *pEXT:DMI3* construct[52], hereafter called *dmi3 + pEXT:DMI3*, was kindly provided by C. Remblière (LIPME, Toulouse). The three R108 *Tnt1* mutants carrying insertions in *MtAnn1* were obtained from Oklahoma State University and are designated *ann1-1* (NF0830), *ann1-2* (NF17737) and *ann1-3* (NF4963). These lines were self-crossed and genotyped by PCR and sequencing at each generation using primers −344 *pMtAnn1-Fw*, *Mtr14183-Rev*, −914 *pMtAnn1-Fw*, *LTR6-* and *LTR4-* (Supplementary Table 1). Homozygous mutants and sibling wild-type lines (referred to as R108S) were selected for seed multiplication and phenotyping. R108S is a wild-type sibling derived from line NF4963. Seeds were scarified and surface-sterilised prior to germination on inverted soft agar plates[45] and used for *A. rhizogenes* transformation or nodulation experiments. *N. benthamiana* seeds were germinated in potting soil in glass-covered trays before transfer to pots (7 cm × 7 cm × 6.5 cm) at 21 °C in a growth chamber under a 16 h photoperiod and 70 μE/m/s light intensity for 3 weeks until bacterial infiltration. The strains *A. rhizogenes* ARquA1, *A. tumefaciens* GV3103 & GV3101 and *S. meliloti* 2011 (*Sm2011-lacZ*), constitutively expressing a *hemA-lacZ* fusion, have been previously described[45]. The *S. meliloti* 2011 strain *Sm2011-cCFP*, constitutively expressing a cyan fluorescent protein was kindly provided by P. Smit, Wageningen (The Netherlands). *R. irregularis* spores (DAOM197198) were purchased from Agronutrition (Labège, France).

### DNA constructs
*pMtAnn1:MtAnn1-GFP*[13] in pLP100, *pENOD11:GFP-ER*[35] and *p35S:GFP-ER*[86] in pBIN121 and *p2x35S:mCherry-ER* in pBIN20[87] were used to study live pre-infection priming responses in *M. truncatula*. *p2x35S:NR-GECO1* in pCAMBIA-CR1ΔDsRed[46] was used to monitor $Ca^{2+}$ spiking in *M. truncatula*. *pMtAnn1:GUS* in pLP100[13], *pEXPA:NIN* in pK7WG2-R, *p35S:3xHA-NIN* and *p35S:3xHA-NINΔ* in PAMPAT-GTW[54], were used in transcription activation studies in *M. truncatula* or *N. benthamiana*. *p35S:MtAnn1-GFP* and *p35S:GFP* in pBIN121[13], were used in mutant complementation studies. The *pMtAnn1:RNAi-MtAnn1* and *pMtAnn1:-GUS* (control) constructs were generated here by Golden Gate cloning into the pCAMBIA-CR1 vector[46]. Briefly, DNA fragments corresponding to 2231 bp of the genomic *MtAnn1* sequence upstream of the ATG, 942 bp of the *MtAnn1* coding sequence in sense and antisense orientations, and the GUS coding sequence, were generated by PCR amplification using the primer pairs *pMtAnn1-GG-A-Fw/pMtAnn1-GG-B-*

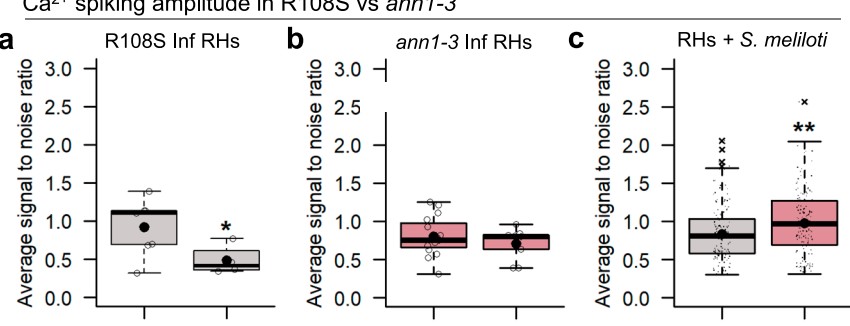

Ca²⁺ spiking amplitude in R108S vs *ann1-3*

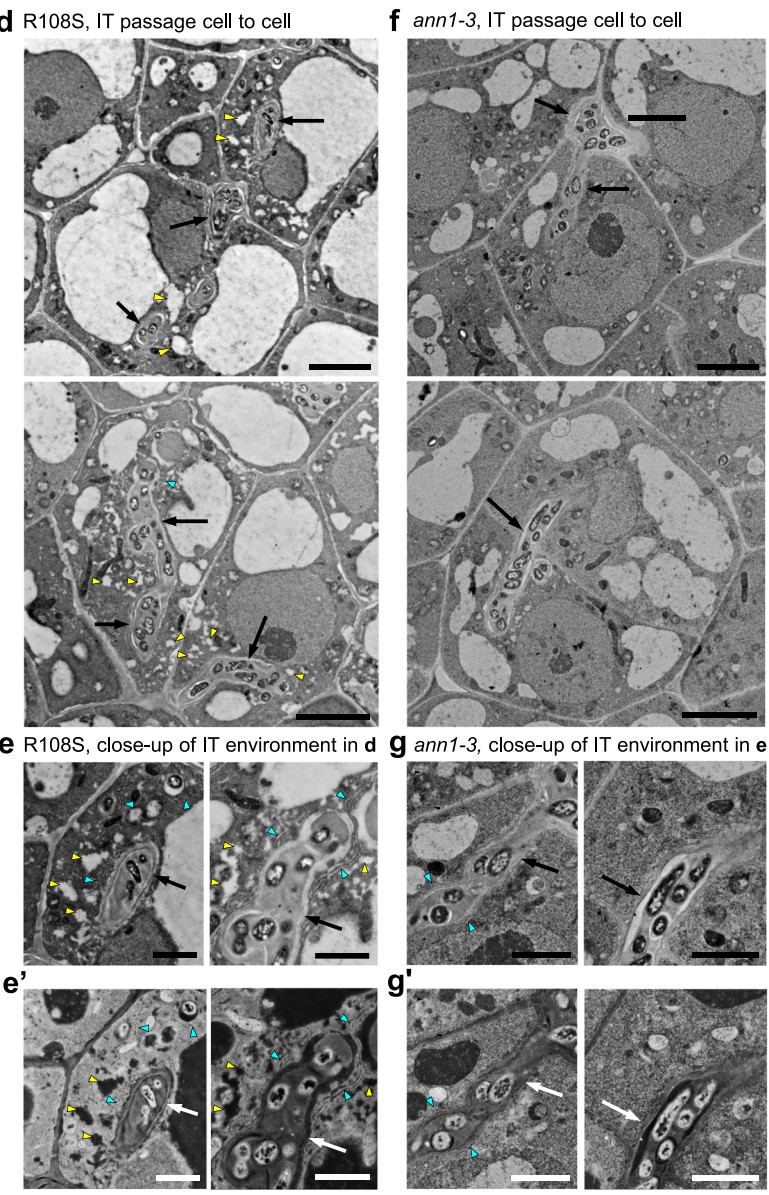

*Rev, MtAnn1-ATG-GG-B-Fw/MtAnn1-STOP-GG-C-Rev, MtAnn1 antisens-STOP-GG-X-Fw/MtAnn1 antisens-ATG-GG-D-Rev* and *GUS-GG-B-Fw/GUS-GG-D-Rev*, respectively (Supplementary Table 1). DNA fragments were then cloned into a pBluescript SK vector and validated by DNA sequencing prior to Golden Gate assembly. Golden Gate reactions with plasmids comprising (i) the *MtAnn1* promoter, *MtAnn1* coding

sequences (sense and antisense) and a 1300 bp intron spacer[24] or (ii) the *MtAnn1* promoter and the *GUS* coding sequence, were used to generate RNAi *MtAnn1* (*pMtAnn1:MtAnn1sens-intron-MtAnn1antisens*) or GUS control (*pMtAnn1:GUS*) constructs in the binary pCAMBIA-CR1 vector. All binary vectors used for *A. rhizogenes* transformation of *M. truncatula* (pLP100, pCAMBIA-CR1, pK7WG2-R, pBIN121) include a

**Fig. 7 | Modified Ca²⁺ spiking and IT cytoplasmic environment in the absence of *MtAnn1*. a–c** Transgenic roots expressing the NR-GECO1 Ca²⁺ sensor were generated to monitor nuclear Ca²⁺ oscillations in *ann1-3* mutant or wild-type (R108S) plants. Variations in Ca²⁺ ion concentration, reflected by changes in the relative intensity of NR-GECO1 fluorescence, are expressed as signal-to-noise ratio (SNR, cf. 'Methods' section). Box plots represent average amplitude of spikes (spike SNR), calculated separately for each nucleus of root hairs with entrapped rhizobia (RHE) or with a growing IT (IT) in (**a**) R108S (RHE, $n = 7$; IT, $n = 4$) or in (**b**) *ann1-3* (RHE, $n = 15$; IT, $n = 9$) or in *S. meliloti*-responsive root hairs in the close competence zone (**c**) (R108S, $n = 160$ and *ann1-3*, $n = 156$) at 1–4 dpi. Box plots show the distribution of values (open circles or black dots) obtained from three independent experiments. First and third quartiles (horizontal box edges), minimum and maximum (outer whiskers), median (centerline), mean (solid black circle) and outliers (crosses) are shown. Asterisks indicate statistical difference in RHE vs. IT in R108S RHs in **a** ($p = 0.0285$, one-tailed Student t-test) and in RHs of R108S vs. *ann1-3* in (**c**) ($p = 0.0018$, two-tailed Mann-Whitney test). Differences were not significant in RHE vs. IT in *ann1-3* RHs in **b** ($p = 0.1735$, one-tailed Student t-test). (**d-g**) TEM analysis of infected cells from apical zone II of R108S (**d**, **e**, **e'**) or *ann1-3* (**f**, **g**, **g'**) were performed in 80 nm sections of 14–16 dpi nodules ($n = 5$ R108S nodules and $n = 4$ *ann1-3* nodules derived from 2 independent experiments). ITs (arrows) in wild-type R108S are embedded in vesicles (yellow arrowheads) and ER-rich (blue arrowheads) cytoplasmic bridges, which are less visible in *ann1-3*. **e'**, **g'** are complementary inverted LUT images of (**e-g**) that were generated by ImageJ. Scale bars: **d**, **f** = 5 μm, **e**, **e'**, **g**, **g'** = 2 μm. See also Supplementary Fig. 15. Source data are provided as a Source Data file.

kanamycin resistance cassette in the T-DNA region. pK7WG2-R and pCAMBIA-CR1 vectors also include a DsRed cassette.

### Generation of *M. truncatula* transgenic material

*A. rhizogenes*-mediated transformation of *M. truncatula* A17 (wild-type, *sunn*, *ern1* and *dmi3*) or R108 (R108S wild-type sibling and *ann1-3*) was performed as described[45]. Briefly, germinated seedlings had their root tips removed with a scalpel and were placed on Fahraeus medium plates (12 cm × 12 cm) supplemented with 0.5 mM NH4NO3 and 20 mg/L kanamycin, before inoculation with a drop (~3 μL) of *A. rhizogenes* aqueous suspension (OD$_{600nm}$ 0.5). For co-transformation experiments with strains *pMtAnn1:MtAnn1-GFP + p2x35S:NR-GECO1*, *p2x35S:NR-GECO1 + pENOD11:GFP-ER*, *pEXPA:NIN + pENOD11:GFP-ER* or *pEXPA:NIN + pMtAnn1:GUS* (in pLP100), equal volumes of bacterial suspensions were mixed prior to seedling inoculation. The lower ¾ part of the plates were sealed with parafilm and placed vertically in plastic boxes under controlled 16 h light/ 8 h dark photoperiod conditions, first at 20 °C for 1 week, then at 25 °C for two to three weeks. Composite *M. truncatula* plants with kanamycin resistant roots were selected. In some experiments, kanamycin-resistant composite roots were re-selected on the basis of constitutive fluorescence of the DsRed marker (for *pEXPA:NIN* in pK7WG2-R and *pMtAnn1:RNAi-MtAnn1* and *pMtAnn1:GUS* in pCAMBIA-CR1 constructs), fusion fluorescence in lateral roots (for *pMtAnn1:MtAnn1-GFP* and *pENOD11:GFP-ER* constructs) or in the nucleus (for *p2x35S:NR-GECO1*). Selected composite plants were then transferred to appropriate nitrogen- and antibiotic-free plates or pots for subsequent inoculation with rhizobia.

### Plant growth and microbial inoculation procedures

For in vivo imaging experiments, kanamycin-resistant and fluorescent-positive composite plants of A17 (wild-type, *sunn*, *ern1* and *dmi3*) or R108 (R108S and *ann1-3*) carrying *pMtAnn1:MtAnn1-GFP + p2x35S:NR-GECO1*, *pEXPA:NIN + pENOD11:GFP-ER* or *p2x35S:NR-GECO1 + pENOD11:GFP-ER* constructs were transferred to nitrogen-free Fahraeus plates supplemented with Amoxycillin sodium/Clavulanate potassium 5:1 (200 mg/L) for 3–7 days, then transferred to nitrogen-free 0.5% [w/v] phytagel Fahraeus plates supplemented with 50 nM 2-amino ethoxyvinyl glycine (AVG) for 3 days, as described previously[27]. Whole root systems were then inoculated with 0.5 to 1 mL of the *Sm2011-cCFP* suspension (OD$_{600nm}$ 0.001), applied between the medium and the LUMOX film (Sarstedt, UK). The inoculated root systems were kept protected from light, by wrapping ¾ of the plates with dark plastic bags in a culture room at 20 °C or 25 °C with a 16-h photoperiod and a light intensity of 70 mE/s/m², until microscopy observations.

For high-resolution microscopy analyses of pre-infection priming in A17 and nodule differentiation in R108S/*ann-3*, germinated seedlings of *M. truncatula* A17 and R108 (R108S and *ann1-3*) were grown for 3 days on nitrogen-free paper/Fahraeus Kalys HP696 agar plates[45] before roots were either spot or flood-inoculated with water (control) or a *Sm2011-lacZ* (OD$_{600nm}$ 0.01) suspension, as described[24]. For expression analyses of *pMtAnn1:GUS* (in pLP100) in A17 and *nin* (co-expressing or not *pEXPA:NIN*), kanamycin-resistant and/or DsRed fluorescence-positive composite plants were transferred 3 weeks after *A. rhizogenes* transformation to the same nitrogen-free paper/Fahraeus Kalys HP696 agar plates and grown for 3 days prior to flood inoculation (for 1 h) of the entire root system with *Sm2011-lacZ* (OD$_{600nm}$ 0.01). Plants grown in vitro on plates were all cultivated at 25 °C under a 16-h photoperiod and a light intensity of 70 mE/s/m² with their root systems protected from light by wrapping ¾ of the plates with dark plastic bags.

For phenotyping experiments, germinated seedlings of *M. truncatula* R108 lines (R108, R108S and *ann1-1 to ann1-3* mutants) and selected composite plants (2–3 weeks after transformation) of A17-transformed with *RNAi-MtAnn1*/control constructs or R108 (R108S or *ann1-3*) transformed with *p35S:GFP/p35S:MtAnn1-GFP* constructs, were transferred to 8 × 8 × 7 cm pots (3 plants/pot for germinated seedlings and 2 plants/pot for composite plants) filled with inert attapulgite substrate (Oil Dri US Special; http://www.oildri.com/), supplemented with 10 mL nitrogen-free Fahraeus medium. For longer growth periods (>3 wpi), plants were grown in 9 × 9 × 8 cm pots (3 plants/pot) supplemented with 14 mL medium. Pots were placed in small greenhouses at 25 °C, with a 16 hours photoperiod and a light intensity of 100 mE/s/m² and inoculated with a suspension of *Sm2011-lacZ* (OD$_{600nm}$ 0.1), after 3 days of nitrogen starvation.

To assess mycorrhizal root colonisation of R108S and *ann1-3* mutant lines, germinated seedlings were transferred to 8 × 8 × 7 cm pots (1 plant per pot), filled with a 1:1 mix of Zeolite substrate fractions 1.0–2.5 mm and 0.5–1.0 mm (Symbiom LTD, Lanskroun, Czech Republic), and inoculated with *R. irregularis* spores (strain DAOM197198, 150 spores per pot). Pots were placed in 60 × 40 × 12 cm trays in a 16-h photoperiod chamber (light intensity: 300 μmol/s/m²) at a day-time temperature of 22 °C and a night-time temperature of 20 °C, and 70% humidity. Plants were watered weekly with a modified low-phosphate and low-nitrogen Long Ashton solution (7.5 μM Na$_2$HPO$_4$, 750 μM KNO$_3$, 400 μM Ca(NO$_3$)$_2$, 200 mg/L MES buffer, pH 6.5).

### β-glucuronidase (GUS) and β-galactosidase enzymatic assays

Root segments from *M. truncatula* composite plants expressing *pMtAnn1:GUS* (in pLP100) or *pEXPA:NIN + pMtAnn1:GUS* constructs were collected from control (non-inoculated) or rhizobia-inoculated roots and incubated in 0.5% paraformaldehyde/0.1 M potassium phosphate buffer pH 7.0, for 1 h, prior to histochemical (blue) staining for GUS activity for 2–5 h at 37 °C using 1 mM of the substrate X-Gluc (5-bromo-4-chloro-3-indoxyl-b-D-GlcA, cyclohexylammonium salt, B7300; Biosynth, Staad, Switzerland) as described[51]. Histochemical GUS staining of mycorrhizal roots expressing *pMtAnn1:GUS* or *N. benthamiana* leaf discs was carried out in the same X-Gluc substrate but supplemented with 0.1% Triton X-100, first under vacuum for 20–25 min at room temperature before incubation at 37 °C for 1-3 hours. Enzymatic GUS fluorimetric assays of *N. benthamiana* leaf discs were done according to ref. 45. Briefly, tissues were ground using

*pMtAnn1:GUS* in *M. truncatula* roots

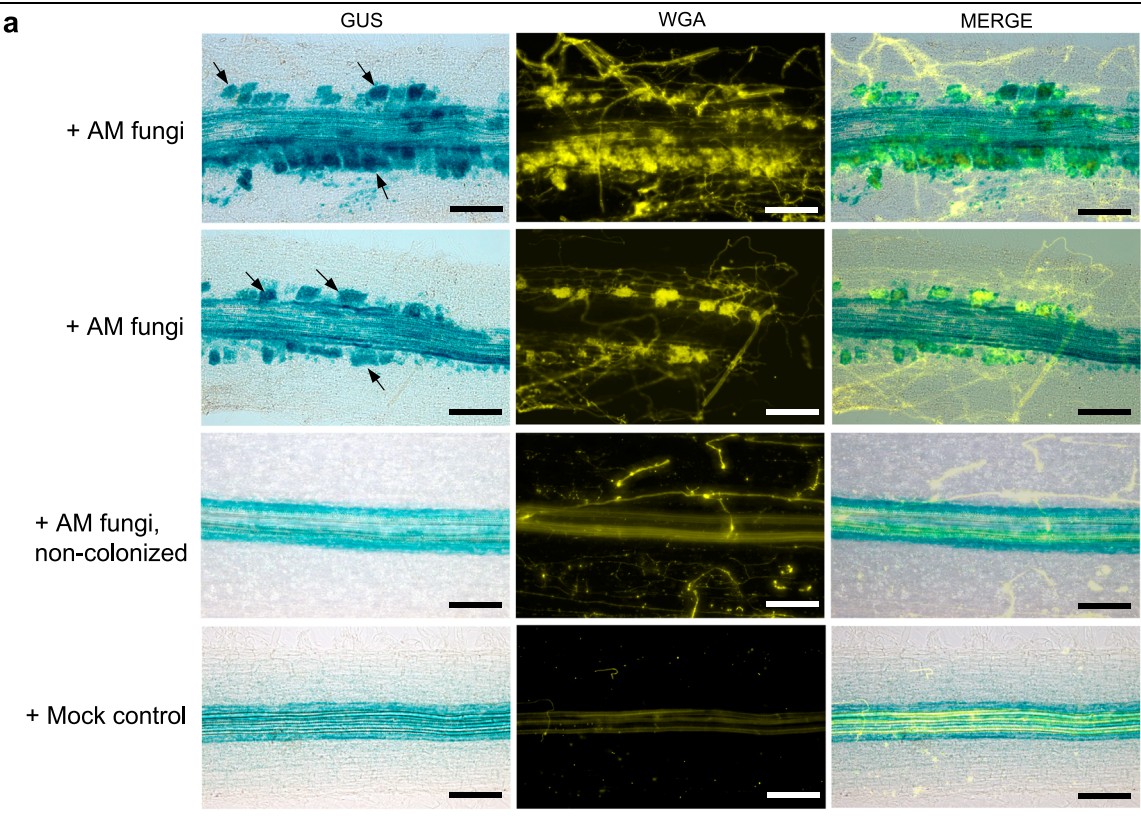

Mycorrhization of R108S vs *ann1-3* roots

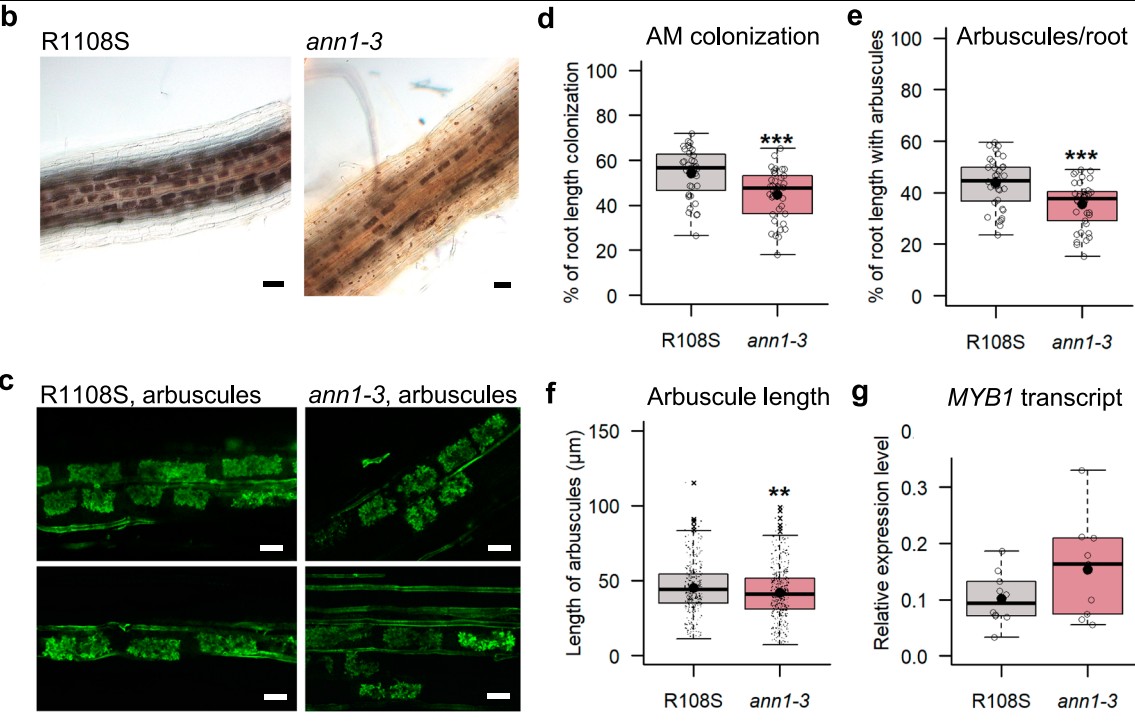

a MM400 grinder (Retsch) and homogenised in GUS extraction buffer before 1 μg of total protein extracts were used for enzymatic reactions at 37 °C using 1 mM of the 4-MUG substrate (4-Methylumbelliferyl-β-D-glucuronide hydrate, Biosynth M-5700). GUS activity was measured using a FLUOstar Omega 96 microplate reader (BMG LABTECH, France) through quantification of the fluorescence of 4-MU (4-

Methylumbelliferone, Sigma) reaction product. To reveal the constitutive β-galactosidase activity of the *S. meliloti* strain *Sm2011-lacZ* in ITs, root samples (sometimes pre-stained for GUS activity) were rinsed and fixed for 1 h in 1.25% glutaraldehyde/Z buffer (10 mM KCl, 1 mM $MgCl_2$ and 0.1 M phosphate buffer, pH 7.0) as described[45]. After rinsing in Z-buffer, root samples were incubated overnight in the dark at 28 °C

**Fig. 8 | MtAnn1 is required for efficient mycorrhization of M. truncatula roots. a** *pMtAnn1:GUS* activity in *M. truncatula* A17 roots 4 wpi with *R. irregularis* (+AM fungi, *n* = 10) or mock control (*n* = 8) was analysed in 2 independent experiments. Representative images of root sectors inoculated with *R. irregularis* (+AM fungi) with or without fungal colonisation are shown. *pMtAnn1:GUS* (GUS) blue staining in root cortex (arrows) correlates with areas colonised by *R. irregularis*, visualised yellow by WGA-Alexa Fluor 488 fluorescence staining (WGA) and merged images (MERGE). **b–g** Phenotypic analysis of R108S or *ann1-3 M. truncatula* roots 6 wpi with *R. irregularis*. Representative images of ink-stained AM-colonised roots (**b**) or WGA-stained arbuscules (**c**) are shown. **d, e** Intraradical colonisation by *R. irregularis* was determined in R108S wild-type (*n* = 36) and *ann1-3* (*n* = 38) using the gridline intersect method. Box plots show the relative surface of the root system with vesicles, arbuscules and/or intraradical hyphae (**d**) and the relative area corresponding specifically to arbuscules (**e**). **f** Arbuscule lengths were compared in R108S (*n* = 371, from 8 plants) and *ann1-3* (*n* = 440, from 7 plants) roots at 6 wpi. **g** Q-RT-PCR expression analysis of *MYB1* in total RNA samples from independent R108S (*n* = 10) or *ann1-3* (*n* = 9) plants inoculated with *R. irregularis* (6 wpi). Box plots (**d–g**) show single values distributions (open circles or black dots) from 2 independent experiments. First and third quartiles (horizontal box edges), minimum and maximum (outer whiskers), median (centerline), mean (solid black circle) and outliers (crosses) are shown. Asterisks indicate statistical difference between R108S and *ann1-3* in (**d–f**) (*p* = 0.0004 in **d**, *p* = 0.0002 in **e**, two-tailed Student t-tests; *p* = 0.0046 in **f**, two-tailed Mann-Whitney test). Although there is a trend towards higher *MYB* expression in *ann1-3* vs. R108S (**g**), the difference is not significant (*p* = 0.1228, two-tailed Student t-test). Scale bars: **a** = 100 μm, **b** = 50 μm, **c** = 20 μm. See also Supplementary Fig. 16. Source data are provided as a Source Data file.

in Z-buffer containing 2 mM Magenta-Gal (5-bromo-6-chloro-3-indoxyl-b-D-galactopyranoside; B7200; Biosynth) or X-gal (5-bromo-4-chloro-3-indolyl-b-D-galactopyranoside, W5376C; Thermo Fisher Scientific, Guilford, CT). GUS and/or LacZ-stained tissues were cleared for 30 seconds with 12% sodium hypochlorite solution before microscopy observations.

### Tissue harvesting and microscopy methods

Roots of *M. truncatula* composite plants expressing different fluorescent fusions and grown under LUMOX film were observed using a Leica TCS SP8 AOBS laser scanning confocal microscope equipped with a 40x long distance water immersion objective (HCX APO L U-V-I 40x/0.80 WATER). Confocal images were recorded using Leica LAS-X software, before and after rhizobia inoculation (from 1 to 7 dpi). For GFP, CFP and DsRed or mCherry fluorescent proteins, 458 nm and 488 nm Argon laser lines and a 561 nm diode were used for excitation, respectively, with emission windows set at 465–495 nm, 500–525 nm and 570–600 nm (DsRed) or 600–630 nm (mCherry) (hybrid detector), respectively. To avoid potential interference, sequential mode was used to acquire CFP and red fluorescent protein fluorescence separately from GFP fluorescence and bright-field images. For confocal imaging of $Ca^{2+}$ responses, the 561 nm diode was used to excite the mApple red fluorescent protein from the NR-GECO1 sensor, and the emitted fluorescence was recovered using a hybrid detector in the 600–643 nm emission window. Time series were acquired at 5 s intervals for 10–15 min, with pinhole diameter set to 3 Airy units, as described[46].

Roots or nodulated roots of *M. truncatula* A17 and R108 (R108S and *ann1-3*) or composite plants were selected based on their fluorescence and/or analysed before or after GUS and/or β-galactosidase staining using stereomicroscopy (Leica S6E) or light microscope (AxioPlan II Imaging; Carl Zeiss, Oberkochen, Germany). For *pMtAnn1:GUS* (in pLP100) expression analysis, samples were harvested before and after (4 dpi) with rhizobia. For comparative phenotyping of wild-type and *MtAnn1* mutant or RNAi roots, β-galactosidase stained nodulated root samples were harvested 4–5 or 8–10 dpi with rhizobia. Roots expressing *pEXPA:NIN + pMtAnn1:GUS* were dehydrated in ethanol series and embedded in glycol methacrylate (Technovit 7100; Haereus-Kulzer) according to the manufacturer's instructions. Sections of 10 μm were counterstained with aqueous Basic Fuchsin solution (0.007%) for observations. To quantify nodule size and rhizobia infection levels, β-galactosidase-stained nodulated root systems (10 dpi) from R108S and *ann1-3* plants were scanned (Objectscan 1600, Microtek) and the acquired images (TIFF) were used for image quantification (see details in the following part of the methods).

For high-resolution microscopy analyses of pre-infection priming responses in A17, control roots and rhizobia-infected root regions or nodule primordia were carefully isolated from spot-inoculated or flood-inoculated root systems 5–6 dpi, after histochemical revelation of rhizobial β-galactosidase activity. Harvested root sections or nodule primordia were fixed in 2% glutaraldehyde, diluted in 0.2 M cacodylate buffer pH=7.3 for a few hours at room temperature or 1–3 days at 4 °C, then rinsed with buffer, before progressive dehydration in an ethanol series and final inclusion in LR White resin following manufacturer's instructions (EMS). For comparative ultrastructural analysis of isolated R108S and *ann1-3* nodules, nodule samples were carefully isolated from rhizobia-inoculated root systems 14–16 dpi and fixed under vacuum in 2 successive batches of 2.5% glutaraldehyde diluted in 0.2 M cacodylate buffer pH=7.3, the first batch containing saponin or triton (0.1% final) for 3–5 days at 4 °C. After rinsing in buffer, post-fixation in 2% osmium tetroxyde (diluted in 0.2 M sodium cacodylate), rinsing again in buffer, progressive dehydration in an ethanol series and incubation in propylene oxide (2 × 1 h), samples were finally embedded in Epon 812 resin (EMS) following the manufacturer's instructions. For all samples, semi-thin (1 μm) and ultra-thin (80 nm) sections were generated using an Ultracut E ultramicrotome (Reichert Jung). Semi-thin sections of root/nodule primordia (5–6 dpi) were stained with Basic Fuchsin (0.07% in water) to help visualise ITs, while mature nodule sections (14–16 dpi) were stained in an aqueous solution with methylene blue (0.2%), Toluidine Blue (1%), borax (1%) and Basic Fuchsin (0.07%) before observation by bright field microscopy (Axioplan II Imaging; Carl Zeiss). Ultrathin sections (80 nm) were contrasted with Uranyless and lead citrate (Delta microscopy) and observed using a Hitachi 7700 electron microscope (Hitachi High-Tech).

For histochemical GUS staining of mycorrhizal roots, selected composite plants transformed with *pMtAnn1:GUS* were transferred to 7 × 7 × 8 cm pots (5 plants/pot) filled with washed quartz sand (grain size 0.7–1.2 mm) pre-watered with modified half-strength Hoagland solution containing 20 μM phosphate[88]. Each plant was inoculated with 500 spores of *R. irregularis* DAOM197198 (C-grade, Agronutrition, Toulouse, France). Plants were grown in a Polyklima cabinet at 22 °C constant temperature, 60% air humidity and 16-h-light/8-h-dark cycles, at 200 μE/s/m2 light intensity and a combination of warm and cold LED 'True DayLight' at a 40:25 ratio. Pots were watered twice per week with 30–40 ml of autoclaved de-ionised tap water and fertilised once per week with 30–40 ml of modified half-strength Hoagland solution containing 20 μM phosphate. To visualise the fungal structures, the GUS staining solution was exchanged for 10% KOH and the roots incubated for 15 minutes at 95 °C. After KOH removal, the roots were washed 3 times with de-ionised water. After addition of 0.1 M HCl, the roots were further incubated for 2 h in the dark at room temperature. HCl was removed and the roots were washed 3 times with de-ionised water and once with PBS. This was followed by an overnight incubation at room temperature in the dark with 200 ng/μL WGA Alexa Fluor 488 (Invitrogen W11261) in PBS. Roots were imaged with a LEICA DM6B epifluorescence microscope. To compare the extent of mycorrhizal colonisation of roots between R108S and *ann1-3* lines, whole root systems were collected 6 weeks post-inoculation with *R. irregularis* and cleared by boiling (95 °C) in 10% KOH for 5 min, rinsed with water, then stained by boiling in an acidic ink solution (5% acetic acid, 5% Sheaffer

black ink #94321, in water) to reveal fungal structures. Overall intraradical colonisation rates and arbuscular density were estimated in ink-stained roots under a stereomicroscope (Leica S6E) using the gridline intersect method.

## Microscopy image analyses

Analysis of confocal time-series was performed using the Fiji software[89]. Intensity data were calculated from selected regions of interest (ROIs), corresponding to single nuclei, imaged in independent roots, in 2–3 independent experiments. Maximal z-projections of stacks and merged confocal images for illustration purpose were prepared using Leica confocal software or Fiji. To quantify relative amplitudes of $Ca^{2+}$ spiking, the mean fluorescence of each ROI was measured as a function of time and used to determine the Signal-to-Noise Ratio (SNR). The SNR measures the difference between the fluorescence at each time-point ($F_t$) and the fluorescence baseline, calculated as the average of 4 fluorescence values taken 10 and 15 s before and after t ($F_{t-15}$; $F_{t-10}$; $F_{t+10}$; $F_{t+15}$). A threshold of 0.3 was applied to the SNR to define individual peaks, and the amplitude of $Ca^{2+}$ spiking was calculated for each nucleus as the mean of peak SNR values. Spiking frequency, defined as the number of peaks observed in 10 minutes, was also calculated. In each experiment where $Ca^{2+}$ spiking was monitored, nuclei displaying no spiking were also included, resulting in the sample size (n) being higher in the frequency graph than in the associated amplitude graph.

For quantification of MtAnn1-GFP fluorescence levels, root hairs were imaged at either the RHE or IT stages with identical acquisition settings in 3 independent experiments. In each acquired confocal z-stack, 4–6 successive confocal sections encompassing the cytoplasmic aggregation around the site of rhizobia entrapment (RHE) or the cytoplasmic bridge characteristic of the IT stage were selected. In ImageJ (https://imagej.net/ij/), the threshold function was used to generate a mask based on a maximal z-projection of the selected sub-stack, from which the contours of the cytoplasmic zone (ROI) were defined. The mean grey value was then measured in the defined ROI on an average z-projection of the same sub-stack.

Scanned images of β-galactosidase-stained root systems from R108S and *ann1* mutants (10 dpi) were imported to Ilastik software[90] to enable machine-learning recognition of blue β-galactosidase-stained nodules (pixel-based classification method). Objects identified as nodules were exported in a new single segmentation file (.TIFF format) for analysis in ImageJ to measure specific nodule features (size and infection level, based on the blue intensity of rhizobial β-galactosidase activity).

## Transient expression assays in N. benthamiana leaves

An *A. tumefaciens* GV3101 strain carrying the *pMtAnn1:GUS* fusion construct (in pLP100) was used for infiltration studies in *N. benthamiana* alone or with *A. tumefaciens* GV3103 strains carrying *p35S:3HA-NIN* or *p35S:3HA-NINΔ* constructs[54], designed to constitutively express NIN or NINΔ under the 35S promoter in transactivation experiments. Briefly, *A. tumefaciens* strains grown overnight at 28 °C in LB medium with appropriate antibiotics were harvested and resuspended in Agromix (10 mM MgCl₂, 10 mM MES/KOH pH 5.6 and Acetosyringone in DMSO Sigma-Aldrich 150 µM) and kept in the dark for at least 2 h, at room temperature. Equal volumes of *A. tumefaciens* cultures ($OD_{600nm} = 0.25$) were used to co-infiltrate leaves of 3-week-old *N. benthamiana* plants. Infiltrated plants were kept at 21 °C in a growth chamber (16-h photoperiod and a light intensity of 70 mE/s/m2) for 36 h prior to collecting discs from infiltrated leaves, and subsequent histochemical GUS assays or storage at −70 °C prior to protein extraction for Western-blot or fluorimetric GUS assays.

## Western blot analyses

*N. benthamiana* leaf discs, previously stored at −70 °C, were ground using a MM400 (Retsch) crusher and resuspended in Laemmli 2X sample buffer for SDS-PAGE (Bio-Rad). Samples were then placed at 95 °C for 3 min and centrifuged at $16,000 \times g$ for 1 min. For each sample, 10 µL of the supernatant was loaded in a polyacrylamide 4–15% Mini-PROTEAN precast gel (Bio-Rad) along with 5 µL of pre-stained protein ladder marker (ThermoScientific, Lithuania). The Mini-Protean tank Electrophoresis System (Bio-Rad, USA) was used for gel migration in 1X Tris/Glycine/SDS Buffer (Bio-Rad). Protein transfer to nitrocellulose membranes (Bio-Rad) was performed using a Trans-Blot Turbo semi-dry transfer system (Bio-Rad). Membranes were stained with Ponceau Red before incubation for 1 h in blocking 1 × TBS solution (Tris base, NaCl and H₂O), 0.1% Tween, 5% milk at room temperature, with mild agitation, before rinsing 3× in TBS-Tween 0.1% for 10 min and incubated overnight in the same solution at 4 °C with a 6000X dilution of anti-HA-peroxidase antibodies (Sigma-Aldrich) in 0.5% milk in TBS-Tween 0.1% to reveal HA-tagged NIN proteins following chemiluminescence revelation using the Clarity Western ECL Substrate (Bio-Rad) and the ChemiDoc Touch imaging system (Bio-Rad).

## Acetylene reduction assay

To assess nitrogenase activity, grouped nodules were isolated from root systems of plants grown in pots 3 weeks after inoculation with *Sm2011-lacZ* and tested for acetylene reduction. Nodulated root fragments were incubated at 25 °C in sealed 60 mL vials containing 0.5 mL of nitrogen-free Fahraeus liquid medium in the presence of 10% (v/v) acetylene for 3 h. 400 µL of gas was then collected from each vial and ethylene production was quantified by gas chromatography (model no. GC 7280A; Agilent Technologies, Lexington, MA).

## RNA extraction and quantitative RT-PCR analysis

Total RNA was extracted from *M. truncatula* control and rhizobia-inoculated roots (4–5 dpi) using the Macherey-Nagel total RNA isolation kit according to the manufacturer's instructions. DNA-free RNA samples were quantified, and RNA integrity was verified by Agilent RNA Nano Chip (Agilent Technologies). First-strand complementary DNA synthesis was performed using 1 µg of total RNA using an anchored oligo (dT) and Transcriptor Reverse Transcriptase (Roche) following the manufacturers' protocol. Quantitative RT-PCR was performed in 384-well plates, with Roche LightCycler 480 or Bio-Rad CFX OPUS 384 Real-Time PCR systems and using the SYBR Green MasterMixes from Roche or Takyon (Eurogentec), according to manufacturer's instructions. Reactions and cycling conditions were performed as described before[24] using Primer pairs listed in Supplementary Table 1. The specificity of the PCR amplification was verified by analysing the PCR dissociation curve and sequencing of the PCR product. Transcript levels were normalised to the endogenous Ubiquitin reference.

## Phylogenetic analysis

Protein sequence of *MtAnn1* (MtrunA17_Chr8g0352611) was used as query to search against a database containing 227 plant genomes covering the main lineages of green plants and five SAR genomes as outgroups using the BLASTp v2.14.0+ with an e-value threshold of 1e-10[58]. Homologous proteins were then aligned using Muscle v5.1 with the 'super5' option[91] and trimmed to remove positions with more than 60% of gaps using trimAl v1.4.rev22[92]. The trimmed alignment served as matrix for phylogenetic reconstruction using FastTree v2.1.11-OpenMP[93]. To gain insights about the evolution of Ann1 in relation with the nitrogen-fixing symbiosis, the flowering clade containing orthologs of MtAnn1 has been extracted and proteins re-aligned using Muscle with default parameters before trimming as described above. Few spurious sequences with obvious mispositioning and abnormal short sequences were removed. Then, maximum likelihood tree has been reconstructed using IQ-TREE v2.1.2[94] after testing the best evolution model using ModelFinder[95] as implemented in IQ-TREE2 and according to the Bayesian Information Criteria. Branch supports have

been tested with 10,000 replicates of both SH-aLRT[96] and UltraFast Bootstraps with UFBoot2[97]. Finally, trees were visualised and annotated in the iTOL platform v6.8.1[98].

### Graph generation and statistical analyses

Graph preparation and statistical analyses were performed using R, except for graphs in Fig. 5c, d, which were generated using Microsoft Excel and for which contingency analyses were performed using GraphPad Prism 10. For all other analyses, data were presented as box plots, normal distribution of the data was evaluated using the Shapiro-Wilk test and homogeneity of variance was assessed using Fisher or Bartlett tests. When applicable, a transformation was performed to normalise data distribution ($Log_{10}$ or BoxCox). Parametric statistical tests (t-test, ANOVA) were used to analyse data with a normal distribution, while non-parametric equivalents (Mann-Whitney, Kruskal-Wallis) were used for data with a non-normal distribution. Number of individually analysed samples (n), replicates and $p$ significance levels are indicated in Figure legends. In detail, data was analysed as follows: In Fig. 2e values do not follow a normal distribution and were thus analysed using a two-tailed Mann-Whitney test ($W = 315$, $p = 0.1531$). Values in Fig. 2f follow a normal distribution and display homogeneous variance, so a two-tailed Student t-test was performed ($t = -3.0927$, df = 39, $p = 0.0037$). $Log_{10}$-transformed values of Fig. 4e follow a normal distribution and variance homogeneity and were thus analysed using one-way ANOVA followed by Tukey honest significant difference (HSD) tests ($F = 149.5$, df = 2, $p < 2e$-$16$). Values in Figs. 5a, b do not follow a normal distribution, hence Kruskal-Wallis tests were carried out ($K = 6.499$, $p = 0.0388$ for 5a and K = 6.204, $p = 0.0449$ for 5b). Figure 5c, d were built from contingency tables reporting the number of nodule primordia belonging to different categories: not infected (−) or partially to fully infected (+) in c, or with a single IT extending to the cortex (1 IT) or multiple ITs (≥2 ITs) in d. Relative proportions of these categories were compared one-on-one between R108S and each other genotypes (R108, *ann1-2* and *ann1-3*), by running separate Fisher's exact tests. Results are for Fig. 5c, R108S vs R108 $p = 0,8408$, R108S vs *ann1-2* $p = 0.0027$, R108S vs *ann1-3* $p = 0.0198$, and for Fig. 5d, R108S vs R108 $p = 0.7947$, R108S vs *ann1-2* $p = 0.0226$, R108S vs *ann1-3* $p = 0.001$. Values in Fig. 5m follow a normal distribution and variance homogeneity and were thus analysed using one-way ANOVA followed by Tukey HSD tests (F = 2.372, df = 3, $p = 0.076$), and values in Fig. 5n do not follow a normal distribution and were thus analysed by a Kruskal-Wallis test ($K = 55.44$ and $p = 9,17e$-$13$). In Fig. 5o, values follow a normal distribution and variance homogeneity, hence a two-tailed Student t-test was carried out ($t = 2.4153$, df = 99, $p = 0.0176$), while values in Fig. 5p do not follow a normal distribution, thus two-tailed Mann-Whitney tests were applied ($W = 1724$, $p = 0.0023$). Values in Fig. 6g follow a normal distribution but not homogeneity of variances, thus a Welsh two-sample t-test was carried out ($t = 4.8433$, df = 8.2905, $p = 0.0012$). $Log_{10}$-transformed values in Fig. 6h show a normal distribution and variance homogeneity and were thus analysed using a two-tailed Student t-test ($t = -3.066$, df = 27, $p = 0.0049$). Values in Fig. 7a, b, which show a normal distribution and variance homogeneity were analysed using one-tailed Student t-tests, due to the low number of values (7a: $t = 2.1821$, df = 9, $p = 0.0285$; 7b: $t = 0.9612$, df = 22, $p = 0.1735$). In Fig. 7c, values do not follow a normal distribution, thus a two-tailed Mann-Whitney test was performed ($W = 9944$, $p = 0.0018$). Values in 8d and BoxCox-transformed values in 8e ($\lambda = 1.353535$) show a normal distribution and homogeneity of variance and were thus analysed using two-tailed Student t-tests (8d: $t = 3.6816$, df = 72, $p = 0.00045$; 8e: $t = 3.8831$, df = 72, $p = 0.0002$). Distribution of values is not normal in Fig. 8f, so data was analysed via a two-tailed Mann-Whitney test ($W = 91032$, $p = 0.0046$), values of Fig. 8g follow a normal distribution so a two-tailed Student t-test was applied ($t = -1.6238$, df = 17, $p = 0.1228$). In Supplementary Fig. 1b, values do not follow a normal distribution, thus a Mann-Whitney test was carried out ($W = 35$,

$p = 0.005349$), while in Supplementary Fig. 1c values show a normal distribution and variance homogeneity, so a two-tailed Student t-test was performed ($t = 3.9724$, df = 10, $p = 0.0026$). In Supplementary Fig. 2e–h, values show a non-normal distribution and were thus analysed using two-tailed Mann-Whitney tests (Supplementary Fig. 2e: $W = 853$, $p = 1.811e$-$07$; 2f: $W = 205$, $p = 2.893e$-$05$, Supplementary Fig. 2g: $W = 348$, $p = 3.377e$-$09$, 2h: $W = 237$, $p = 1.926e$-$09$). In Supplementary Fig. 7c, values follow a normal distribution and variance homogeneity, hence a two-tailed Student $t$-test was performed ($t = 1.9307$, df = 6, $p = 0.1017$). Supplementary Fig. 12b, c, $Log_{10}$-transformed data follow a normal distribution and variance homogeneity, so one-way ANOVAs followed by Tukey HSD tests were carried out (respectively $F = 321.1$, df = 9, $p < 2e$-$16$ for Supplementary Fig. 12b and $F = 1.701$, df = 9, $p = 0.106$ for Supplementary Fig. 12c). In Supplementary Fig. 12d, values do not have a normal distribution, so a Kruskal-Wallis test was used ($K = 43.0318$, $p = 4.5265e$-$10$). In Supplementary Fig. 12e, data follows a normal distribution and variance homogeneity, so a one-way ANOVA followed by Tukey HSD tests was carried out ($F = 4.593$, df = 2, $p = 0.0143$). $Log_{10}$-transformed data in Supplementary Fig. 12f, and non-transformed data in Supplementary Fig. 12g follow a normal distribution and variance homogeneity, hence two-tailed Student $t$-tests were employed (respectively $t = 4.4053$, df = 22, $p = 0.0002$ for Supplementary Fig. 12f, and $t = 2.0775$, df = 22, $p = 0.0496$ for Supplementary Fig. 12g). In Supplementary Fig. 13c, e, data follow a normal distribution (after a $Log_{10}$-transformation for Supplementary Fig. 13c) and variance homogeneity, hence one-way ANOVAs followed by Tukey HSD tests were used for comparisons (respectively $F = 2.223$, df = 4, $p = 0.0765$ for Supplementary Fig. 13c and $F = 4.398$, df = 4, $p = 0.0034$ for Supplementary Fig. 13e). Values in Supplementary Fig. 13d do not follow a normal distribution, so statistical analysis was done using a Kruskal-Wallis test ($K = 9.15$, $p = 0.0575$). In Supplementary Fig. 13f, values follow a normal distribution and variance homogeneity, hence a two-tailed Student $t$-test was performed ($t = -3.51$, df = 89, $p = 0.0007$). Values in Supplementary Fig. 15c do follow a normal distribution and variance homogeneity, and were analysed using a two-tailed Student $t$-test ($t = -1.0271$, df = 10, $p = 0.3286$). In Supplementary Fig. 15d, e, values do not follow a normal distribution and were thus analysed using two-tailed Mann-Whitney tests ($W = 86$, $p = 0.57$ for Supplementary Fig. 15d, and $W = 18215$, $p = 0.3726$ for Supplementary Fig. 15e).

### Reporting summary

Further information on research design is available in the Nature Portfolio Reporting Summary linked to this article.

## Data availability

The authors declare that all data supporting the results of this study are available within the article and its Supplementary Information Files. Materials generated in this study, including the ImageJ macro used for quantification of nodule size and X-gal staining, are available from the corresponding author upon request. Source data are provided with this paper.

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

## Acknowledgements

The *M. truncatula* R108 mutant plants utilised in this research project, which are jointly owned by the Centre National De La Recherche Scientifique, were obtained from Noble Research Institute, LLC and were created through research funded, in part, by grants from the National Science Foundation, NSF #DBI-0703285 and NSF #IOS-1127155. We thank the Toulouse CMEAB platform service for assistance with electron microscopy, S. Bensmihen for providing the *pEXT:DMI3* construct[52] and Celine Remblière for providing seeds of the *M. truncatula dmi3* line carrying the *pEXT:DMI3* construct. This work was supported by French National Research grants TULIP ANR-10-LABX-41, COME-IN ANR-14-CE35-0007-01 to F.d.C.N. and LIVE-SWITCH ANR-19-CE20-0026-01 to F.d.C.N. and J.F. and by the European Research Council (ERC) under the European Union's Horizon 2020 research and innovation programme (grant No. 759731) to C.G. A.G. was funded by an ANR postdoctoral grant (ANR-19-CE20-0026-01), A.K., A.D. and L.P. were funded by PhD grants from the French Ministry of National Education and Research. We acknowledge the TRI-FRAIB imaging facility, member of the national infrastructure France-BioImaging supported by the French National Research Agency (ANR-10-INBS-04).

## Author contributions

Coordinated the work: F.d.C.N.; designed experiments: F.d.C.N., A.G., J.F., A.K., K.H., M.C.A., L.F., N.F.D.F., C.G.; performed experiments: A.G., J.F., A.K., K.H., M.C.A., L.F., A.D., N.F.D.F.; Machine learning methods development: A.G., A.L.R.; phylogenetic analysis: J.K., P.M.D.; analysed data: A.G., J.F., A.K., K.H., M.C.A., L.F., A.D., L.P., J.K., P.M.D., C.G., N.F.D.F., F.d.C.N.; wrote the manuscript: F.d.C.N., A.G. and J.F.

## Competing interests

The authors declare no competing interests.
