## [Peer Review File · Nature Communications]

REVIEWER COMMENTS

Reviewer #1 (Remarks to the Author):

The manuscript focuses on Ann1 and describes cellular events prior to and during intracellular infection. The data show that Ann1 is a marker for cytoplasmic alterations. The authors describe, in detail, ultrastructural changes in the cells prior to, and during, infection and reveal subtle alterations in the calcium spiking. They show that infection foci trigger Ann1 expression and that DMI3 is required for cytoplasmic bridge formation. They also demonstrate that ERN1 is required to maintain the correct spatial pattern of MtAnn1 expression and Ca²⁺ spiking. They show NIN is a regulator of Ann. ann1 insertion mutants show minor changes in infection and the amplitude of Ca²⁺ spiking. The mutants also show alterations in nodule shape along with modifications to the ultrastructure of cells within certain zones of the nodule.

The data are interesting and advance knowledge of Ann1 and infection related events. Most of the work is of very high quality and several of the techniques are challenging, so it is impressive to see their results. There are some aspects that appear to lack verification in a second allele, so those conclusions may need additional support (outlined below). Additionally, some of the stronger conclusions lack support and in some cases, a causal relationship has not been established. For example, the data in figure 2 are very interesting and beautifully done. However, they do not indicate a causal relationship between Ca spikes, Ann1 and infection, so the conclusion in lines 163 -165 should be adjusted.

Major points to be addressed:

Figure 1 highlights an unusual composition of ER, vacuole-like structures and perinuclear mitochondria. In how many independent samples has this been observed?

When points are made about the high density of a feature (eg mitochondria), some quantification is needed and also a statement indicating what this is being compared with ie 'unusually high density of mitochondria' relative to ?

Figure 4D. pEXPA:NIN appears to result in ANN1-GUS also in inner cell layers. Is this interpretation correct? How tight is the spatial expression pattern of this promoter? This is relevant for the following experiments but is not discussed in the text. Please add some discussion about this.

What is the significance of the drop in Ca²⁺ spiking amplitude? It would be useful to discuss this point.

It is unclear what is meant by resolute TEM?

How many of these phenotypes were observed in more than one Mtann allele? For the main phenotypes, evidence that this occurs in more than one allele must be presented. Do all alleles

display this nodulation phenotype? Do all alleles (or at least more than one allele) display the multiple IT phenotype shown in figure 6?

Are the calcium spiking changes in the ann1-3 allele also seen in the other alleles?

It may not be necessary to repeat everything in all alleles but it is essential to report whether or not these phenotypes were observed in more than one allele.

Is the MtAnn1 gene expressed in nodules? Detail information is provided for expression in roots hairs and cells adjacent to root hairs but not for later stages of development, yet there appears to be phenotypes associated with the nodule.

The significance is overstated. Yes, it has an effect but it is subtle. Are there other annexins that might compensate (even if not showing a strong induction?)

The discussion is very long and overly speculative.

Reviewer #2 (Remarks to the Author):

In the manuscript entitled “Annexin and calcium-regulated priming of legume root cells for endosymbiotic infection”, Guillory and co-authors demonstrate that an annexin of *Medicago truncatula*, MtAnn1, plays some role in pre-infection and infection by rhizobia and arbuscular mycorrhizal fungi and, therefore, represents one more example of a mechanism of this latter symbiosis recruited for root nodule symbiosis. The authors try to connect Ann1 to changes in calcium signatures during infection. However, I am far less convinced by this part of the manuscript.

Here are my main concerns:

1- The authors report (Figure 3) that cells with a growing infection thread (IT) have the same calcium spiking frequency as cells with a root hair entrapment (RHE) but a lower amplitude. Later, in Figure

7, they report changes in spiking amplitude in the *ann1-3* mutant. The frequency data are solid. However, without a proper reference for signal amplitude, the authors cannot draw any definitive conclusion on changes in signal amplitude. The NR-GECO calcium sensor is affected by physicochemical parameters such as pH or oxygen tension. The reported amplitude changes can result from such changes independent of calcium concentrations. These results are possibly interesting, but a positive control for signal amplitude in plant nuclei is required.

2- As far as I can see, the results presented in Figure 3 are not quantified. A qualitative estimate for calcium spiking presence (+ or ++) or absence (-) is presented. Still, there needs to be information about how many nuclei were observed for each cell type, calcium spiking frequency, and even which criteria were used to determine if a cell presents spiking. Would 2 spikes over 10 minutes, as presented in C1a, be considered + or ++? Is such a low frequency even relevant for the activation of downstream signaling?

3- Sieberer et al. (2011) have reported a switch to high-frequency calcium spiking concomitant with the entry of rhizobia and arbuscular mycorrhizal fungi into cortical root cells. By reading this manuscript, it is unclear whether the authors observed the same phenomenon or not. The authors should show this increased frequency in cortical cells if they observed it and discuss this if they did not.

4- The authors did not try to rescue any nodulation or mycorrhization phenotype of the *Medicago ann1-3* mutant. They used a sibling of the mutant as a positive control (WT), but it is still possible that some phenotypes reported may be due to other *Tnt1* mutations. This is, unfortunately, an important control when using heavily mutagenized plants.

5- The results showing ENOD11 expression in response to the epidermal-specific expression of NIN seem out of place in this manuscript. There is no connection with the rest of the manuscript on *MtAnn1*. I suggest removing the data presented in Figure 4 F-K from the manuscript as its importance is limited, and it distracts from the main message of the manuscript.

6- All the results on root nodule symbiosis or mycorrhizal phenotypes show statistically discernible differences due to their hard work and high number of replicates. However, the effect size is consistently small. The authors should acknowledge this small effect of the *ann1* mutation more clearly in the result section and discuss its causes in the discussion. Why is the effect size so small? Is there redundancy with other annexins?

Minor comments

* The novelty of the first paragraph of the result section and Figure 1 is limited as it shows pre-infection threads that have already been widely reported in the literature. I suggest using these images as a supplemental figure.

* Line 121, change 0,5 to 0.5.

* The authors need to check the consistency of their manuscript for the tense used to describe experiments. Most experiments are described in the past tense, but sometimes, they use the present tense, as in line 177.

* The role of NIN in mycorrhizal associations is quite debated (Guillautin et al., 2016 in *Frontiers in Plant Sciences*). The authors state that NIN is specifically involved in nodulation and not mycorrhizal associations. I recommend removing this statement and presenting the literature more objectively.

Reviewer #3 (Remarks to the Author):

The key aim of the manuscript by Guillory et al is to uncover the role of Ann1 in regulating cell specific Ca²⁺ responses to prime *M. truncatula* plant cells during transvacuolar cytoplasmic PIT/PPA formation. The authors show a correlation/co-occurrence between Ca²⁺ spiking (NR-GECO1) and Ann1-GFP localisation in cortical cells beneath ITs, however, direct evidence for a role of Ann1 in PIT and PPA ultrastructural remodelling remains inconclusive and as such conclusions drawn in this study are flawed (see point by point comments below). This is largely based on flaws in experimental design (ie absence of comparable controls), flaws in data analysis ie lack of adequate quantification of spiking data in mutant and comparable WT lines; also, flaws in imaging data with absence of suitable cytosolic marker to show ultrastructural remodelling/priming and lack of comparable controls when comparing various mutant and complemented material. The link between MtAnn1 expression and priming for RNS in the pre-infection zone and independent of IT formation has been known for some time and GUS data simply confirms this (10.1094/MPMI.1998.11.6.504). Phenotypic characterisation of RNS and AMS in *ann1* mutants also does not convincingly demonstrate the importance of Ann1 in modulating calcium-regulated cellular priming during PIT and PPA formation in RNS and AMS, respectively. A “deregulated Ca²⁺ amplitude profile” in the *ann1* mutants is also not convincing, nor is the lack of cytoplasmic

remodelling in *ann1* mutants because key cytoplasmic markers are lacking and TEM images are not sufficient to capture the dynamic nature of vesicle trafficking and cellular remodelling. Based on the above and comments below my recommendation is not to accept the manuscript in its current form for publication in Nature Communications.

Major Points:

Line 112: The manuscript includes beautiful TEM micrographs in Fig 1 that shows distinct ultrastructural features of cytoplasmic bridges in primed outer cortical cells. Please could the authors include confocal images of a fluorescently tagged cytoplasmic marker (e.g. exocytotic, mitochondrial, actin and/or ER) to define features of cytoplasmic bridge formation in WT and across all mutant and complemented material. This is particularly important in correlating Ann1 function/Ann1-GFP localisation with NR-GECO Ca²⁺ reporter and cytoplasmic restructuring/priming.

Line 133-165: The authors suggest an *in vivo* approach to monitor both responses simultaneously in individual root cells, which amounts to co-localisation of MtAnn1-GFP and the Ca²⁺ reporter. The authors should remove the word 'new' as a co-localisation approach of reporters is not novel. Second, MtAnn1-GFP fusions provide spatiotemporal information based on fusion protein localisation not function. Also, it remains to be shown at this point if MtAnn1-GFP is functional. Third, subcellular localisation of MtAnn1-GFP is not quantitative and should not be compared with quantitative data from Ca²⁺ spiking.

Line 134-136: "MtAnn1, encoding a Ca²⁺ and membrane phospholipid binding annexin protein, is expressed in outer cortical cells preparing for rhizobia infection¹³, where coincidentally Ca²⁺ spiking responses are transiently regulated⁴²." This should be shown/confirmed in a Supplementary figure. Indeed this could be shown using available GUS reporter lines.

Line 142-144: "These challenging analyses were performed in *M. truncatula* A17 (wild-type, WT) and in the *sun1* mutant." Please could the authors clearly indicate in figure legends if images shown are from WT or *sun1* mutants.

Following on from this in Fig 1 and Fig S1:

- Line 142 mentions that analyses were conducted in WT and *sun1* backgrounds, but only *sun1* is immediately apparent in Fig 2 and Supplementary figure S1. Please include WT within the supplementary figure or as new supplementary figure as we would like confirmation that *sun1* mutation does not affect early calcium spiking events. Omit "challenging"

- From the images in Fig S1A, B it is clear that not every root hair is responding. Please could the authors show the number of root hairs responding out of the total number of root hair cells imaged in WT and *sun1* images (could be shown as nr spike/nr imaged).

- Could the authors confirm if the Nr of spikes +Sm of n=72 and n=56 shown in Fig S1 E & F respectively represent the spiking frequency and amplitude across all cells and what proportion of cells showed spiking out of all cells imaged.

Line 146-7: The authors mention induction of Ca²⁺-spiking induced by Nod factors which supposedly resemble that of the slower response with Rhizobia, but this is not shown. Nod-factor response (in the absence of Rhizobia) would highlight the “susceptible zone” in WT and sunn2 mutants and should be shown. This is essential to define the ‘susceptible zone’ in non-infected (i) and subsequent infected cells.

Lines 152-153: Yes, GFP intensity for Ann1 labelling does appear higher in IT containing cells as opposed to RHE, however, the authors shouldn't say “clearly” without some measurements and more images in supplementary, not just one representative image. Additionally, this is not a direct measure of Ann1 expression, only protein accumulation, it would be more accurate to say that IT correlates with increased ANN1 accumulation in the root hair. However, this should still be quantified using fluorescence intensity measurements of the Ann1-GFP fusion and qRT-PCR/western blotting. If the authors also wish to mention Ann1 expression levels then this should be quantified directly using qPCR or GUS under the native pAnn1 and quantify expression, as was done later in the paper. Additionally, we assume this was conducted in a wild-type background, not an ann1 mutant- in which case, there is also native Ann1 expression which is not being quantified through this GFP visualisation, therefore please do not give this emphasis as proof of “higher” Ann1 expression.

Figure 2 “Ca²⁺ spiking amplitude drops along root hair IT development.”

- This figure only shows Ca spiking data for the sunn-2 mutant at 2-4dpi whilst WT is not shown. The authors should provide evidence that the same observation applies under WT conditions at this time point (albeit at lower frequency) to prove that it's not a specific to the sunn-2 mutant phenotype. Instead of providing comparable data for WT in Fig 2 the authors instead refer to Fig S1 which is not comparable to Fig 2 as Ca spiking is recorded at a different stage (ie the nodulation susceptible zone at 1dpi). Indeed in Fig S1 the authors should also include quantification of Ca²⁺ spiking in the sunn-2 mutants at 1dpi rather than showing Ca spiking in WT (A17) only.

Lane 159-161: “This labelling is specific to one or a few cells around the infection site (Fig. 3A-C). Strikingly, these MtAnn1-labelled cells consistently show low frequency Ca²⁺ spiking, whereas nearby cells not labelled with MtAnn1 do not show any Ca²⁺ spiking (Compare C1a-c in Fig. 3).”

- There are several nuclei showing red fluorescence in cells from NR-GECO1 below MtAnn1-GFP labelling and they all lack Ann1-GFP signal; moreover, cells to the left and right of MtAnn1-GFP is not clearly visible and C1c shows only half of a cell. A single spike in C1a and C1b profiles is also

not convincing. How frequently was this observed? This is a key finding of the paper yet quantification of Ca responses is not provided.

- Moreover, localisation of a cytoplasmic marker (e.g. ER marker as mentioned above) should also be shown in these cells to confirm that these cells have cytosolic hallmarks of primed cells rather than using MtAnn1-GFP to demonstrate cytosolic rearrangement.

Line 175-176: “This mutant (ern1) shows Ca²⁺ spiking and cytoplasmic-bridge formation (visualized with MtAnn1-GFP) in early infected root hairs”

- As mentioned above, localisation of a cytoplasmic marker should also be shown in cortical cells below IF to confirm that these cells have cytosolic hallmarks of primed cells. Since MtAnn1-GFP localises to only a few cells and in the presence of Ca²⁺ localises to membranes it not a suitable marker for cytosolic priming and does not evidence that adjacent cells are not undergoing cytoplasmic priming.

- Also, appropriate controls are lacking and therefore the conclusion that “ERN1 is not required for promoting pre-infection priming, but is somehow required to spatially restrict it” is not supported.

Fig 3 shows MtAnn-GFP translational fusions in WT (A17) at 3-4dpi and this time the authors observe cells with strong fluorescent signal, however this is only shown in WT (A17) but not in all mutant lines. Indeed Fig 3D only shows a schematic representation of Ca²⁺ spiking in Sm responsive cells in WT (A17) with a table summarising responses in mutants. Ca²⁺ spiking responses are shown as (+) vs (-) with the response in ern1 mutants in zone (3) shown as (++). What does this mean? How is this quantified and what values are presented by the (+). Is the observed (++) in zone 3 statistically significant?

- Imaging data ie localisation of MtAnn-GFP and Ca²⁺ spiking in ern and all other mutants should be shown in a main or supplementary figure rather than only showing image data for ern1 (Fig S2) and these should be shown next to comparable WT (R108) lines.

- The authors should show spiking amplitude and frequency in Fig S2D rather than individual spiking patterns in C1 cells of the ern1 mutant as well as in and adjacent cells lacking MtAnn1-GFP for all mutants.

- Again appropriate cytosolic markers should be included to make the link between spiking and cytosolic priming and comparable EV and WT (R108) controls should be included.

- Fig 3D also summarises the localisation and responses in ern1ern2 mutants but the image data is not shown nor are WT (108) controls included.

- Fig S2 shows AtAnn-GFP in the dmi3 epidermal specific complemented line, however, no controls are included – e.g. images for EV transformed dmi3 mutants should be provided (rather than referring to responses in a published dmi3 report which is not directly comparable). Comparable WT controls from same genetic background should also be included.

- Similarly spiking amplitude and frequency in Fig S2D should be quantified rather than showing single/individual spiking patterns in C1 cells (no quantification of spiking frequency or amplitude is provided and how often these patterns are observed is not clear).

Line 189-192:” The *ern1 ern2* and *dmi3* mutants, which are totally deficient in root hair bacterial entrapment and infection, show Ca^{2+} spiking in root hairs (49 and Supplementary Fig S2E and Movie 3) but no clear signs of cytoplasmic remodelling in root hairs or cortical cells”

- The authors cannot conclude that there’s a lack of cytoplasmic remodelling in mutants based solely on the expression of MtAnn1-GFP. Appropriate cytoplasmic markers (as mentioned above) are needed in these mutants to verify that cells adjacent to GFP signal lacks cytoplasmic remodelling whilst those with Ann1-GFP has cytoplasmic priming.

- In the absence of appropriate controls and image data conclusions drawn by the authors are not supported by the data.

Line 197-199: “While MtAnn1-labelled cytoplasmic bridges were strongly detected in infected root hairs of complemented *dmi3*, no pre-infection priming events were observed in adjacent cortical cells (no signs of MtAnn1 labelling, Ca^{2+} spiking or cytoplasmic remodelling, Supplementary Fig S2E-F).

- As mentioned earlier the authors need to include an cytoplasmic marker (ER or Actin) to confirm lack of cytoplasmic remodelling in *dmi3* mutants vs WT (R108).

Since the authors show TEM images for *ann1* mutants to demonstrate lack of cytoplasmic bridges they should similarly include TEM in *dmi3* and restoration of cytosolic bridge in complemented lines.

- Line 191-192:”these root hairs show only residual (in *ern1 ern2*) or no (in *dmi3*) MtAnn1-GFP expression” – please confirm in which Fig these images are shown?

Line 193-194:”Thus, the formation of MtAnn1-labelled cytoplasmic bridges”

- It has not been proven that MtANN1 is part of the cytoplasmic bridge (ie. this is a circular argument)

- experiments such as IGL-TEM (immunogold labelling) should be performed to show MtANN1 is part of the cytoplasmic bridge

- also see later comment – live cell confocal images showing lack of cytoplasmic markers in *ann1* mutant should be shown.

Line 226 – 235 “the in vivo dynamics of a fluorescent GFP-ER fusion was live tracked in nin complemented or not in the epidermis by the expression of a pEXPA:NIN construct” and Fig 4 addresses the dependence of MtAnn expression on NIN. Fig4 A-E shows pMtAnn1:GUS fusions at infection sites.

- To make the link between Ca²⁺ spiking, Ann1 levels and cytoplasmic remodelling the authors should transform nin complemented mutants with an appropriate cytosolic marker and co-localise with Ann1-GFP expression in IT, epidermal and cortical layers.

- ...”or not” Please could the authors show respective EV controls of all constructs

- In Fig. 4F-K an ER marker is initially used as a cytoplasmic marker (Fig 4F and I) and the authors then switch to DsRED in epidermal complemented lines. In other figures MtAnn1-GFP is used to show cytoplasmic bridges. This is confusing and inconsistent. It also makes it hard to compare cytoplasmic priming in nin mutants versus complemented lines where DsRED is used. Use the same cytoplasmic marker (ER, actin, mitochondrial) across experiments (see comments above).

- Lines 234-235: Would be nice to also complement nin mutants with NIN under a cortex-specific promoter and with MtAnn1-GFP to further emphasise this point

Fig 4E: Deletion of the NIN DNA binding domain does not prove direct interaction with the promoter of MtAnn1 as implied in line 221-222 but shows that it is needed to activate MtAnn1 expression. This activation could be direct or indirect. Therefore, the sentence should be revised to “These data establish NIN as a possible direct or indirect regulator of MtAnn1.” Please define NIN² in figure legend.

In Fig 3S WB shows a smaller band that cross-react with the anti-HA tag suggesting there might be a truncated HA-tagged NIN protein product that could be responsible for the transactivation of MtAnn1:GUS. Please could the authors comment on this?

- CHIP-PCR and DNA gel shift assays are needed to verify the possible direct interaction between NIN and the MtAnn1 promoter.

Fig S4 should include molecular validation of ann1-1/2/3 mutants.

- Fig S4B shows two lanes labelled as WT. To avoid confusion label first lane as R108WT and R108ann1 or similar.

- Fig S4C lacks nodule number in the ann1-1 mutant – please could the authors include this data as significant reduction in expression is shown in FigS4B.

- Figure S4D shows Medicago plants with colonised roots. ann1-2 and ann1-3 appears to have reduced size/developmental phenotype compared to WT – please could the authors quantify plant biomass in mutants. Include mock inoculated controls and the ann1-1 mutant.

- Lines 255-284 (and figures 5G-N): Was more than one nodule imaged for each figure- can it be confirmed that these images are representative and could some additional images be included in the supplementary. If this is the case and multiple nodules were imaged to assess the sizes of the different nodule zones, it would also be nice to show some statistical analysis to prove these are significantly differently sized between ann1-3 and WT.

- Also, why was only ann1-3 chosen for further phenotypic analysis? Some justification in the text would be preferable, especially since it appears from Supplementary Figure S4F that the differences in nodule phenotype e.g. colour and form are a bit more pronounced when comparing ann1-2 and WT than with ann1-3 versus WT

- Authors should include images of ann1-2 and ann1-3 functional complemented MtANN1-GFP to show restoration of nodulation comparable to EV or R108 WT controls.

Line 259-260: ann1-3 nodules show an abnormally large IZ like region, with over-accumulation of amyloplasts and less clearly defined zone III (Fig. 5H and J) and Fig 5 legend “The amyloplast-rich IZ and zone III are clearly defined in WT, while ann1-3 nodules (H, J) show a much larger IZ like region, with starch over-accumulation and no zone III comparable to that observed in WT nodules”

- This is an interesting finding. Could the authors include iodide staining confirm the overaccumulation of starch in the IZ in ann1-3 mutant vs WT and the ann1-3 complemented line.

- To confirm overaccumulation of starch the authors should also quantify starch levels in nodules from WT, ann1 mutants and ann1 complemented lines.

Lane 264-268. Although ann1-3 nodules include IC cells with apparent differentiated bacteroids (Fig. 5L, N), they comprise small vesicles (asterisks in Fig. 5N) that are normally found in IZ rather than zone III of WT nodules⁵⁴.

- Fig 5 shows beautiful TEM images, but TEM images should be shown for all nodule zones in WT to make it clear if there is an arrest of a zone compared to with the ann1 mutant. In other words does the IZ form normally in ann1 but does not transition into a fully defined zone III in the ann1 mutants?

- The TEM image in Fig 5K,M shows WT zone III but not WT IZ so it's difficult to determine if the IZ in WT and ann1 are comparable.

Lane 281-282 “Root hair ITs are formed in mutant roots at 4 dpi with rhizobia, but at a lower frequency compared with WT (Fig. 6A)”

- Although the authors show stained roots of both ann1-3 and ann1-2 quantification data is only shown for the ann1-3 allele. Please could the authors include quantitative data for ann1-2 allele and ideally also for the ann1-3 complemented roots?

- Fig 6D-I shows images of infected roots stained for rhizobia, however, images are not comparable across all genotypes. For example Fig 6D, F and H (WT) and 6E, 6I (ann1-3) should include staining of ann1-2 at a comparable stage and magnification.

- Similarly Fig 6F (WT) and 6G (ann1-2) should include a low magnification ie comparable image of ann1-3 mutant. Images of ann1-3 complimented roots should also be included.

Fig S6 is confusing as X-gal staining from bacteria and the GUS reporter both result in blue nodules and therefore bacterial infection cannot be compared between the RNAi knock down and control lines.

- RNAi knock down lines show a reduction in nodule number (and severity) compared to control lines, however, in Line 247-248 the authors note that “Phenotyping of ann1-2 and ann1-3 did not reveal any major changes in.... ..nodule number compared with the WT lines at 21 dpi (Supplementary Fig. S4C).” Please could the authors comment on this discrepancy? Could the RNAi construct be targeting additional Annexins to Ann1 that might be required for nodule development? To confirm this the authors should show expression levels of Ann1 and other members of the Annexin family in the RNAi lines.

- Following on comments relating to Fig 6 above, quantitative analysis of ITs in RNAi lines should be shown alongside that of the knockout lines (ann1-2 and ann1-3).

Fig 7 A-B: Spiking amplitude for WT and ann1-3 are shown in separate graphs which would indicate that these experiments were not carried out at the same time? Spiking amplitude in ann1-3 RHE seems reduced compared to WT although it's unclear if this reduction is significant compared with WT. Indeed, spiking amplitude in ann1-3 IT seems comparable to that of WT. Are these differences significant. Especially considering the low number of biological replicates used in RHE and IT with WT compared with mutants.

Lines 301-303: Ca²⁺ spiking and MtAnn1 dynamics accompany cytoplasmic bridge formation (Fig. 2-3), so we wondered how MtAnn1 mutation and deregulated Ca²⁺ spiking amplitude in ann1-3 (Fig. 7) could affect cytoplasmic bridge formation. We thus analysed the cytoplasmic composition of infected cells in apical nodule zones of WT and ann1-3 by resolute TEM.

- To illustrate the role of Ann1 in cytoplasmic bridge formation the authors should include confocal data with cytoplasmic markers (as suggested earlier) in the ann1 mutant (and MtAnn1-GFP) complemented lines which would be comparable to data shown in Fig 2-3.

- At the TEM level the extent of cytoplasmic restructuring is difficult to confirm because of the dynamic nature of these structures. Also, are cytoplasmic rearrangements found in primed outer cortical cells (as shown in Fig1C-E) lacking in ann1 mutants?

Fig 8A-B shows MtAnn1-GUS activity in AMF colonised roots. Although there appears to be a correlation between GUS staining and WGA-Alexa488, since both are shown in blue it is difficult to distinguish GUS from fungal structures. Please could the authors use a different false colour for WGA staining and clearly highlight the presence of arbuscules vs GUS signal. Empty vector controls should also be shown as well as mock inoculated roots to distinguish reporter activity from background staining.

Fig 8E-H shows ink stained roots colonised by AMF. Interestingly ann1-3 arbuscules appear small compared to WT. Please could the authors include WGA staining to show arbuscule morphology and to confirm if arbuscules are arrested in development or show accelerated turn over. Quantification of arbuscule size should be included.

Supplementary Figure S8: More information is needed in both the legend and the text to understand what outgroups were used to root the phylogeny tree (SAR outgroups where there is no previous information of what SAR means).

Line 418-421: "As MtAnn1 seems to partially co-localize with the peri-nuclear ER network in symbiotically activated cells, it could mechanistically play such a regulatory role, which would explain the disturbed Ca²⁺ signals and ER phenotypes caused by the MtAnn1 mutation."

- In the absence of IGL-TEM showing MtAnn1 localisation to the peri-nuclear ER network and the absence of live cell imaging using appropriate FP-tagged ER biomarkers in the ann1 mutant background the authors cannot draw this conclusion.

Similarly in Line 444-447: "Annexins are among Ca²⁺ sensing proteins involved in endomembrane trafficking in plants, either acting in conjunction with the membrane fusion machinery or regulating exocytosis. Thus, MtAnn1 may directly or indirectly impact Ca²⁺-regulated vesicle trafficking and/or vacuole morphology."

- The causal link between Ann1 and cytoplasmic remodelling is not clearly demonstrated and this could be shown through FP-tagged markers for exocytosis (e.g. Exo70, Vapyrin) and/or vesicle trafficking in the ann1 mutant background.

Small comments:

Line 290-291 "decided to assess if MtAnn1 mutation could somehow impact symbiotic Ca²⁺ spiking." Delete the word 'somehow'

Fig. 2 legend – line 1159(RHE, n=23 in A and n=24 in F) – does the authors mean(RHE, n=23 in E and n=24 in F).

Line 1269 (please check other Fig legends): “in genetic backgrounds mutated (ann1-3) or not (WT) in MtAnn1” please rephrase to ‘in ann1-3 mutant or WT (A17)’

Line 393: ...”in which the annexin MtAnn1 appears to be an important component” implies a role for MtAnn1 is shown in vacuolar biogenesis which is not shown; revise instead to ‘in which the annexin MtAnn1 might play a role’

Line 404-407: “The deregulated Ca²⁺ amplitude profile and defective infection caused by MtAnn1 mutation (Fig. 6 and 7) supports the need for such a strict control and establish a direct functional link between MtAnn1 and the regulation of Ca²⁺ spiking amplitude during rhizobia infection.” A direct mechanistic link between Ca²⁺ spiking and rhizobial infection is not proven in this study. The authors should revise this sentence to “The deregulated Ca²⁺ amplitude profile and defective infection caused by MtAnn1 mutation (Fig. 6 and 7) points to a functional link between MtAnn1 and regulation of Ca²⁺ spiking amplitude during rhizobia infection”

Line 687: ...”were contrasted with Uranylless and lead citrate...” does the authors mean ...’were contrasted with Uranyl Acetate and lead citrate...’

Dear Editor and Reviewers,

Please find attached a revised version of our manuscript, which addresses most of the points raised by the reviewers. To accommodate these changes, the manuscript text and figures have been revised and amended where necessary. New supplementary material (S1-S16) is also included. We also thank the reviewers for their comments, which together helped to improve the current version of the manuscript. A point-by-point response to the reviewers' comments (>>>) is provided below, including a description of the changes made in this new manuscript version (highlighted in yellow in the text).

Point-by-point responses to reviewers (>>>):

Reviewer #1 (Remarks to the Author):

The manuscript focuses on Ann1 and describes cellular events prior to and during intracellular infection. The data show that Ann1 is a marker for cytoplasmic alterations. The authors describe, in detail, ultrastructural changes in the cells prior to, and during, infection and reveal subtle alterations in the calcium spiking. They show that infection foci trigger Ann1 expression and that DMI3 is required for cytoplasmic bridge formation. They also demonstrate that ERN1 is required to maintain the correct spatial pattern of MtAnn1 expression and Ca²⁺ spiking. They show NIN is a regulator of Ann. ann1 insertion mutants show minor changes in infection and the amplitude of Ca²⁺ spiking. The mutants also show alterations in nodule shape along with modifications to the ultrastructure of cells within certain zones of the nodule.

The data are interesting and advance knowledge of Ann1 and infection related events. Most of the work is of very high quality and several of the techniques are challenging, so it is impressive to see their results. There are some aspects that appear to lack **verification in a second allele**, so those conclusions may need additional support (outlined below). Additionally, **some of the stronger conclusions lack support** and, in some cases, a causal relationship has not been established. For example, the data in figure 2 are very interesting and beautifully done. However, they **do not indicate a causal relationship between Ca spikes, Ann1 and infection, so the conclusion in lines 163-165 should be adjusted**.

>>>Thank you for your positive and constructive assessment.

-Phenotypic analysis in a second allele is now included, as explained below in the response to point n°6.

-Regarding the comment about lines 163-165: We now provide a more complete analysis of calcium spiking responses and MtAnn1-GFP dynamics in more cells from independent sites and genotypes (A17 and *sun1*) (see Fig. 2-3 and S7-S8). In all cases, calcium spiking was only seen in MtAnn1-GFP-labelled cells, strengthening their interconnection. As requested by the reviewer, we have adjusted the conclusions of the new revised version, in lines 199-201.

Major points to be addressed:

- 1) Figure 1 highlights an unusual composition of ER, vacuole-like structures and perinuclear mitochondria. In how many independent samples has this been observed?

>>>Detailed TEM analysis of primed cells was performed on n=8 sections from 5 independent infection sites from 2 independent experiments. This information has been added in the legend of Fig. 1. Images of other independent sites with primed cells are now shown in Supplementary Fig. S1.

- 2) When points are made about the high density of a feature (eg mitochondria), some quantification is needed and also a statement indicating what this is being compared with ie 'unusually high density of mitochondria' relative to?

>>>Thanks for this pertinent comment. The clustering of mitochondria around the nucleus was striking, as it was not seen either in mock control roots (now shown in Fig 1H-J) or in other supposedly metabolically active dividing cells in the nodule primordia. To quantify this, we counted the number of mitochondria visible in a given plane of independent 80 nm TEM sections (from different sites and biological experiments, as now detailed in the legends of Fig. 1 and S1), in close contact with the nuclear envelope or within a 2 μm radius. This revealed statistical differences between primed cells and NP dividing cells. The data are described in Fig. S1 and in the Result section (lines 127-131).

- 3) Figure 4D. pEXPA:NIN appears to result in ANN1-GUS also in inner cell layers. Is this interpretation correct? How tight is the spatial expression pattern of this promoter? This is relevant for the following experiments but is not discussed in the text.

>>>The pEXPA promoter does not drive expression in internal tissues, it is an epidermal-specific promoter (See Vernié et al. 2015). Sorry for the misunderstanding, this is due to the image shown in Fig. 4A, taken from an entire root system by focusing on root hairs in one plane. The blue colour at the bottom was due to staining of root hairs on the other side of the root and not in the inner tissues. To clarify this, we have replaced the image with two other images: one of a whole root system, which better shows the epidermal expression (Fig. 4E) and a second image corresponding to a 10 μm longitudinal root section (Fig. 4F), which unambiguously shows GUS activity only in the epidermal cell layer.

- 4) Please add some discussion about this: What is the significance of the drop in Ca^{2+} spiking amplitude? It would be useful to discuss this point.

>>> A decreased Ca^{2+} spiking amplitude was also reported to occur in outer cortical cells during IT passage (Sieberer et al 2012). Thus, our data are consistent with what Sieberer et al showed in the cortex. This, combined with our data, suggests that reduction in Ca^{2+} spiking amplitude is a common feature of infected cells. Ca^{2+} amplitude regulation fine-tunes several biological processes in animals and plants, and this is discussed in lines 410-417 of the revised manuscript.

- 5) It is unclear what is meant by resolute TEM?

>>>Transmission electron microscopy (TEM) gives access to ultrastructural changes occurring in the cell (organelle features and distribution) which cannot be provided by confocal imaging. As such, it is a more resolute approach. We clarified this in the manuscript, lines 119-121 and 331-334.

- 6) How many of these phenotypes were observed in more than one Mtann allele? For the main

phenotypes, evidence that this occurs in more than one allele must be presented. Do all alleles display this nodulation phenotype? Do all alleles (or at least more than one allele) display the multiple IT phenotype shown in figure 6?

>>>Fig. 5 and Fig. S13 now provide a detailed description of the nodulation and rhizobia infection phenotyping that were performed in two independent *MtAnn1* mutant alleles (*ann1-2* and *ann1-3*). We validated the knockdown of *MtAnn1* in both mutant alleles by Q-RT-PCR (Fig. S12) and performed a detailed phenotypic analysis of nodule shape, nodule number, root and shoot weight (Fig. S13) and infection by rhizobia (Fig. 5). Both alleles show a similar phenotype of infection deficiency with the multiple IT phenotype, which is also observed in the RNAi lines (Fig. 5). These new data are now described in the result session lines 268-311 and figures (Fig. 5, S12 and S13).

7) Are the calcium spiking changes in the *ann1-3* allele also seen in the other alleles? It may not be necessary to repeat everything in all alleles, but it is essential to report whether these phenotypes were observed in more than one allele.

>>> We believe that we can justify the use of a single allele here to perform these non-trivial analyses, especially during infection, as we show that *MtAnn1* is downregulated in both alleles (Fig. S12) and that they show similar phenotypes (Fig. 5 and Fig. S13). Note that there are many publications investigating symbiotic calcium responses in mutants, mostly using a single mutant allele (Waiss et al., 2020; Sun et al. 2007; Morieri et al., 2013...).

8) Is the *MtAnn1* gene expressed in nodules? Detail information is provided for expression in roots hairs and cells adjacent to root hairs but not for later stages of development, yet there appears to be phenotypes associated with the nodule.

>>>Detailed analyses of *MtAnn1* expression in nodules have been published previously by RNA in situ hybridization (de Carvalho-Niebel et al., 1998). The gene is expressed in nodule pre-infection and infection zones. Thus, the nodule phenotypes (Fig. 6, Fig.7 and S13-S14) are compatible with the spatial expression profile of the gene in nodules. Defective nodule differentiation in the mutant is likely a consequence of the maintained deregulated infection in nodules. We now clarify this in the text, lines 329-331.

9) The significance is overstated. Yes, it has an effect, but it is subtle. Are there other annexins that might compensate (even if not showing a strong induction?)

>>>We agree that the effect is not drastic. However, it is visible at different stages and significantly affects the proportion of infected nodules and their differentiation over time (see Fig. 5-6, S12-S14). Although *MtAnn2* showed increased levels in *ann1-2* and *ann1-3* mutant alleles in rhizobia inoculated samples, suggestive of a possible compensatory effect, these were not statistically significant in comparison to controls (Fig. S12). On the other hand, downregulation of *MtAnn1* and *MtAnn2* via an RNAi strategy (see Fig. S12) showed a stronger effect, both on nodule numbers and infection levels (see Fig. 5Q-Z), testifying the likely redundancy between them. See lines 268-311, Fig. 5 and S12-S13 for data description.

10) The discussion is very long and overly speculative.

>>>We reduced the discussion, from 1883 to 1323 words.

Reviewer #2 (Remarks to the Author):

In the manuscript entitled “Annexin and calcium-regulated priming of legume root cells for endosymbiotic infection”, Guillory and co-authors demonstrate that an annexin of *Medicago truncatula*, MtAnn1, plays some role in pre-infection and infection by rhizobia and arbuscular mycorrhizal fungi and, therefore, represents one more example of a mechanism of this latter symbiosis recruited for root nodule symbiosis. **The authors try to connect Ann1 to changes in calcium signatures during infection. However, I am far less convinced by this part of the manuscript.** Here are my main concerns:

1) The authors report (Figure 3) that cells with a growing infection thread (IT) have the same calcium spiking frequency as cells with a root hair entrapment (RHE) but a lower amplitude. Later, in Figure 7, they report changes in spiking amplitude in the *ann1-3* mutant. The frequency data are solid. However, without a proper reference for signal amplitude, the authors cannot draw any definitive conclusion on changes in signal amplitude. The NR-GECO calcium sensor is affected by physicochemical parameters such as pH or oxygen tension. The reported amplitude changes can result from such changes independent of calcium concentrations. These results are possibly interesting, but a positive control for signal amplitude in plant nuclei is required.

>>> We see a deregulated Ca^{2+} spiking amplitude pattern in the *ann1-3* mutant both in infected root hairs (Fig. 7A-B, compared to the wild-type R108S control) and in non-infected rhizobia-responsive root hairs (Fig. 7C). Although the numbers are low in infected cells (7A-B) because these events are difficult to measure, they are significant.

>>>In the case of root hairs from the nodulation competence zone, we were able to analyse a high number of independent cells (R108S, $n=160$ and *ann1-3*, $n=156$, from three independent experiments) and the differences are highly significant in the case of calcium amplitude and NOT for calcium frequency (see Fig 7C and Fig. S15). We are quite confident about the conclusions we can draw here since these data are based on the analysis of an exceptionally high number of independent events, which is not often done in this field (for comparison, see Fig. 3, del Cerro et al 2022 PNAS, a recent publication by a well-recognised team in this field). This is now better explained in the text (lines 319-326).

>>>The reviewer expressed a reservation about the validity of the GECO sensor for the analyses that we are presenting in the paper. This reservation is largely unjustified and we have several arguments, either from the literature or from our data, to support our position:

- With regard to the reviewer's statement that R-GECO1 may have pH sensitivity: It is likely that the reviewer is referring to the work of Keinath et al, 2015 (Mol Plant), which reported a certain degree of sensitivity of R-GECO1, specially observed under extreme pH conditions (close to 8). However, given the estimated pH in the nuclear compartment (close to 7) (Shen et al 2013, Mol Plant) where our NR-GECO1 probe is expressed, it is unlikely that this would be an issue in our system. In addition, this pH sensitivity was not supported by another recent study (Li et al., 2020 New Phytol) where Ca^{2+} (with R-GECO1) and pH were measured together in different plant cell types.

- Furthermore, comparative sensor studies have shown that R-GECO1 reliably measures calcium responses in a comparable qualitative manner to other ratiometric sensors (camelon or R-GECO1-mTurquoise) during ABA signalling, pollen tube growth or symbiosis (Waadt et al. New Phytol 2017;

Ngo et al. 2014 Developmental Cell; Kelner et al., 2018 Frontiers in Plant Sci.). This, together with the fact that R-GECO1 has been the sensor of choice in a large number of reliable studies in animals and plants, including two recently published high quality plant field papers (Howell et al 2023 Nature Plants; Leitao et al Nature Comm), supports the reliability of R-GECO1 for our studies.

>>>In the new manuscript version, we better justify the use of the nuclear R-GECO1 sensor and present data confirming its ability to detect the same infection-related calcium responses previously detected with the cameleon FRET sensor (Sieberer et al. 2012). This is shown in Figures S2-S3 and Movies 1-2, and amended in the text in lines 151-167.

2) As far as I can see, the results presented in Figure 3 are not quantified. A qualitative estimate for calcium spiking presence (+ or ++) or absence (-) is presented. Still, there needs to be information about how many nuclei were observed for each cell type, calcium spiking frequency, and even which criteria were used to determine if a cell presents spiking. Would 2 spikes over 10 minutes, as presented in C1a, be considered + or ++? Is such a low frequency even relevant for the activation of downstream signaling?

>>> Thanks for your comment, the symbols referred to presence or absence of responses. We realized that the information in the Table was unclear. To avoid misunderstanding we removed it and added information on the number of cells analysed/event in Figure legends. Calcium data analysis details are described in "Microscopy image analysis" methodological section (line 695 on).

>>>The low frequency calcium spiking described in this study are consistent with data shown by Sieberer et al 2012, using a FRET-cameleon sensor. These low frequency signatures, are most likely functionally relevant, since (i) they are consistently observed in outer cortical cells below an infection site, using NR-GECO1 alone or combined with MtAnn1-GFP in both wild-type and *sun1* backgrounds (See Fig. 3, S3 and S8, and explanation in the text, lines 156-167 and 193-201) and (ii) are impaired in defective symbiotic mutants (e.g. *dmi3* complemented and *nin* backgrounds, see Fig. S9 and Fig. 4).

3) Sieberer et al. (2011) have reported a switch to high-frequency calcium spiking concomitant with the entry of rhizobia and arbuscular mycorrhizal fungi into cortical root cells. By reading this manuscript, it is unclear whether the authors observed the same phenomenon or not. The authors should show this increased frequency in cortical cells if they observed it and discuss this if they did not.

>>>The focus of this paper was to analyse the priming response of outer cortical cells preparing for infection. Nevertheless, we also observed the switch to high-frequency reported by Sieberer et al. 2014. This is now shown in Figure S8.

4) The authors did not try to rescue any nodulation or mycorrhization phenotype of the *Medicago ann1-3* mutant. They used a sibling of the mutant as a positive control (WT), but it is still possible that some phenotypes reported may be due to other *Tnt1* mutations. This is, unfortunately, an important control when using heavily mutagenized plants.

>>> We have now included detailed phenotypic analysis of two mutant alleles (Fig. 5, S12 and S13), which showed similar phenotypes and reduced *MtAnn1* expression levels. The impact of *MtAnn1* down-regulation was also supported by the RNAi approach.

>>>Complementation studies with mutants that have partial phenotypes is not an easy task. However, we have shown by molecular analysis that the reduced expression of *VPY* in the mutant (n=19) is at an

intermediate level in complemented plants (n=17) compared to the wild-type control (n=22). The data are from two independent experiments and are shown in Figs. S12F-G.

5) The results showing ENOD11 expression in response to the epidermal-specific expression of NIN seem out of place in this manuscript. There is no connection with the rest of the manuscript on MtAnn1. I suggest removing the data presented in Figure 4 F-K from the manuscript as its importance is limited, and it distracts from the main message of the manuscript.

>>> Our objective here was to monitor if (1) cytoplasmic bridges were still formed in the *nin* background and (2) if we could restore their formation by restoring NIN expression through epidermal complementation. The *pMtAnn1:MtAnn1-GFP* construct was successfully used in A17 and *sunn* to detect bridge formation. However, as *MtAnn1* expression is genetically dependent on NIN (see Fig. 4A-G) we cannot use it in the *nin* mutant, otherwise the fusion will not be expressed (no NIN, no fusion activation). To circumvent that, we used another fusion (*pENOD11:GFP-ER*) that is not under the genetic dependency of NIN (so it will be expressed regardless of if NIN is present or not). Using this fusion, we can then visualize GFP-ER-labelled cytoplasmic bridges (Fig. S4-S6) that are restored in root hairs when NIN is expressed under the epidermis-specific promoter. We have now better explained this in the results section (lines 250-263).

6) All the results on root nodule symbiosis or mycorrhizal phenotypes show statistically discernible differences due to their hard work and high number of replicates. However, the effect size is consistently small. The authors should acknowledge this small effect of the *ann1* mutation more clearly in the result section and discuss its causes in the discussion. Why is the effect size so small? Is there redundancy with other annexins?

>>>We agree that the effect is not drastic. However, it is visible at different stages and significantly affects the proportion of infected nodules and their differentiation over time (see Fig. 5-6, S12-S14). On the other hand, downregulation of *MtAnn1* and *MtAnn2* via an RNAi strategy (see Fig. S12) showed a stronger effect, both on nodule numbers and infection levels (see Fig. 5Q-Z), testifying the likely redundancy between them. See lines 285-293 and legends of Figs. Fig. 5 and S12-S13 for data description.

Minor comments

7) The novelty of the first paragraph of the result section and Figure 1 is limited as it shows pre-infection threads that have already been widely reported in the literature. I suggest using these images as a supplemental figure.

>>> Pre-infection structures have been shown to occur in nodules of legumes and actinorhizal plants (Van Brussel et al 1992; Berg et al., 1999) or to involve microtubule rearrangements in *Medicago sativa* and *M. truncatula skl* mutant roots. However, knowledge on the cellular features of this process was largely unknown. Our work is therefore novel and provides for the first time a detailed ultrastructural analysis of this process, which has never been done before in any system. We now explain it better in lines 116-121.

8) Line 121, change 0,5 to 0.5.

>>> Thanks for noticing this typo, which was corrected.

9) The authors need to check the consistency of their manuscript for the tense used to describe experiments. Most experiments are described in the past tense, but sometimes, they use the present.

>>> We harmonized tenses throughout the manuscript.

10) The role of NIN in mycorrhizal associations is quite debated (Guillautin et al., 2016 in *Frontiers in Plant Sciences*). The authors state that NIN is specifically involved in nodulation and not mycorrhizal associations. I recommend removing this statement and presenting the literature more objectively.

>>> We agree that the role of NIN in mycorrhizal interactions is up to debate. We edited the sentence (lines 468-469) to avoid misleading interpretations.

Reviewer #3 (Remarks to the Author):

The key aim of the manuscript by Guillory et al is to uncover the role of Ann1 in regulating cell specific Ca²⁺ responses to prime *M. truncatula* plant cells during transvacuolar cytoplasmic PIT/PPA formation. The authors show a correlation/co-occurrence between Ca²⁺ spiking (NR-GECO1) and Ann1-GFP localisation in cortical cells beneath ITs, **however, direct evidence for a role of Ann1 in PIT and PPA ultrastructural remodelling remains inconclusive and as such conclusions drawn in this study are flawed** (see point by point comments below). This is largely based on flaws in experimental design (ie absence of comparable controls), flaws in data analysis ie lack of adequate quantification of spiking data in mutant and comparable WT lines; also, **flaws in imaging data with absence of suitable cytosolic marker to show ultrastructural remodelling/priming and lack of comparable controls when comparing various mutant and complemented material**. The link between MtAnn1 expression and priming for RNS in the pre-infection zone and independent of IT formation has been known for some time and GUS data simply confirms this (10.1094/MPMI.1998.11.6.504). Phenotypic characterisation of RNS and AMS in *ann1* mutants also does not convincingly demonstrate the importance of Ann1 in modulating calcium-regulated cellular priming during PIT and PPA formation in RNS and AMS, respectively. A “deregulated Ca²⁺ amplitude profile” in the *ann1* mutants is also not convincing, nor is the lack of cytoplasmic remodelling in *ann1* mutants because key cytoplasmic markers are lacking, and TEM images are not sufficient to capture the dynamic nature of vesicle trafficking and cellular remodelling. Based on the above and comments below my recommendation is not to accept the manuscript in its current form for publication in *Nature Communications*.

Major Points:

1) Line 112: The manuscript includes beautiful TEM micrographs in Fig 1 that shows distinct ultrastructural features of cytoplasmic bridges in primed outer cortical cells. Please could the authors include confocal images of a fluorescently tagged cytoplasmic marker (e.g. exocytotic, mitochondrial, actin and/or ER) to define features of cytoplasmic bridge formation in WT and across all mutant and complemented material. This is particularly important in correlating Ann1 function/Ann1-GFP localisation with NR-GECO Ca²⁺ reporter and cytoplasmic restructuring/priming.

>>> The formation of the cytoplasmic bridge in root hairs has been visualized using several *in vivo* markers. This includes a GFP-HDEL marker (Fournier et al 2008), labelling the ER endomembrane cytoplasmic network, or a cytoplasmic DsRed fluorescent marker (Liu et al 2019, *Nat Comm*), which we

now used here to certify cytoplasmic remodelling. We chose to use MtAnn1 here because: (i) MtAnn1 is a biochemically active annexin (de Carvalho-Niebel et al 2002), that (ii) localizes to the cytoplasm (as validated by confocal imaging and immunolocalization methods, de Carvalho-Niebel et al 2002) and is additionally expressed in rhizobia-infected root hairs and primed cortical cells (de Carvalho-Niebel et al 2002; Breakspear et al., 2014). Thus, it is an excellent tool to study cytoplasmic restructuring/priming in parallel with calcium responses during rhizobia infection.

To clarify this point:

- We have provided new confocal data of cytoplasmic bridge labelling by both MtAnn1-GFP and an ER-GFP marker (Fig. S4-S6). Both markers label cytoplasmic bridges in root hairs and cortex of A17 and *sunn* plants. Furthermore, expression of MtAnn1-GFP under its native promoter allows specific labelling of cytoplasmic bridges of pre-infection primed cells and not of neighbouring non-activated cells (see Fig. S6), making it an ideal tool for the remodelling studies we wanted to perform. The changes in the text can be found in the text (lines 168-182) and in Fig. S4-S6.

2) Line 133-165: The authors suggest an *in vivo* approach to monitor both responses simultaneously in individual root cells, which amounts to co-localisation of MtAnn1-GFP and the Ca²⁺ reporter. The authors should remove the word 'new' as a co-localisation approach of reporters is not novel. Second, MtAnn1-GFP fusions provide spatiotemporal information based on fusion protein localisation, not function. Also, it remains to be shown at this point if MtAnn1-GFP is functional. Third, subcellular localisation of MtAnn1-GFP is not quantitative and should not be compared with quantitative data from Ca²⁺ spiking.

>>> What is novel is the simultaneous imaging of calcium spiking and cytoplasmic remodelling. We have nonetheless removed the word "new". The sentence has been changed in lines 141-143.

>>> Indeed, the subcellular dynamic analysis of calcium spiking and MtAnn1 provides information on whether these responses occur in the same spatiotemporal frame. We have removed the word "functionally" in lines 140-143.

>>> We agree that "the subcellular localisation of MtAnn1-GFP is not quantitative". However, the relative fluorescence of the fusion (Fig. S7) was often stronger in infected root hairs, and this increase was consistent with root hair transcriptomic data (Breakspear et al 2014) on *MtAnn1* expression in infected root hairs. Thus, we are not claiming this, but suggesting an inverse correlation between these responses. We have now included the relative quantification of the MtAnn1-GFP fluorescence signal (Fig. S7) and changed the text accordingly (lines 189-192).

3) Line 134-136: "MtAnn1, encoding a Ca²⁺ and membrane phospholipid binding annexin protein, is expressed in outer cortical cells preparing for rhizobia infection¹³, where coincidentally Ca²⁺ spiking responses are transiently regulated⁴²." This should be shown/confirmed in a Supplementary figure. Indeed, this could be shown using available GUS reporter lines.

>>> There is already published data showing *MtAnn1* expression by promoter-GUS in outer cortical cells (de Carvalho-Niebel et al 2002). This is again illustrated by promoter-GUS in Fig 4B-C. The data description was included in lines 239-241.

4) Line 142-144: "These challenging analyses were performed in *M. truncatula* A17 (wild-type, WT) and

in the sunn2 mutant." Please could the authors clearly indicate in figure legends if images shown are from WT or sunn mutants.

>>>Information is now clearly indicated in Fig. legends.

5) Following on from this in Fig 1 and Fig S1: Line 142 mentions that analyses were conducted in WT and sunn2 backgrounds, but only sunn2 is immediately apparent in Fig 2 and Supplementary figure S1. Please include WT within the supplementary figure or as new supplementary figure as we would like confirmation that sunn2 mutation does not affect early calcium spiking events. Omit "challenging"

>>>The tracking of live responses in single infected root hairs is truly a very challenging experiment. Therefore, these analyses cannot provide as much data in the wild-type background as in *sunn*. This is because infection events are rarer in the wild-type context for *in vivo* follow up by confocal microscopy. It should be noted that each site of interest was imaged multiple times on living roots, allowing responses to be monitored dynamically and confirmed over time by follow-up observations. This has been clarified in the text in lines 146-150. (The word "challenging" has also been removed).

>>>We now provide data on early calcium spike responses in both A17 and in the *sunn* mutant, as requested by the reviewer. The data are shown in Figure S2 and Movies 1-2 and described in lines 156-159.

6) From the images in Fig S1A, B it is clear that not every root hair is responding. Please could the authors show the number of root hairs responding out of the total number of root hair cells imaged in WT and sunn images (could be shown as nr spike/nr imaged).

>>>Yes, not all root hairs respond, this is the normal "real time" variation. But the differences between control and *S. meliloti* inoculated roots are very clear. The cartoon (Fig. S1A) and image (Fig. S1C) illustrate the residual fluorescence of NR-GECO1 in control root hairs, which in most cases do not spike, compared to *S. meliloti*-treated root hairs (Fig. S1B and D). The 13-22% residual 'spiking' root hair nuclei observed in controls are very low compared to the average spiking levels in *S. meliloti*-inoculated root hairs (Fig. S2E-G), generally close to 5 spikes in 10 min. The ratios 'number of spiking nuclei/number of imaged nuclei (= in focus)' have been added to the figures for controls and *S. meliloti*-inoculated root hairs. Details can now be seen in the new supplementary Fig. S2.

7) Could the authors confirm if the Nr of spikes +Sm of n=72 and n=56 shown in Fig S1 E & F respectively represent the spiking frequency and amplitude across all cells and what proportion of cells showed spiking out of all cells imaged.

>>> The information has been added to the legend of Fig. S2 and the proportion of spiking cells has been added to the Fig. S2 as described above.

8) Line 146-7: The authors mention induction of Ca²⁺-spiking induced by Nod factors which supposedly resemble that of the slower response with Rhizobia, but this is not shown. Nod-factor response (in the absence of Rhizobia) would highlight the "susceptible zone" in WT and sunn2 mutants and should be shown. This is essential to define the 'susceptible zone' in non-infected (i) and subsequent infected cells.

>>> The response of root hairs from the nodulation susceptible zone (just below the infection zone) has been extensively studied previously (Charpentier et al 2018, a review) and shown to have indistinguishable calcium profiles whether challenged with *S. meliloti* or purified Nod factors (see e.g. Weiss et al 2002). We have clarified this in the text on lines 156-158. However, it is completely out of the scope of the paper to add Nod factor treated root responses.

9) Lines 152-153: Yes, GFP intensity for Ann1 labelling does appear higher in IT containing cells as opposed to RHE, however, the authors shouldn't say "clearly" without some measurements and more images in supplementary, not just one representative image. Additionally, this is not a direct measure of Ann1 expression, only protein accumulation, it would be more accurate to say that IT correlates with increased ANN1 accumulation in the root hair. However, this should still be quantified using fluorescence intensity measurements of the Ann1-GFP fusion and qRT-PCR/western blotting.

>>> Indeed, we changed our phrasing to "protein fusion accumulation". We now consolidated our statement by adding the quantification of the relative fluorescence intensity of the MtAnn1-GFP fusion (Fig. S7). Previous published root hair transcriptomic data agree with these observations (Breakspear et al 2014). However, it is not realistic to perform qRT-PCR/western blotting on a single infected root hair. Note that we are dealing here with dynamic and non-synchronous modifications that occur in individual cells, and are thus not accessible by qRT-PCR or Western blotting.

10) If the authors also wish to mention Ann1 expression levels then this should be quantified directly using qPCR or GUS under the native pAnn1 and quantify expression, as was done later in the paper. Additionally, we assume this was conducted in a wild-type background, not an *ann1* mutant- in which case, there is also native Ann1 expression which is not being quantified through this GFP visualisation, therefore please do not give this emphasis as proof of "higher" Ann1 expression.

>>> See the response just above. Besides, promoter-GUS does not provide dynamic information, and is not adapted here neither. We have included the GFP fluorescence relative quantification data (Fig. S7) and changed the description in the text, lines 189-192.

11) Figure 2 "Ca²⁺ spiking amplitude drops along root hair IT development." - This figure only shows Ca spiking data for the *sun-2* mutant at 2-4 dpi whilst WT is not shown. The authors should provide evidence that the same observation applies under WT conditions at this time point (albeit at lower frequency) to prove that it's not a specific to the *sun-2* mutant phenotype. Instead of providing comparable data for WT in Fig 2 the authors instead refer to Fig S1 which is not comparable to Fig 2 as Ca spiking is recorded at a different stage (ie the nodulation susceptible zone at 1dpi). Indeed, in Fig S1 the authors should also include quantification of Ca²⁺ spiking in the *sun-2* mutants at 1dpi rather than showing Ca spiking in WT (A17) only.

>>> The Ca²⁺ amplitude drop is reproducible in *sun-2* and in wild-type backgrounds and this is shown in Fig. 2 for *sun-2* and wild-type R108S (Fig. 7). As mentioned above, Fig. S2 now comprise the *sun-2* data for non-infected root hairs of the nodulation susceptible zone.

12) Line 159-161: "This labelling is specific to one or a few cells around the infection site (Fig. 3A-C). Strikingly, these MtAnn1-labelled cells consistently show low frequency Ca²⁺ spiking, whereas nearby cells not labelled with MtAnn1 do not show any Ca²⁺ spiking (Compare C1a-c in Fig. 3)."

- There are several nuclei showing red fluorescence in cells from NR-GECO1 below MtAnn1-GFP labelling and they all lack Ann1-GFP signal; moreover, cells to the left and right of MtAnn1-GFP is not clearly visible and C1c shows only half of a cell. A single spike in C1a and C1b profiles is also not convincing. How frequently was this observed? This is a key finding of the paper yet quantification of Ca responses is not provided.

>>> Cells with red nuclei are those expressing the GECO fusion under the p35S constitutive promoter, and these may or may not spike. The beauty of this data is that it is only the cells that are MtAnn1 positive that are spiking. As the low frequency calcium spiking is spaced in time it can happen that we do not record all peaks in a 10 minutes time frame. Our data are consistent with those obtained with the FRET-cameleon (Sieberer et al., 2012) and are very convincing because non-primed outer cortical cells in the vicinity never spike. We have now strengthened this conclusion by providing data from different sites and independent experiments in plants expressing NR-GECO (Fig. S3) or NR-GECO1+MtAnn1-GFP in both A17 and *sunn* backgrounds (Fig. 3 and S8). In all cases, low frequency calcium spiking is consistently observed in outer cortical cells below an infected root hair cell. The description of these modifications can be found in lines 159-167 and 193-201 (see Figs. 3, S3, S8).

13) Moreover, localisation of a cytoplasmic marker (e.g. ER marker as mentioned above) should also be shown in these cells to confirm that these cells have cytosolic hallmarks of primed cells rather than using MtAnn1-GFP to demonstrate cytosolic rearrangement.

>>> In these experiments we are imaging three fluorescent markers simultaneously: the red NR-GECO1, the green MtAnn1-GFP and the blue CFP-labelled rhizobia. It would be very challenging to add a fourth fluorescent marker due to possible fluorescent signal overlaps. We do not really understand why the reviewer persists on this point. MtAnn1 is an excellent cytoplasmic marker, validated previously (de Carvalho-Niebel et al 2002) and in this study (see Fig. S4-S6) to label cytoplasmic bridges similarly to ER fluorescent markers.

14) Line 175-176: "This mutant (*ern1*) shows Ca²⁺ spiking and cytoplasmic-bridge formation (visualized with MtAnn1-GFP) in early infected root hairs". As mentioned above, localisation of a cytoplasmic marker should also be shown in cortical cells below IF to confirm that these cells have cytosolic hallmarks of primed cells. Since MtAnn1-GFP localises to only a few cells and in the presence of Ca²⁺ localises to membranes it not a suitable marker for cytosolic priming and does not evidence that adjacent cells are not undergoing cytoplasmic priming.

>>> We believe the reviewer misinterpreted the definition of annexins. Annexins are calcium binding proteins, largely localized at the cytoplasm, but can associate with phospholipids when calcium rises, under certain stimuli. As we have previously shown by confocal imaging and immunolocalization (de Carvalho-Niebel, 2002) and in this study by co-localization studies with ER-markers (Fig. S4-S6), MtAnn1 is a cytosolic protein, partially localized to ER endo-membrane system which reliably labels cytoplasmic bridges. So, we see no point of using another cytoplasmic marker here.

15) Also, appropriate controls are lacking and therefore the conclusion that "ERN1 is not required for promoting pre-infection priming, but is somehow required to spatially restrict it" is not supported.

>>> We realized after reviewers' comments that Table 3 in ancient Fig. 3 and some of the mutant description data were confusing. So, we have simplified this section (see changes in lines 204-213). Major conclusions about *ern1* is that calcium spiking responses and MtAnn1-GFP signal are still observed in this mutant despite the absence of ERN1. So, we can reasonably conclude that ERN1 is not

essential to generate this priming response. We do not really understand which appropriate controls the reviewers are referring to? As the *ern1* mutant is from the A17 genetic background (Middleton et al 2007), the appropriate control, the A17 wild-type plant, is already shown in Fig. 3.

16) Fig 3 shows MtAnn-GFP translational fusions in WT (A17) at 3-4dpi and this time the authors observe cells with strong fluorescent signal; however this is only shown in WT (A17) but not in all mutant lines. Indeed, Fig 3D only shows a schematic representation of Ca²⁺ spiking in Sm responsive cells in WT (A17) with a table summarising responses in mutants. Ca²⁺ spiking responses are shown as (+) vs (-) with the response in *ern1* mutants in zone (3) shown as (++). What does this mean? How is this quantified and what values are presented by the (+). Is the observed (++) in zone 3 statistically significant?

>>> There is a misunderstanding here, the numbers do not represent relative levels but whether the responses were observed (+) or not (-) in different mutant backgrounds, as clearly stated in the legends. As this table was causing confusion, we removed it and simplified this section. We thus removed all early signalling information from the mutants and focused the data on infection-related events only. Data is described in lines 204-227 and 250-263 (see Fig. S9 and Fig. 4).

17) Imaging data ie localisation of MtAnn-GFP and Ca²⁺ spiking in *ern* and all other mutants should be shown in a main or supplementary figure rather than only showing image data for *ern1* (Fig S2) and these should be shown next to comparable WT (R108) lines.

>>>There is a misunderstanding here, the mutant lines *ern1*, *dmi3* and *nin*, are EMS or fast neutron mutants, all obtained in the A17 genetic background. The wild-type A17 control is thus shown in Fig. 3.

>>> We have now simplified this section (see Fig. S9 and associated description in lines 204-227) and cleared stated in the text that the wild type control is shown in Fig. 3.

18) The authors should show spiking amplitude and frequency in Fig S2D rather than individual spiking patterns in C1 cells of the *ern1* mutant as well as in and adjacent cells lacking MtAnn1-GFP for all mutants.

>>As mentioned above, we simplified this section and removed the table. The questions we wanted to address with the mutants were whether or not we could still see calcium spiking and MtAnn1 labelling (+ remodelling) in these backgrounds. This is a YES and NO question, and the results shown in Fig. S9 clearly answer it. The detailed analyses of the calcium profiles in these mutant backgrounds is another question beyond the scope of the paper.

>> Modifications of the mutant section (shown in S9) are found in lines 204-227. Due to space constrains we cannot show mutant images together with the wild type in the same Figure. The corresponding wildtype control (A17) is shown in Fig. 3.

19) Again, appropriate cytosolic markers should be included to make the link between spiking and cytosolic priming and comparable EV and WT (R108) controls should be included.

>>> Due to a misunderstanding of the reviewer (MtAnn1 is a cytoplasmic protein!), the reviewer keeps asking the same question about the need of using a cytoplasmic marker. Please see our answers in points 1, 14 and 15.

>>> As mentioned just above in point 17, the mutants are in the A17 background, so the A17 wild-type control is shown in Fig. 3. The reviewer tells us to use an EV control but doesn't develop the usefulness of this control in this context. Controls need to be used in the light of the biological question being addressed. There's no point in blindly using a control without questioning its usefulness. In this particular case, we are not doing a functional analysis. We are asking how the fusions (MtAnn1-GFP or NR-GECO) are dynamically regulated during infection in WT vs mutant, and we cannot answer this question if we do not express the fusions. In addition, control GECO roots clearly show no responses (Fig. S2). The *MtAnn1* promoter, which drives expression of the MtAnn1-GFP fusion, is not active in root hairs under control conditions (see Fig. 4A).

20) Fig 3D also summarises the localisation and responses in *ern1ern2* mutants, but the image data is not shown nor are WT (R108) controls included.

>>> As mentioned above, *ern1 ern2* is from an A17 background, respective control in shown in Fig. 3. To simplify and for space constrains we have removed the *ern1 ern2* mutant data.

21) Fig S2 shows MtAnn1-GFP in the *dmi3* epidermal specific complemented line, however, no controls are included – e.g. images for EV transformed *dmi3* mutants should be provided (rather than referring to responses in a published *dmi3* report which is not directly comparable). Comparable WT controls from same genetic background should also be included.

>>> CCaMK/DMI3 is critical for PPA formation in *Medicago* (Genre et al 2009), but it is not known whether it is also required for PIT formation. Since *dmi3* is impaired in root hair infection, it is not possible to access PIT formation in this genetic background. We therefore took advantage of stable transgenic lines complemented with DMI3 only in root hairs (Rival et al. 2012) to investigate whether Ca²⁺ spiking and MtAnn1-GFP could still label cortical cells in this line. As the genetic background of this mutant is A17, the corresponding control is the wild-type A17 line shown in Fig. 3. The data on the complemented *dmi3* line have been completed (Fig. S9) and the corresponding text in the results section (lines 214-227) has been improved to better justify the use of this line and to make the link to its corresponding A17 control (shown in Fig. 3).

>>>As just discussed in point 19, controls need to be used in the light of the biological question being addressed. An empty vector control (expressing neither the MtAnn1-GFP fusion nor NR-GECO) would not provide any valuable information.

22) Similarly spiking amplitude and frequency in Fig S2D should be quantified rather than showing single/individual spiking patterns in C1 cells (no quantification of spiking frequency or amplitude is provided and how often these patterns are observed is not clear).

>>> See our response to point 18.

23) Line 189-192:” The *ern1 ern2* and *dmi3* mutants, which are totally deficient in root hair bacterial entrapment and infection, show Ca²⁺ spiking in root hairs (ref49 and Supplementary Fig S2E and Movie 3) but no clear signs of cytoplasmic remodelling in root hairs or cortical cells”. The authors cannot

conclude that there's a lack of cytoplasmic remodelling in mutants based solely on the expression of MtAnn1-GFP. Appropriate cytoplasmic markers (as mentioned above) are needed in these mutants to verify that cells adjacent to GFP signal lacks cytoplasmic remodelling whilst those with Ann1-GFP has cytoplasmic priming. In the absence of appropriate controls and image data conclusions drawn by the authors are not supported by the data.

>>> As discussed above, MtAnn1-GFP is an excellent cytoplasmic marker to detect priming responses. For the complemented *dmi3* line, which no longer shows pMtAnn1:MtAnn1-GFP fusion expression in the cortex, we have now included analysis of a cytoplasmic GFP marker (Fig. S9). This clearly confirmed the absence of priming in the cortex of the complemented *dmi3* line. See description of changes in text lines 214-227 and Fig. S9.

24) Line 197-199: "While MtAnn1-labelled cytoplasmic bridges were strongly detected in infected root hairs of complemented *dmi3*, no pre-infection priming events were observed in adjacent cortical cells (no signs of MtAnn1 labelling, Ca²⁺ spiking or cytoplasmic remodelling, Supplementary Fig S2E-F)". As mentioned earlier the authors need to include a cytoplasmic marker (ER or Actin) to confirm lack of cytoplasmic remodelling in *dmi3* mutants vs WT (R108).

>> Due to the reviewer's misunderstanding of the cytoplasmic localisation of MtAnn1, he/she continues to ask for the use of another cytoplasmic marker. This has been explained several times in points 1, 13, 14, 19 and 23. We have provided clear information in Fig. S9 on the absence of cytoplasmic remodelling in the complemented *dmi3* line by using a GFP marker (just explained in point 23).

25) Since the authors show TEM images for *ann1* mutants to demonstrate lack of cytoplasmic bridges they should similarly include TEM in *dmi3* and restoration of cytosolic bridge in complemented lines.

>> Cytoplasmic bridges are formed in *ann1* mutants and due to TEM analysis, we could find the relative differences in bridge composition/organization compared to the wild-type line. There is no cytoplasmic bridge formed in the cortex of the *dmi3* complemented line (as clearly shown in Fig. S9 and described in lines 214-227), so it is incomprehensible why the reviewer requests this analysis. He/she also clearly does not realize how work-heavy TEM analysis is.

26) Line 191-192: "...these root hairs show only residual (in *ern1 ern2*) or no (in *dmi3*) MtAnn1-GFP expression". Please confirm in which Fig these images are shown?

>> As mentioned above, we modified the mutant section to focus only on mutants which can provide information on priming responses in the cortex (see explanation in lines 204-227).

27) Line 193-194:"Thus, the formation of MtAnn1-labelled cytoplasmic bridges". It has not been proven that MtANN1 is part of the cytoplasmic bridge (ie. this is a circular argument). Experiments such as IGL-TEM (immunogold labelling) should be performed to show MtANN1 is part of the cytoplasmic bridge

>> As clearly described in the manuscript and explained numerous times in the above points (1,13,14,19,23,24) MtAnn1 is an excellent cytoplasmic marker labelling cytoplasmic bridges, proved by immunolabelling and confocal images (de Carvalho-Niebel et al 2002 and this work). Please find the modifications in the text (lines 168-182) and check Fig. S4-S6.

28) also see later comment – live cell confocal images showing lack of cytoplasmic markers in *ann1* mutant should be shown.

>> GFP-ER labels cytoplasmic bridges *in vivo* but does not provide sufficient resolution for allowing distinguishing reliable differences as TEM analyse does. The cytoplasmic bridge is still formed in the mutant, but as our TEM data shows, changes are in its composition not in its ability to form (see Fig. 7). Illustrating these differences by *in vivo* confocal microscopy could be better achieved using fluorescent markers labelling small vacuole-like components observed in TEM analysis. However, since we don't know the nature of these components, this question is a totally new project.

29) Line 226 – 235 “the *in vivo* dynamics of a fluorescent GFP-ER fusion was live tracked in *nin* complemented or not in the epidermis by the expression of a *pEXPA:NIN* construct” and Fig 4 addresses the dependence of MtAnn expression on NIN. Fig4 A-E shows *pMtAnn1:GUS* fusions at infection sites. To make the link between Ca²⁺ spiking, Ann1 levels and cytoplasmic remodelling the authors should transform *nin* complemented mutants with an appropriate cytosolic marker and co-localise with Ann1-GFP expression in IT, epidermal and cortical layers. - ...”or not” Please could the authors show respective EV controls of all constructs

>> Again, this is a circular point due to the reviewer's clear misunderstanding of the localisation pattern of MtAnn1. MtAnn1 is an excellent cytoplasmic marker for the studies we propose here (see comments above in points 1,13,14,19,23,24,25). Except in the case of *nin*, as the expression of *MtAnn1* is under the transcriptional dependence of NIN (shown in Figures 4A-G and in Breakspear et al, 2014). Instead, we used the GFP-ER marker under the regulation of *pENOD11*, which is not under the dependency of NIN at infection sites (see Fig. 4).

>>>The reviewer's request to add another cytoplasmic marker is incomprehensible. We have used two cytoplasmic markers: GFP-ER (that labels the cytoplasmic ER network) and Ds-Red (cytosol), which together allow strong conclusions to be drawn about the absence of cytoplasmic bridges in the cortex of *nin* (see Fig. 4 and S11).

>>> It is unclear what the reviewer means by "EV controls of all constructs". We have used a very relevant control here, which is the *nin* mutant co-transformed with the *pENOD11:GFP-ER* fusion and a *pEXPA-GUS* control construct for comparison with epidermally complemented *nin* transformed with the *pENOD11:GFP-ER* fusion and a *pEXPA-NIN* construct.

>>>For details of the modifications made in the text and figures, see lines 250-263 of the text and Figs. 4 and S11.

30) In Fig. 4F-K an ER marker is initially used as a cytoplasmic marker (Fig 4F and I) and the authors then switch to DsRED in epidermal complemented lines. In other figures MtAnn1-GFP is used to show cytoplasmic bridges. This is confusing and inconsistent. It also makes it hard to compare cytoplasmic priming in *nin* mutants versus complemented lines where DsRED is used. Use the same cytoplasmic marker (ER, actin, mitochondrial) across experiments (see comments above).

>>> It is not inconsistent or confusing, please read carefully the explanations in the text (lines 250-263). The *pMtAnn1:MtAnn1-GFP* fusion construct is used as a cytoplasmic marker in the paper, except in *nin*, because *MtAnn1* expression at rhizobial infection sites is NIN-dependent (see Figure 4A-G). We therefore used the cytoplasmic GFP-ER fusion (under a promoter not regulated by NIN) and the

cytoplasmic Ds-Red (under p35S) markers to assess cytoplasmic remodelling in *nin*. See explanation above in point 29.

31) Fig 4E: Deletion of the NIN DNA binding domain does not prove direct interaction with the promoter of *MtAnn1* as implied in line 221-222 but shows that it is needed to activate *MtAnn1* expression. This activation could be direct or indirect. Therefore, the sentence should be revised to “These data establish NIN as a possible direct or indirect regulator of *MtAnn1*.” Please define NIN in figure legend.

>>>Rhizobia-induced activity of the *pMtAnn1:GUS* fusion is NIN-dependent (see Fig. 4A-G) and this is consistent with previous transcriptomic data (Breakspear et al., 2024). This genetic dependency together with the ability of NIN to transactivate *pMtAnn1:GUS* activity in both *M. truncatula* (See Fig. 4E-F) and in *N. benthamiana* (Fig. 4G) strongly favour NIN as a direct regulator of *MtAnn1* (which additionally comprises NIN binding sites in its promoter). We were very careful not to make this overstatement, we only suggested that NIN may act as a direct regulator of *MtAnn1* (it is not a statement but a hypothesis). Please see the modified sentence in lines 247-249.

32) In Fig S3, WB shows a smaller band that cross-react with the anti-HA tag suggesting there might be a truncated HA-tagged NIN protein product that could be responsible for the transactivation of *pMtAnn1:GUS*. Please could the authors comment on this?

>>>There might be a truncated form too, but which is far less represented than the non-truncated form. The main objective here was to validate that transactivation of the gene is only seen in the presence of NIN harbouring its DNA binding region (which is clearly demonstrated!). It is unknown if this residual truncated form has any transcriptional activity, but this information does not seem very relevant on the scope of the paper. Besides, transactivation of the fusion in *Medicago* in the absence of rhizobia (Fig. 4E-F) corroborate the conclusion of the strong activation of the *pMtAnn1:GUS* fusion by NIN.

33) CHIP-PCR and DNA gel shift assays are needed to verify the possible direct interaction between NIN and the *MtAnn1* promoter.

>>> We clearly concluded in the paper that NIN is upstream *MtAnn1* for its rhizobia-induced transcriptional regulation, but we did not make any absolute statement that NIN has direct DNA binding activity towards the promoter of *MtAnn1*. Determining whether NIN is acting as a direct or indirect regulator of *MtAnn1* would indeed require additional DNA binding experiments, but this is largely beyond the scope of this work.

34) Fig S4 should include molecular validation of *ann1-1/2/3* mutants.

>>> The molecular validation of the mutants was done by PCR and DNA sequencing. This data was not included (but can be included under request). More importantly, we have provided Q-RT-PCR data on mutants compared to wild-type R108 and R108S lines (see Fig. S12).

35) Fig S4B shows two lanes labelled as WT. To avoid confusion label first lane as R108WT and R108ann1 or similar.

>>> Thank you for pointing that out, there was a typo in this panel. To avoid confusion, we renamed wildtype lines to R108 and R108S (for the WT sibling derived from the same mutagenized population).

36) Fig S4C lacks nodule number in the *ann1-1* mutant – please could the authors include this data as significant reduction in expression is shown in FigS4B.

>>>This data is now presented in Fig. S13. Nodule number is now described for all mutants in the figure legend.

37) Figure S4D shows Medicago plants with colonised roots. *ann1-2* and *ann1-3* appears to have reduced size/developmental phenotype compared to WT – please could the authors quantify plant biomass in mutants. Include mock inoculated controls and the *ann1-1* mutant.

>>> Growth of the mutant lines was not visibly affected in non-inoculated conditions, as seen along the successive rounds of propagation for these lines. We now added the biomass quantification data for shoots and roots of inoculated plants from both wild-type and mutant plants in the new Fig. S13.

38) Lines 255-284 (and figures 5G-N): Was more than one nodule imaged for each figure- can it be confirmed that these images are representative and could some additional images be included in the supplementary. If this is the case and multiple nodules were imaged to assess the sizes of the different nodule zones, it would also be nice to show some statistical analysis to prove these are significantly differently sized between *ann1-3* and WT.

>>>Yes, we have analysed 1 μm nodule sections (n=7 in wild-type and n=8 in mutants) from different nodules, from different plants and independent experiments, and these have now been included in Fig. 6 and S14. While nodule differentiation was quite homogeneous in wild-type nodules, zonation in mutant nodule was variable and with a tendency of having large interzones relative to the nodule size. Thanks to the reviewer we compared via ImageJ the relative size of IZs of the different nodules, and interestingly, we do see statistically significant differences between wild-type vs mutant nodules. These data were included in the new Fig. 6 and S14, and described in lines 294-311.

39) Also, why was only *ann1-3* chosen for further phenotypic analysis? Some justification in the text would be preferable, especially since it appears from Supplementary Figure S4F that the differences in nodule phenotype e.g. colour and form are a bit more pronounced when comparing *ann1-2* and WT than with *ann1-3* versus WT.

>>>*ann1-2* and *ann1-3* mutant lines showed similar phenotypes in terms of reduced infection, multiple ITs on NPs (also seen in RNAi plants), nodule appearance and size and reduced *MtAnn1* expression levels (see Fig. 5, S12 and S13), so we estimate that we can select one of them for performing work-heavy analyses of thin-ultrathin sections (of 1 μm or 80 nm) for resolutive bright field and TEM analyses. The choice of the *ann1-3* allele was also based on the fact that we can partially revert VPY expression levels (lower in *ann1-3*) when the mutant was complemented with a *p35S:MtAnn1* construct (Fig. S12). Nodule structures are variable in both mutants, and they are consistently more similar to one another than to wild-type controls (R108 and R108S) (S13). The variability in nodule differentiation is now more clearly illustrated in different sections of *ann1-3* nodules (see new Fig. 6 and S14).

40) Authors should include images of *ann1-2* and *ann1-3* functional complemented MtANN1-GFP to show restoration of nodulation comparable to EV or R108 WT controls.

>>>Complementation studies with mutants which partial phenotypes it is not an easy task. However, we did show by molecular analysis that the reduced expression of *VPY* in the *ann1-3* mutant (n=19) was partially restored in complemented roots expressing *MtAnn1* under the p35S promoter (n=17), compared to the wild type R108S control (n=22). The data is from two independent experiments and is shown in Fig S12F-G.

41) Line 259-260: *ann1-3* nodules show an abnormally large IZ like region, with over-accumulation of amyloplasts and less clearly defined zone III (Fig. 5H and J) and Fig 5 legend “The amyloplast-rich IZ and zone III are clearly defined in WT, while *ann1-3* nodules (H, J) show a much larger IZ like region, with starch over-accumulation and no zone III comparable to that observed in WT nodules”. This is an interesting finding. Could the authors include iodide staining confirm the overaccumulation of starch in the IZ in *ann1-3* mutant vs WT and the *ann1-3* complemented line. - To confirm overaccumulation of starch the authors should also quantify starch levels in nodules from WT, *ann1* mutants and *ann1* complemented lines.

>> Starch accumulation is tightly correlated with nodule IZ while in ZIII starch only accumulates in the periphery of infected cells and not in non-infected cells. As the mutant has more pronounced IZs we do see a tendency to see starch granules more prominently. However, we did not find an easy way to quantify these differences. Iodine in our hands led to background staining, not easy to quantify. So, we decided to lower down this hypothesis and only focus on the description of the IZ data, significantly different between wildtype and mutants. See detailed description in lines 294-311.

42) Line 264-268. Although *ann1-3* nodules include IC cells with apparent differentiated bacteroids (Fig. 5L, N), they comprise small vesicles (asterisks in Fig. 5N) that are normally found in IZ rather than zone III of WT nodules⁵⁴. Fig 5 shows beautiful TEM images, but TEM images should be shown for all nodule zones in WT to make it clear if there is an arrest of a zone compared to with the *ann1* mutant. In other words, does the IZ form normally in *ann1* but does not transition into a fully defined zone III in the *ann1* mutants? The TEM image in Fig 5K,M shows WT zone III but not WT IZ so it's difficult to determine if the IZ in WT and *ann1* are comparable.

>> TEM analysis of nodule infection zone (see Fig. 7), interzone (see Fig. 6) and zone III (see Fig. 6) was included for both wild-type and mutant backgrounds. From the EM analyses we always see this white vesicle-rich environment in mutant nodules (in IZ or IZ/ZIII regions), indicative that the IZ is defective since the beginning of IZ differentiation. As the ZII is also altered in *ann1-3* mutant nodules, this may be at the origin of the defects in underlying zones. Please check lines 305-311 and Fig. 6 for data description.

43) Line 281-282 “Root hair ITs are formed in mutant roots at 4 dpi with rhizobia, but at a lower frequency compared with WT (Fig. 6A)”. Although the authors show stained roots of both *ann1-3* and *ann1-2* quantification data is only shown for the *ann1-3* allele. Please could the authors include quantitative data for *ann1-2* allele and ideally also for the *ann1-3* complemented roots?

>>>We now provide a complete analysis of early root and nodule infection in both mutant alleles relative to wild-type plants. We validated by Q-RT-PCR the knockdown of *MtAnn1* in both mutant

alleles (shown in Fig. S12) and through a detailed phenotypic analysis of nodule shape, nodule number, root and shoot weight (now shown in Fig. S13) and rhizobia root and nodule infection (Fig. 5). Both quantitative data and representative images are shown for both alleles. Both alleles consistently showed reduced infection of NP with multiple ITs on top, similarly seen in RNAi plants.

>>>Description of this new data is now included in lines 268-284 and Fig. 5, S12 and S13.

44) Fig 6D-I shows images of infected roots stained for rhizobia; however, images are not comparable across all genotypes. For example, Fig 6D, F and H (WT) and 6E, 6I (*ann1-3*) should include staining of *ann1-2* at a comparable stage and magnification.

>>> Pictures of all genotypes (R108, R108S, *ann1-2* and *ann1-3*) at 4 dpi and 10 dpi with rhizobia were included in the new Fig. 5.

45) Similarly, Fig 6F (WT) and 6G (*ann1-2*) should include a low magnification ie comparable image of *ann1-3* mutant. Images of *ann1-3* complemented roots should also be included.

>>> Pictures of all genotypes (R108, R108S, *ann1-2* and *ann1-3*) at 4 dpi and 10 dpi with rhizobia were included in the new Fig. 5.

46) Fig S6 is confusing as X-gal staining from bacteria and the GUS reporter both result in blue nodules and therefore bacterial infection cannot be compared between the RNAi knock down and control lines.

>>> Plants were transformed with a *pMtAnn1:GUS* fusion as control, but there is no GUS staining in the images presented, only lac-Z staining of the bacteria (see figure legends). This data is now included in Fig. 6.

47) RNAi knock down lines show a reduction in nodule number (and severity) compared to control lines, however, in Line 247-248 the authors note that "Phenotyping of *ann1-2* and *ann1-3* did not reveal any major changes in.... ..nodule number compared with the WT lines at 21 dpi (Supplementary Fig. S4C)." Please could the authors comment on this discrepancy? Could the RNAi construct be targeting additional Annexins to Ann1 that might be required for nodule development? To confirm this the authors should show expression levels of Ann1 and other members of the Annexin family in the RNAi lines.

>>> Yes, it is correct, differences in nodule number was only seen in RNAi plants. This is likely due to the downregulation of *MtAnn1* together with its close homologue *MtAnn2*, as testified by Q-RT-PCR data (shown in Fig. S12). This is now clearly described in lines 285-293 and Fig. 5 and Fig. S12.

48) Following on comments relating to Fig 6 above, quantitative analysis of ITs in-RNAi lines should be shown alongside that of the knockout lines (*ann1-2* and *ann1-3*).

>>> Due to time constrains we could not perform quantification data in RNAi transgenic roots but we did include new data with quantification of infection events in the two mutant alleles. Moreover, quantification in RNAi roots showed highly significant differences in nodule/NP infection levels, and this led to the formation of either infected nodules (with the same multiple ITs above as seen in

mutants) and abnormal NPs with poor or no colonization (as shown in Fig. 5), all convergent with the conclusion of reduced infection rates upon *MtAnn1* and/or *MtAnn1/MtAnn2* downregulation.

49) Fig 7 A-B: Spiking amplitude for WT and *ann1-3* are shown in separate graphs which would indicate that these experiments were not carried out at the same time? Spiking amplitude in *ann1-3* RHE seems reduced compared to WT although it's unclear if this reduction is significant compared with WT. Indeed, spiking amplitude in *ann1-3* IT seems comparable to that of WT. Are these differences significant. Especially considering the low number of biological replicates used in RHE and IT with WT compared with mutants.

>> We added the data in separate graphs to better illustrate the RHE to IT difference per genotype. Spiking amplitude in wild-type RHE versus *ann1-3* RHE is not significantly different (one-tailed t-test: $t=0.8814$, $df=20$, $p=0.1943$: **NS**). However, spiking amplitude values in wild-type ITs versus *ann1-3* ITs was significantly different (one-tailed t-test: $t=1.861$, $df=11$, $p=0.0448$, *significant). Numbers are low due to the difficulty of doing these analyses in the R108 background, where early infection events are less represented compared to *sunn*. However, the data are from 3 independent experiments, and the calcium amplitude drop seen in R108S was consistently seen in *sunn* (and the same trend was seen in A17, but we only have a few sites).

50) Lines 301-303: "Ca²⁺ spiking and *MtAnn1* dynamics accompany cytoplasmic bridge formation (Fig. 2-3), so we wondered how *MtAnn1* mutation and deregulated Ca²⁺ spiking amplitude in *ann1-3* (Fig. 7) could affect cytoplasmic bridge formation. We thus analysed the cytoplasmic composition of infected cells in apical nodule zones of WT and *ann1-3* by resolutive TEM." To illustrate the role of *Ann1* in cytoplasmic bridge formation the authors should include confocal data with cytoplasmic markers (as suggested earlier) in the *ann1* mutant (and *MtAnn1*-GFP) complemented lines which would be comparable to data shown in Fig 2-3.

>>> These analyses are not realistic, since *MtAnn1* mutation affects cytoplasmic column composition not its formation. Column formation in *ann1-3* was additionally verified using the *pENOD11:GFP-ER* cytoplasmic marker (co-expressed with the NR-GECO1 sensor), but this was not included in the paper to not over-complexify the main message, as it does not bring more information than the TEM analysis. Please see additional comments in point 28.

51) At the TEM level the extent of cytoplasmic restructuring is difficult to confirm because of the dynamic nature of these structures. Also, are cytoplasmic rearrangements found in primed outer cortical cells (as shown in Fig1C-E) lacking in *ann1* mutants?

>>>The TEM analysis in the wild-type context has been completed, and we now provide data on this priming state at independent sites (see Fig. 1 and S1). As the reviewer clearly pointed out, due to the dynamic nature of these structures, it is very difficult to capture these events, especially during early root colonization stages (e.g. ~ 200-250 sections per site were analysed to capture one primed cell event). As the expression of *MtAnn1* is maintained in the infection zone of nodules (shown by RNAi in situ hybridization, de Carvalho-Niebel et al 1998), the analysis of mutant nodules, which are more accessible, was a relevant alternative to study this process in the mutant context.

52) Fig 8A-B shows *MtAnn1*-GUS activity in AMF colonised roots. Although there appears to be a

correlation between GUS staining and WGA-Alexa488, since both are shown in blue it is difficult to distinguish GUS from fungal structures. Please could the authors use a different false colour for WGA staining and clearly highlight the presence of arbuscules vs GUS signal. Empty vector controls should also be shown as well as mock inoculated roots to distinguish reporter activity from background staining.

>>>The colour scheme was modified. The WGA staining now appears in yellow.

53) Fig 8E-H shows ink stained roots colonised by AMF. Interestingly ann1-3 arbuscules appear small compared to WT. Please could the authors include WGA staining to show arbuscule morphology and to confirm if arbuscules are arrested in development or show accelerated turn over. Quantification of arbuscule size should be included.

>>> Mycorrhized roots were WGA stained and quantification of arbuscule size was performed. This revealed slight but significant differences in arbuscule size in wild-type versus mutants. The quantification data and morphology of WGA-stained arbuscules are now show in Fig. Data description has been amended in lines 362-367 of the text.

54) Supplementary Figure S8: More information is needed in both the legend and the text to understand what outgroups were used to root the phylogeny tree (SAR outgroups where there is no previous information of what SAR means).

>>> Five genomes of the SAR (*Stramenopiles/Alveolata/Rhizaria*) lineage, as representative species outside of the green lineage were used as outgroups (Radhakrishnan et al., 2020), as now described in the legend of Fig. S16.

55) Line 418-421: "As MtAnn1 seems to partially co-localize with the peri-nuclear ER network in symbiotically activated cells, it could mechanistically play such a regulatory role, which would explain the disturbed Ca²⁺ signals and ER phenotypes caused by the MtAnn1 mutation." In the absence of IGL-TEM showing MtAnn1 localisation to the peri-nuclear ER network and the absence of live cell imaging using appropriate FP-tagged ER biomarkers in the ann1 mutant background the authors cannot draw this conclusion.

>>>We showed previously the immunolocalization of MtAnn1 to peri-nuclear cytoplasmic regions (de Carvalho-Niebel et al, 2002) and we now show here in Fig. S4-S6 the co-localization of MtAnn1-GFP to cytoplasmic bridges, similarly to ER fluorescent markers. See also explanation to point 1. This discussion point is now developed in lines 434-437.

56) Similarly in Line 444-447: "Annexins are among Ca²⁺ sensing proteins involved in endomembrane trafficking in plants, either acting in conjunction with the membrane fusion machinery or regulating exocytosis. Thus, MtAnn1 may directly or indirectly impact Ca²⁺-regulated vesicle trafficking and/or vacuole morphology." The causal link between Ann1 and cytoplasmic remodelling is not clearly demonstrated, and this could be shown through FP-tagged markers for exocytosis (e.g. Exo70, Vapyrin) and/or vesicle trafficking in the ann1 mutant background.

>>>This discussion point is now described in lines 441-445. We do not see why we cannot advance this discussion point here. A discussion session is partly about hypotheses for future work.

Small comments:

57) Line 290-291 “decided to assess if MtAnn1 mutation could somehow impact symbiotic Ca²⁺ spiking.” Delete the word ‘somehow’

>>>OK (modified in lines 315-316).

58) Fig. 2 legend – line 1159(RHE, n=23 in A and n=24 in F) – does the authors mean(RHE, n=23 in E and n=24 in F).

>>>Thanks, edition was done.

59) Line 1269 (please check other Fig legends): “in genetic backgrounds mutated (ann1-3) or not (WT) in MtAnn1” please rephrase to ‘in ann1-3 mutant or WT (A17)’

>>>OK, done (with the correct genetic background, R108).

60) Line 393: ...”in which the annexin MtAnn1 appears to be an important component” implies a role for MtAnn1 is shown in vacuolar biogenesis which is not shown; revise instead to ‘in which the annexin MtAnn1 might play a role’

>>>The sentence has been modified (lines 403-406).

61) Line 404-407: “The deregulated Ca²⁺ amplitude profile and defective infection caused by MtAnn1 mutation (Fig. 6 and 7) supports the need for such a strict control and establish a direct functional link between MtAnn1 and the regulation of Ca²⁺ spiking amplitude during rhizobia infection.” A direct mechanistic link between Ca²⁺ spiking and rhizobial infection is not proven in this study. The authors should revise this sentence to “The deregulated Ca²⁺ amplitude profile and defective infection caused by MtAnn1 mutation (Fig. 6 and 7) points to a functional link between MtAnn1 and regulation of Ca²⁺ spiking amplitude during rhizobia infection”.

>>>Thanks, done (lines 415-417).

62) Line 687:”were contrasted with Uranylless and lead citrate...” does the authors mean ...’were contrasted with Uranyl Acetate and lead citrate...’

>>> The information originally given is correct. Uranylless is a non-radioactive substitute for Uranyl acetate.

REVIEWERS' COMMENTS

Reviewer #1 (Remarks to the Author):

All of the positive points noted in my previous review still stand.

The authors have addressed all my comments and concerns. I have just one additional minor point that can be addressed via text editing.

Supplemental Figure 1 legend needs a slight adjustment. The phrase. '.....shows a primed cortical C1 cell before (red square) and during IT penetration' suggests that it is one cortical cell which is not the case. It shows two cells, one before and one during IT penetration.

Reviewer #3 (Remarks to the Author):

I would like to thank the authors for addressing each of the points mentioned in my review of their manuscript and their careful and methodical adjustments to experiments and text where needed. I am satisfied that all points have been adequately addressed and for their clear explanations where questions stemmed from misunderstanding. I would also like to congratulate the authors on their beautiful cell biology and microscopy - these are indeed challenging techniques. Following revisions to the ms I am happy for the manuscript to be published in Nature Communications without further revision.

Reviewer #4 (Remarks to the Author):

The authors have provided explanations and performed experiments to answer the questions raised.

We suggest authors to strengthen the relationship between MtAnn1 and calcium spiking by performing calcium spiking assays in ann1-3 mutant complemented with MtAnn1

Please change Line 320: sunn should be italicized.

Point-by-point response to the reviewers' comments (>>>):

Reviewer #1 (Remarks to the Author):

All of the positive points noted in my previous review still stand.

The authors have addressed all my comments and concerns. I have just one additional minor point that can be addressed via text editing.

Supplemental Figure 1 legend needs a slight adjustment. The phrase. '.....shows a primed cortical C1 cell before (red square) and during IT penetration' suggests that it is one cortical cell which is not the case. It shows two cells, one before and one during IT penetration.

>>> Many thanks to reviewer #1 for his/her very positive feedback. We have changed the sentence in the legend of Supplementary Fig. 1 as requested.

Reviewer #3 (Remarks to the Author):

I would like to thank the authors for addressing each of the points mentioned in my review of their manuscript and their careful and methodical adjustments to experiments and text where needed. I am satisfied that all points have been adequately addressed and for their clear explanations where questions stemmed from misunderstanding. I would also like to congratulate the authors on their beautiful cell biology and microscopy - these are indeed challenging techniques. Following revisions to the ms I am happy for the manuscript to be published in Nature Communications without further revision.

>>> We're delighted with the very positive feedback by Reviewer #3 and would like to thank him/her for acknowledging the quality of our work and recognising the challenging aspects of it.

Reviewer #4 (Remarks to the Author):

The authors have provided explanations and performed experiments to answer the questions raised. We suggest authors to strengthen the relationship between MtAnn1 and calcium spiking by performing calcium spiking assays in *ann1-3* mutant complemented with MtAnn1.

>>> First, we would like to thank Reviewer #4 for recognizing our efforts to answer the numerous questions raised by previous Reviewers #1, #2, #3.

>>>Strengthening the link between *Mtann1* and calcium spiking signatures was requested by Reviewers #1, #2 and #3 in the first review cycle. We strengthened this link by providing new experimental evidence in the revised manuscript, both by live cell imaging (Figures 2-3, and Supplementary Figures 7-8) and mutant analysis (Figure 7), which has been validated by reviewers #1, and #3 in the 2nd revision round. The suggestion of Reviewer #4 to use complementation assays here is very interesting but unfortunately probably not feasible. If the mutant had a marked phenotype (e.g. absence of spiking), complementation of transgenic roots could be 'easily' detected by analysing those that recovered the calcium spiking response. In the case of the *ann1* mutant, complementation can only be assessed by analysis of the calcium spiking amplitude profiles of individual cells, which, as you can imagine, will not be an easy task in the heterogeneous context of complemented roots with different levels of complementation. We are not aware that such an approach has been used successfully before.

Please change Line 320: *sunn* should be italicized.

>>> Modifications were done in line 320.